# Human-specific lncRNAs contributed critically to human evolution by distinctly regulating gene expression

Jie Lin[1,2], Yujian Wen[1], Ji Tang[1], Xuecong Zhang[1], Huanlin Zhang[1], Hao Zhu[1,3,4]*

[1]Bioinformatics Section, School of Basic Medical Sciences, Southern Medical University, Guangzhou, China; [2]College of Biological and Food Engineering, Guangdong University of Petrochemical Technology, Maoming, China; [3]Guangdong-Hong Kong-Macao Greater Bay Area Center for Brain Science and Brain-Inspired Intelligence, Southern Medical University, Guangzhou, China; [4]Guangdong Provincial Key Lab of Single Cell Technology and Application, Southern Medical University, Guangzhou, China

*For correspondence:
zhuhao@smu.edu.cn

**Competing interest:** The authors declare that no competing interests exist.

## eLife Assessment

This **valuable** study uses tools of population and functional genomics to examine long non-coding RNAs (lncRNAs) in the context of human evolution. Analyses of computationally predicted human-specific lncRNAs and their genomic targets lead to the development of hypotheses regarding the potential roles of these genetic elements in human biology. Compared to previous versions, the conclusions regarding evolutionary acceleration and adaptation have become more **solid** by more fully taking data and literature on human/chimpanzee genetics and functional genomics into account.

**Abstract** What genes and regulatory sequences critically differentiate modern humans from apes and archaic humans, which share highly similar genomes but show distinct phenotypes, has puzzled researchers for decades. Previous studies examined species-specific protein-coding genes and related regulatory sequences, revealing that birth, loss, and changes in these genes and sequences drive speciation and evolution. However, investigations of species-specific lncRNA genes and related regulatory sequences, which regulate substantial genes, remain limited. We identified human-specific (HS) lncRNAs from GENCODE-annotated human lncRNAs, predicted their DNA-binding domains (DBDs) and DNA-binding sites (DBSs), analyzed DBS sequences in modern humans (CEU, CHB, and YRI), archaic humans (Altai Neanderthals, Denisovans, and Vindija Neanderthals), and chimpanzees, and investigated how HS lncRNAs and their DBSs have influenced gene expression in archaic and modern humans. Our results suggest that these lncRNAs and DBSs have substantially reshaped gene expression, and this reshaping has evolved continuously from archaic to modern humans, enabling humans to adapt to new environments and lifestyles, promoting brain evolution, and resulting in cross-population differences. The parallel analysis of gene expression in GTEx tissues by HS transcription factors (TFs) and their DBSs indicates that HS lncRNAs have reshaped gene expression in the brain more significantly than HS TFs.

## Introduction

The limited genomic but substantial phenotypic and behavioral differences between humans and other hominids make which sequence changes have critically driven human evolution an enduring

puzzle (*Antón et al., 2014*; *Pollen et al., 2023*). Sequence changes include the birth and loss of genes, as well as the turnover and change of regulatory sequences (*Albalat and Cañestro, 2016*; *Kaessmann, 2010*; *Prud'homme et al., 2007*). Studies have identified human-specific (HS) genes important for promoting human brain development (e.g., *NOTCH2NL* and *ASPM*) (*Evans et al., 2005*; *Fiddes et al., 2018*; *Mekel-Bobrov et al., 2005*; *Pinson et al., 2022*; *Suzuki et al., 2018*). However, gene-centric studies have limitations (*Currat et al., 2006*; *Timpson et al., 2007*; *Yu et al., 2007*), including that genes promoting brain enlargement may not critically determine other traits (e.g., bipedal walking); meanwhile, studies on species-specific lncRNA genes remain limited. LncRNA genes are a major class of new genes in mammals, and HS lncRNAs may have greatly influenced human evolution. On the other hand, multiple regulatory sequence changes important for human evolution have been identified, including new sequences (*Liu et al., 2021*), lost sequences (*McLean et al., 2011*), human accelerated regions (HARs) (*Dong et al., 2016*; *Mangan et al., 2022*; *Prabhakar et al., 2006*; *Whalen and Pollard, 2022*), and turnover of transcription factor (TF) DNA-binding domains (TF DBDs, commonly called DNA-binding motifs) and DNA-binding sites (DBSs) (*Krieger et al., 2022*; *Otto et al., 2009*; *Zhang et al., 2023*). However, no systematic investigation has been reported on lncRNA DBD and DBS.

Studies have generated multiple important findings about lncRNAs and epigenetic regulation. (1) About one-third of human lncRNAs are primate-specific (*Derrien et al., 2012*). (2) Species-specific lncRNAs exhibit distinct expression in tissues and organs of different species (*Sarropoulos et al., 2019*). (3) Many lncRNAs can bind to DNA sequences by forming RNA:DNA triplexes (*Abu Almakarem et al., 2012*), recruit histone and DNA modification enzymes to DBSs, and regulate transcription (*Lee, 2009*; *Kundaje et al., 2015*). (4) Approximately 40% of differentially expressed human genes result from interspecies epigenetic differences (*Hernando-Herraez et al., 2015*). Thus, besides HS TFs and their DBSs, HS lncRNAs and their DBSs also critically regulate gene expression human-specifically.

This study focuses on HS lncRNAs, HS lncRNA DBSs, and their impacts on human evolution by exploring multiple methods and resources. The first is RNA sequence search based on structure and sequence alignment to identify orthologs of lncRNA genes in genomes (*Nawrocki, 2014*; *Nawrocki*

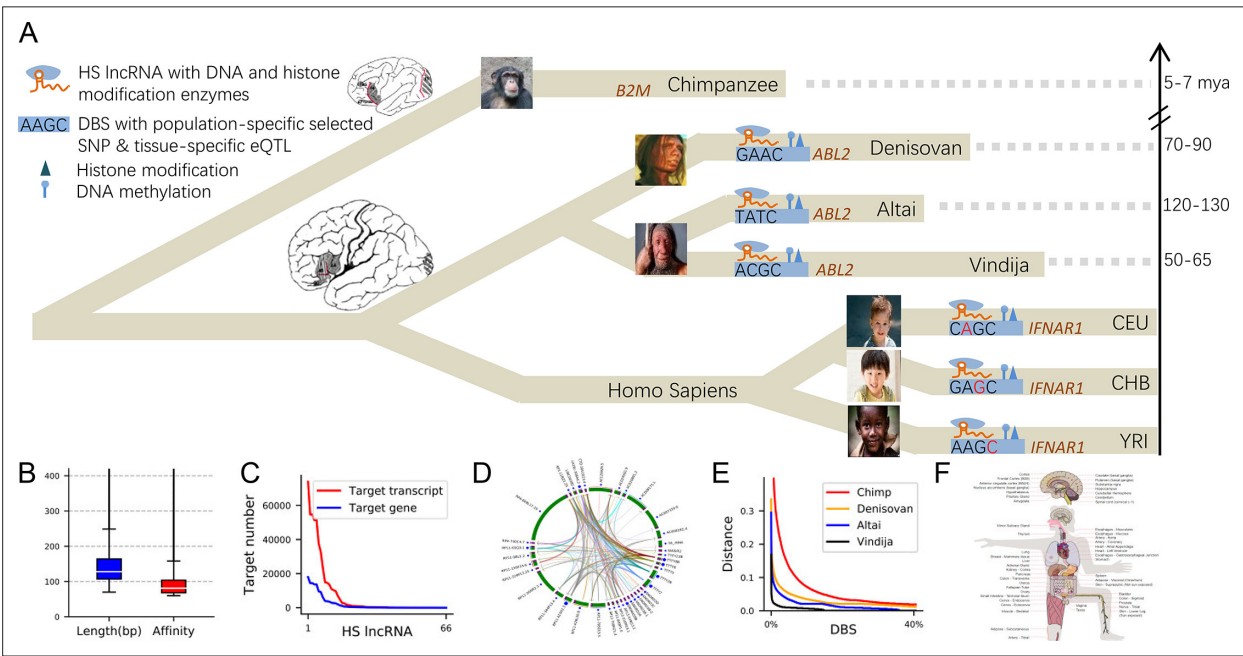

**Figure 1.** Study overview. (**A**) The relationships between chimpanzees, the three archaic humans, and the three modern human populations, with dashed lines indicating the phylogenetic distances from modern humans based on related studies. Based on the left-top icons, the DBS in *B2M* lacks a counterpart in chimpanzees; the DBS in *ABL2* has great differences between archaic and modern humans; the DBS in *IRNAR1* is polymorphic in modern humans (red letters indicate tissue-specific expression quantitative trait loci (eQTLs) or population-specific mutations). (**B**) The mean length and affinity of strong DBSs. (**C**) Numbers of target genes and target transcripts of HS lncRNAs. (**D**) The illustrative figure shows the targeting relationships between HS lncRNAs (*Appendix 2—figure 1*). (**E**) Sequence distances of the top 40% of DBSs from modern humans to chimpanzees and archaic humans. (**F**) The illustrative figure shows the impacts of HS lncRNA–target transcript on gene expression in GTEx tissues (*Figure 2*).

*and Eddy, 2013*). The second is the specific base-pairing rules that RNA and DNA sequences follow to form RNA:DNA triplexes (*Abu Almakarem et al., 2012*); these rules allow computational prediction of lncRNA DBDs and DBSs (*Lin et al., 2019*; *Wen et al., 2022*). The third is gene expression in organs and tissues, especially the Genotype-Tissue Expression (GTEx) project (*GTEx Consortium, 2017*), which provides data for examining and comparing gene expression regulation by HS lncRNAs and HS TFs. Finally, the genomes of modern humans, archaic humans, and multiple apes (especially chimpanzees) allow cross-species genomic and transcriptomic analysis (*The 1000 Genomes Project Consortium, 2012*; *Chimpanzee Sequencing and Analysis Consortium, 2005*; *Meyer et al., 2012*; *Prüfer et al., 2017*; *Prüfer et al., 2014*).

This study identified HS lncRNAs upon the orthologs of the GENCODE-annotated human lncRNA genes in 16 mammalian genomes, predicted their DBDs and DBSs in modern humans (CEU, CHB, and YRI, the three representative populations of Europeans, Asians, and Africans), archaic humans (Altai Neanderthals, Denisovans, and Vindija Neanderthals), and identified the counterparts of HS lncRNA DBSs in chimpanzees (*Figure 1A*). Our DBS prediction method combines a local alignment algorithm with RNA:DNA base-pairing rules to identify RNA:DNA triplexes, thus predicting DBDs and DBSs simultaneously (*Lin et al., 2019*). Based on recent data and methods, we also predicted HS TFs and their DBSs. Analyses based on HS lncRNAs and their DBSs, HS TFs and their DBSs, and substantial genomic and transcriptomic data suggest that HS lncRNAs and their DBSs have distinctly and continuously reshaped gene expression for adaptive human evolution.

## Results
### HS lncRNAs regulate diverse genes and transcripts

How many human lncRNAs are human-specific still lacks a precise estimate. Based on the orthologs of the 13,562 GENCODE-annotated human lncRNA genes (v18) in 16 mammalian genomes we searched using the *Infernal* program (which features in a combined sequence and structure alignment) (*Lin et al., 2019*; *Nawrocki, 2014*; *Nawrocki et al., 2009*), we identified 66 HS lncRNA genes that exist in humans but not in any other species (*Supplementary file 1A*). Using *the LongTarget* program, we predicted DBDs in exons of HS lncRNAs and their DBSs in the 5000 bp promoter regions of the 179,128 Ensembl-annotated transcripts (release 79) (*Lin et al., 2019*). DBS prediction was validated using multiple methods and datasets (*Appendix 1—figures 1–5*). Predicted DBSs overlap well with experimentally identified DNA methylation and histone modification signals in multiple cell lines (https://genome.UCSC.edu), and with experimentally detected DNA-binding sites of NEAT1, MALAT1, and MEG3 (*Mondal et al., 2015*; *West et al., 2014*). Many DBSs also co-localize with annotated cis-regulatory elements (cCREs) in promoter regions (https://genome.ucsc.edu/). We used CRISPR/Cas9 to delete multiple DBDs (100–200 bp) in several cell lines, performed RNA-sequencing (RNA-seq) before and after DBD knockout (KO), and analyzed the resulting differential gene expression. These DBD KO cases demonstrate that the |fold change| of target genes was significantly larger than the |fold change| of non-target genes. Since DBD1's DBSs vastly outnumber any other DBD's DBSs, which suggests that DBD1s regulate more targets than other DBDs and are more reliable, this study analyzed only DBD1s (simply called DBDs) and their DBSs.

To perform quantitative and comparative analysis, we identified HS lncRNAs and their DBSs in archaic humans and chimpanzees (*Figure 1A*), defined the *binding affinity* (simply called *affinity*) of a pair of DBD and DBS as the product of their length and identity (the percentage of paired RNA and DNA nucleotides), and computed DBSs' sequence distances (sequence distances are defined as the distance per base throughout the manuscript) from humans to archaic humans and chimpanzees. LncRNA:DNA-binding analysis at known imprinted genes suggests that affinity better characterizes DBSs than length (*He et al., 2015*). With affinity and distance, DBSs were classified into strong (affinity ≥60) and weak (36 < affinity < 60) ones (as well as old and young ones based on when large sequence changes occurred). 105,141 strong DBSs (mean length >147 bp) were identified in 96,789 transcripts of 23,396 genes, and 152,836 weak DBSs were identified in 127,898 transcripts of 33,185 genes (*Supplementary file 1B, C*). Several HS lncRNAs (especially RP11-423H2.3) have abundant DBSs; only about 1.5% of target genes (0.6% of target transcripts) are human-specific; many targets have DBSs of multiple HS lncRNAs; many HS lncRNAs have multiple DBSs in a target gene; and HS

lncRNAs themselves show complex targeting relationships (*Figure 1B–D*; *Appendix 2—figures 1–6*). These suggest that HS lncRNAs have significantly reshaped gene expression regulation.

## Target genes with strong and weak DBS characterize adaptive evolution and human traits

Genes with significant sequence differences between humans and chimpanzees range from 1.24% to 5% (*Britten, 2002*; *Ebersberger et al., 2002*). However, substantial genes exhibit expression differences due to variations in regulatory sequences. It is therefore interesting to examine whether genes with strong and weak DBSs are enriched in different functions. We performed over-representation analysis (ORA) using the *g:Profiler* program, Gene Ontology (GO) database, and genes sorted by affinity. When choosing a cutoff to distinguish strong from weak DBSs, we found that the GO terms of the top 2000 protein-coding genes contain, but those of the top 1500 protein-coding genes lack, terms such as 'hair follicle development' and 'skin epidermis development' that critically differentiate humans from chimpanzees. We therefore performed ORA using the top 2000 and bottom 2000 protein-coding genes, respectively. In addition to shared GO terms (e.g., 'behavior'), the two gene

**Table 1.** Genes with DBSs that have largest affinity values and mostly changed sequence distances (from modern humans to archaic humans and chimpanzees).

| Target gene | Annotation | Binding affinity | Mostly changed |
|---|---|---|---|
| IFNAR1 | That is, Interferon Alpha and Beta Receptor Subunit 1. | 794 | C, D |
| NFATC1 | A TF that induces gene transcription during immune responses. | 736 | C |
| NFATC1 | A TF that induces gene transcription during immune responses. | 491 | C, A, D |
| ANKLE2 | Diseases associated with ANKLE2 include microcephaly. | 527 | C, D |
| SEMA4D | Regulating phosphatidylinositol 3-kinase signaling, neuron projection development, and phosphate metabolic process. | 495 | C, A, D |
| KIF21B | Essential for neuronal morphology, synapse function, and learning and memory. | 471 | C |
| ALDH3B2 | An aldehyde dehydrogenase for alcohol metabolism. | 444 | C |
| NTSR1 | A brain and gastrointestinal peptide that mediates functions of neurotensin (e.g., hypotension, hyperglycemia, hypothermia, and antinociception). | 402 | C, A, D |
| MC5R | A receptor for melanocyte-stimulating hormone and adrenocorticotropic hormone. | 397 | C |
| THEG | Specifically expressed in the germ cells and involved in spermatogenesis. | 395 | C, D |
| HERC6 | In pathways including class I MHC-mediated antigen processing and presentation, and the innate immune system. | 369 | C, A, D |
| SLC2A11 | Facilitating glucose transporter. | 356 | C |
| NGEF | Playing a role in axon guidance regulating ephrin-induced growth cone collapse and dendritic spine morphogenesis. | 354 | C |
| SHC2 | Involved in the signal transduction pathways of neurotrophin-activated Trk receptors in cortical neurons. | 345 | C, D |
| BAIAP3 | Regulating behavior and food intake by controlling calcium-stimulated exocytosis of neurotransmitters, serotonin, and hormones like Insulin. | 321 | C |
| SLURP1 | A marker of late differentiation of the skin. | 319 | C |
| MLPH | Involved in melanosome transport. | 307 | C |
| TAS1R3 | Responding to the umami taste stimulus and recognizing diverse natural and synthetic sweeteners. | 304 | C, D |
| SLC2A1 | A major glucose transporter in the mammalian blood–brain barrier. | 356 | C |
| CTD-3224I3.3 | An lncRNA is highly expressed in the cerebellum, lung, and testis. | 312 | C, A, D |

'C', 'A', 'D', and 'V' indicate that the DBS has mostly changed sequence distances from modern humans to chimpanzees, Altai Neanderthals, Denisovans, and Vindija Neanderthals, respectively. *NFATC1* is displayed in two rows because the DBSs of SNORA59B and TTTY8/TTTY8B have different affinity values.

sets have many specific ones. GO terms specific to the top 2000 genes include 'renal system development', and GO terms specific to the bottom 2000 genes include 'cellular response to alcohol', 'response to temperature stimulus', and 'female pregnancy' (*Supplementary file 1D*; *Appendix 3—table 1*).

Genes with strongest DBSs (affinity ≥300) include *IFNAR1* and *NFATC1* (important for immune function), *KIF21B* and *NTSR1* (critical for neural development), *SLC2A11* and *SLC2A1* (involved in glucose usage), *BAIAP3* (a brain-specific angiogenesis inhibitor), *TAS1R3* (a receptor for the sweet taste response), and several primate-specific lncRNA genes (e.g., *CTD-3224I3.3* with high expression in the cerebellum, lung, and testis). DBSs in these genes underwent large sequence changes from chimpanzees and archaic humans to modern humans (*Table 1*, *Supplementary file 1E*).

## Genes intensively regulated by HS lncRNAs may have promoted human evolution

The human genome is approximately 99%, 99%, and 98% identical to the genomes of chimpanzees, bonobos, and gorillas (*Chimpanzee Sequencing and Analysis Consortium, 2005*; *Prüfer et al., 2012*). However, the sequence variants that critically differentiate these species remain unclear. To assess whether regulation by HS lncRNAs is critical, we first examined whether DBSs of HS lncRNAs are human-specific or conserved across these species and found that 97.81% of the 105,141 strong DBSs have counterpart sequences in chimpanzees. To further determine whether the remaining 2.2% are human-specific gains, we checked them using the UCSC Multiz Alignments of 100 Vertebrates and found that they are present in the human genome but absent from the chimpanzee genome and all other aligned vertebrate genomes. Since most DBSs have chimpanzee counterparts, yet have evolved considerably from chimpanzees to archaic and modern humans, they share features of HARs. However, of the 312 HARs identified by the Zoonomia Project as important for 3D genome rewiring and neurodevelopment (*Keough et al., 2023*), only eight overlap 26 DBSs of 14 HS lncRNAs, suggesting that DBSs and HARs may contribute differently to human evolution.

HS lncRNAs' DBSs may be generated before, together with, or after HS lncRNAs, and the first situation suggests that HS lncRNAs may reshape gene expression via DBSs. We therefore identified counterparts of HS lncRNAs and their DBSs in Altai Neanderthals, Denisovans, and Vindija Neanderthals (*Meyer et al., 2012*; *Prüfer et al., 2017*; *Prüfer et al., 2014*). While counterparts of both were identified in these archaic humans, sequence distances of HS lncRNAs from humans to archaic humans are smaller than those of DBSs, suggesting that many DBS sequences were generated before HS lncRNAs.

We then computed DBS sequence distances using two methods. The first is from the reconstructed human ancestor (downloaded from the EBI website) to chimpanzees, archaic humans, and modern humans. In this result, many DBS distances from the human ancestor to modern humans are shorter than those to archaic humans. The second is from the human genome to chimpanzees and archaic humans. This set of distances agrees better with the phylogenetic distances between chimpanzees, archaic humans, and modern humans (*Appendix 3—figure 1*).

We postulate that genes with large DBS distances may have contributed more to human evolution than genes with small DBS distances. To test this postulation, we sorted genes by DBS distances from humans to chimpanzees and to Alai Neanderthals (*Supplementary file 1F*), and applied ORA to genes with large and small DBS distances using the *g:Profiler* program and GO database (Benjamini–Hochberg FDR, threshold = 0.05, 50 < terms size < 1000). First, we examined the top 25% and bottom 25% of genes to determine whether they are enriched for different GO terms. The result indicates that the top 25% of genes generate more enriched GO terms and also more human evolution-related GO terms (*Table 2*, *Supplementary file 1G*). Second, we examined genes with the top and bottom DBS distances and also critically differentiate humans from chimpanzees. *Agoglia et al., 2021* fused human and chimpanzee-induced pluripotent stem cells to generate tetraploid hybrid stem cells (hybrid iPS), differentiated these cells into neural organoids, and measured genes with significant allele-specific expression (ASE) in hybrid iPS. We selected the top 50% and bottom 50% of genes, intersected them with the 2891 genes with significant ASE (p-adj <0.01 and |LFC| >0.5), and applied ORA to the four gene sets. The four results indicate that more GO terms, especially more human evolution-related GO terms, were generated by the top 50% of genes (*Table 2*, *Supplementary file 1H*). Using ASE genes with less significance (just p-adj <0.01) yielded similar results (*Appendix 3—table 2*). These results

**Table 2.** GO terms generated by different gene sets with large and small DBS distances from humans to chimpanzees and Altai Neanderthals.

| Top 25% genes (sorted by DBS distance from humans to chimpanzees) in *Supplementary file 1F*, column A | term_id | adj_p | Bottom 25% genes (sorted by DBS distance from humans to chimpanzees) in *Supplementary file 1F*, column A | term_id | adj_p |
|---|---|---|---|---|---|
| Behavior | GO:0007610 | 8.26E−07 | Head development | GO:0060322 | 1.96E−03 |
| Head development | GO:0060322 | 4.87E−05 | Forebrain development | GO:0030900 | 2.26E−03 |
| Brain development | GO:0007420 | 2.69E−04 | Brain development | GO:0007420 | 2.80E−03 |
| Forebrain development | GO:0030900 | 8.21E−03 | Behavior | GO:0007610 | 3.78E−03 |
| Sensory organ development | GO:0007423 | 1.07E−02 | Locomotory behavior | GO:0007626 | 1.75E−02 |
| Learning or memory | GO:0007611 | 1.36E−02 | | | |
| Locomotory behavior | GO:0007626 | 1.63E−02 | | | |
| Sensory system development | GO:0048880 | 1.93E−02 | | | |
| Sensory perception of sound | GO:0007605 | 2.05E−02 | | | |
| Adaptive thermogenesis | GO:1990845 | 3.49E−02 | | | |

| Top 25% genes (sorted by DBS distance from humans to Altai Neanderthals) in *Supplementary file 1F*, column C | term_id | adj_p | Bottom 25% genes (sorted by DBS distance from humans to Altai Neanderthals) in *Supplementary file 1F*, column C | term_id | adj_p |
|---|---|---|---|---|---|
| Behavior | GO:0007610 | 1.28E−09 | Brain development | GO:0007420 | 1.34E−04 |
| Head development | GO:0060322 | 2.16E−05 | Sensory organ development | GO:0007423 | 1.97E−04 |
| Learning or memory | GO:0007611 | 2.66E−05 | Head development | GO:0060322 | 3.98E−04 |
| Brain development | GO:0007420 | 4.15E−05 | Sensory organ morphogenesis | GO:0090596 | 2.03E−03 |
| Locomotory behavior | GO:0007626 | 7.74E−05 | Behavior | GO:0007610 | 9.69E−03 |
| Learning | GO:0007612 | 3.07E−04 | Locomotory behavior | GO:0007626 | 1.66E−02 |
| Forebrain development | GO:0030900 | 3.23E−04 | Sensory system development | GO:0048880 | 4.72E−02 |
| Sensory organ development | GO:0007423 | 3.48E−04 | | | |
| Sensory system development | GO:0048880 | 4.16E−04 | | | |
| Sensory organ morphogenesis | GO:0090596 | 6.43E−03 | | | |
| Associative learning | GO:0008306 | 6.43E−03 | | | |
| Memory | GO:0007613 | 1.18E−02 | | | |
| Social behavior | GO:0035176 | 1.37E−02 | | | |
| Sensory perception of sound | GO:0007605 | 2.91E−02 | | | |

| Intersection of top 50% genes (sorted by DBS distance from humans to chimpanzees) and ASE genes in *Supplementary file 1F*, columns A and F | term_id | adj_p | Intersection of bottom 50% genes (sorted by DBS distance from humans to chimpanzees) and ASE genes in *Supplementary file 1F*, columns A and F | term_id | adj_p |
|---|---|---|---|---|---|
| Cellular pigmentation | GO:0033059 | 3.27E−05 | | | |
| Pigmentation | GO:0043473 | 3.94E−04 | | | |
| Behavior | GO:0007610 | 1.08E−03 | | | |
| Sensory system development | GO:0048880 | 2.61E−03 | | | |
| Learning | GO:0007612 | 3.69E−03 | | | |
| Learning or memory | GO:0007611 | 1.60E−02 | | | |
| Associative learning | GO:0008306 | 1.62E−02 | | | |

*Table 2 continued on next page*

*Table 2 continued*

| Intersection of top 50% genes (sorted by DBS distance from humans to chimpanzees) and ASE genes in *Supplementary file 1F*, columns A and F | term_id | adj_p | Intersection of bottom 50% genes (sorted by DBS distance from humans to chimpanzees) and ASE genes in *Supplementary file 1F*, columns A and F | term_id | adj_p |
|---|---|---|---|---|---|
| Cognition | GO:0050890 | 2.06E−02 | | | |
| Sensory organ development | GO:0007423 | 2.11E−02 | | | |
| Adaptive thermogenesis | GO:1990845 | 3.16E−02 | | | |
| Memory | GO:0007613 | 4.85E−02 | | | |
| Intersection of top 50% genes (sorted by DBS distance from humans to Altai Neanderthals) and ASE genes in *Supplementary file 1F*, columns C and F | term_id | adj_p | Intersection of bottom 50% genes (sorted by DBS distance from humans to Altai Neanderthals) and ASE genes in *Supplementary file 1F*, columns C and F | term_id | adj_p |
| Behavior | GO:0007610 | 3.88E−05 | Pigmentation | GO:0043473 | 7.11E−03 |
| Sensory system development | GO:0048880 | 4.09E−03 | Cellular pigmentation | GO:0033059 | 1.10E−02 |
| Sensory organ development | GO:0007423 | 1.51E−02 | | | |
| Sensory perception of sound | GO:0007605 | 3.95E−02 | | | |
| Learning | GO:0007612 | 4.74E−02 | | | |
| Learning or memory | GO:0007611 | 4.77E−02 | | | |

The presence and absence of human evolution-related GO terms in the ORA results (**Supplementary file 1G, H**). Left: The top genes. Right: The bottom genes. Upper (black): Target genes. Bottom (blue): The intersections of target genes and genes with significant ASE (p-adj <0.01 and |LFC| >0.5). HS lncRNAs' target genes are sorted by DBS distance from humans to chimpanzees and Altai Neanderthals.

support the above postulation. Different GO terms, GO terms with different significance levels, and GO terms generated based on DBS distances to chimpanzees and archaic humans (e.g., 'sensory perception of sound'), provide additional information (e.g., timing) about the influence of reshaped transcriptional regulation on human evolution.

## Regulation by HS lncRNAs shows cross-population differences

To reveal whether DBSs with large human–chimpanzee distances have further evolved in archaic and modern humans, we extracted those that (1) are in the top 20% of genes sorted by human–chimpanzee DBS distance (>0.037) and (2) also have a distance >0.037 to at least one archaic human. We label these DBSs using 'A', 'D', and 'V' (e.g., 'A, D' indicates the distances to both Altai Neanderthals and Denisovan >0.037), calculated SNP number and SNP number per base (for SNPs with minimal allele frequency (MAF) ≥0.1), and computed weighted Fst and Tajima's *D* (**The 1000 Genomes Project Consortium, 2012**). SNP number, weighted Fst, and Tajima's *D* reveal that many DBSs that have undergone significant and continuous sequence change have been selected and are highly polymorphic in specific modern humans (**Supplementary file 1I**). Many genes with these DBSs encode lncRNAs, and those encoding proteins include *SCTR, NCR, IFNAR1, NFATC1, TAS1R3, INS, ST3GAL4*, and *FN3KRP*, which are important for adaptive human evolution (**Table 3**). These suggest that the regulation of genes important for human evolution by HS lncRNAs has undergone continuous evolution.

Genes involved in sugar metabolism are notable. A key feature of modern human diets is high sugar intake, which can cause non-enzymatic oxidation of proteins (i.e., glycation) and deactivate protein functions. Among proteins encoded by genes in this intersection, TAS1R3 recognizes diverse natural and synthetic sweeteners, insulin decreases blood glucose concentration, ST3GAL4 regulates protein glycosylation, and FN3KRP deglycates proteins. DBSs in these genes indicate the continuous evolution of human- and population-specific epigenetic regulation.

To further examine whether SNPs in DBSs are neutral or indicate selection, we used multiple statistical tests with widely adopted parameters, including XP-CLR (**Chen et al., 2010**), iSAFE (**Akbari et al., 2018**), Tajima's *D* (**Tajima, 1989**), Fay–Wu's *H* (**Fay and Wu, 2000**), the fixation index (Fst) (**Weir and Cockerham, 1984**), and linkage disequilibrium (LD) (**Slatkin, 2008**), to detect selection signals in HS lncRNAs and strong DBSs in CEU, CHB, and YRI (**Appendix 4—figures 1–4**, **Appendix 5—figures**

**Table 3.** Genes with DBSs that are most polymorphic and have mostly changed sequence distances from humans to archaic humans and chimpanzees.

| Target gene | Annotation | SNP number | Mostly changed |
|---|---|---|---|
| IFNAR1 | That is, Interferon Alpha and Beta Receptor Subunit 1. | 31 | C, D |
| DECR2 | The related pathways include metabolism and regulation of lipid metabolism. | 17 | C, A, D |
| DOK7 | Essential for neuromuscular synaptogenesis. | 17 | C, D |
| TAS1R3 | Responding to the umami taste stimulus and recognizing diverse natural and synthetic sweeteners. | 17 | C, D |
| NFATC1 | A TF that induces gene transcription during immune responses. | 16 | C, D |
| ST3GAL4 | Involved in protein glycosylation. | 15 | C, D |
| CAMK2B | Calcium/calmodulin-dependent protein kinase important for dendritic spine and synapse formation and maintaining synaptic plasticity. | 13 | C, D |
| HLA-DQB1-AS1 | Highly expressed in EBV-transformed lymphocytes, lung, and spleen. | 13 | C, A, D, V |
| ANKLE2 | Diseases associated with ANKLE2 include microcephaly. | 12 | C, D |
| KRTAP1-3 | The KAP proteins form a matrix of keratin intermediate filaments that contribute to the structure of hair fibers. | 12 | C, D |
| INS, INS-IGF2 | Insulin decreases blood glucose concentration. | 11 | C, A, D |
| SHC2 | Involved in the signal transduction pathways of neurotrophin-activated Trk receptors in cortical neurons. | 11 | C, D |
| FN3KRP | Deglycating proteins to restore their function, important for modern humans adaptive to high glucose intake and functions in all tissues. | 10 | C, D |
| TFB1M | The encoded protein is part of the basal mitochondrial transcription complex and is necessary for mitochondrial gene expression. | 10 | C, A, D |

Some protein-coding genes that have (1) large DBS distances from humans to chimpanzees, (2) large DBS distances to Altai Neanderthals, Denisovans, or Vindija Neanderthals, and (3) dense SNPs. Letters C, A, D, and V indicate that DBS distance from humans to chimpanzees, Altai Neanderthals, Denisovans, and Vindija Neanderthals ≥0.037. Note that different HS lncRNAs' DBSs in a gene may have somewhat different sequences, weighted Fst, and Tajima's *D*.

*1 and 2*, *Supplementary file 1J–L*). Selection signals were detected in many DBSs (more in CEU and CHB than in YRI) but only in several HS lncRNA genes (in CEU and CHB but not in YRI), and the same signals were often detected by multiple tests. These results agree with the findings that fewer selection signals are detected in YRI (*Sabeti et al., 2007*; *Voight et al., 2006*).

## SNPs in DBSs exhibit cis-effects on gene expression

To verify that DBSs and SNPs influence gene expression, we analyzed the GTEx data (*GTEx Consortium, 2017*). We identified expression quantitative trait loci (eQTLs) in DBSs using the widely adopted criteria of MAF ≥0.1 and cis-effect size (ES) ≥0.5. 1727 SNPs, with MAF ≥0.1 in at least one population and |ES| ≥0.5 in at least one tissue, were identified in DBSs in autosomal genes (*Supplementary file 1M, N*). These eQTLs include 372 'conserved' ones (i.e., also in DBSs in the three archaic humans) and 1020 'novel' ones (i.e., only in DBSs in modern humans). A notable eQTL with high derived allele frequencies (DAFs) across all three populations and a positive ES ≥0.5 in 44 of 48 GTEx tissues is rs2246577 in the DBS in *FN3KRP*, whose encoded protein deglycates proteins to restore their function. Many conserved eQTLs are expressed in brain tissues and exhibit high DAFs across all three modern populations. In contrast, many novel eQTLs are tissue-specific and exhibit population-specific DAFs (*Appendix 6—table 1*).

Next, we performed two analyses to examine how eQTLs in DBSs influence gene expression. First, we computed the expression correlation between HS lncRNAs and target transcripts with DBSs having eQTLs in specific tissues. Most (94%) HS lncRNA–target transcript pairs showed significant expression correlation in tissues in which the eQTLs were identified (|Spearman's rho| >0.3 and FDR <0.05). Second, we examined whether eQTLs are more enriched in DBSs than in Ensembl-annotated promoters by computing and comparing the eQTL density in the two classes of regions (promoters were used as the reference because they contain DBSs). The results indicate that eQTLs are more

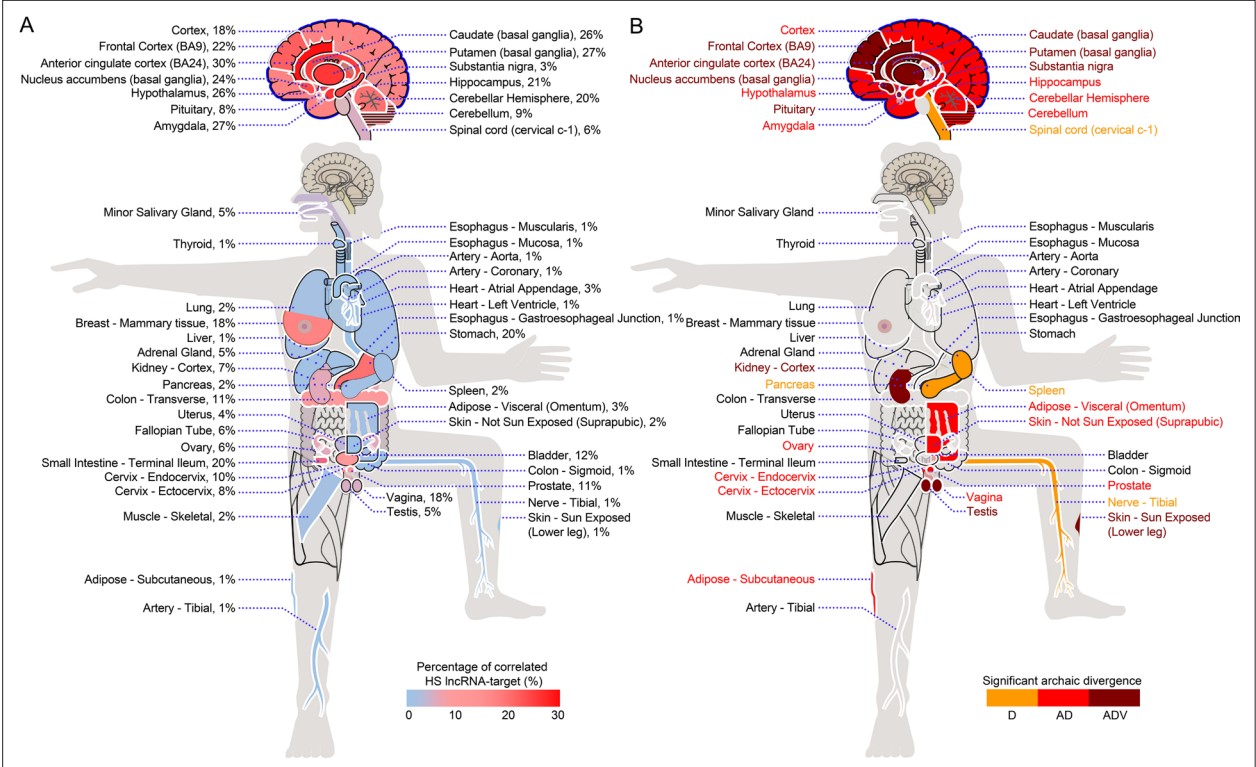

**Figure 2.** The impact of HS lncRNA–DBS interaction on gene expression in GTEx tissues and organs. (**A**) The distribution of the percentage of HS lncRNA–target transcript pairs with correlated expression across GTEx tissues and organs. Higher percentages of correlated pairs are in brain regions than in other tissues and organs. (**B**) The distribution of significantly changed DBSs (in terms of sequence distance) in HS lncRNA–target transcript pairs across GTEx tissues and organs between archaic and modern humans. Orange, red, and dark red indicate significant changes from Denisovans (D), Altai Neanderthals and Denisovans (AD), and all three archaic humans (ADV). DBSs in HS lncRNA–target transcript pairs with correlated expression in seven brain regions (in dark red) have changed significantly and consistently since the Altai Neanderthals, Denisovans, and Vindija Neanderthals (one-sided two-sample Kolmogorov–Smirnov test, significant changes determined by FDR <0.001).

enriched in DBSs than in promoters (one-sided Mann–Whitney test, p = 0.0) (*Appendix 6—figure 1*). Thus, population-specific SNPs and tissue-specific eQTLs at the genome and transcriptome levels support that HS lncRNAs regulate transcription with population- and tissue-specific features.

## HS lncRNAs promote brain evolution from archaic to modern humans

To further verify that SNPs in DBSs have cis-effects on gene expression, we examined the impact of HS lncRNAs and DBSs on gene expression in the GTEx tissues (*GTEx Consortium, 2017*). 40 autosomal HS lncRNAs are expressed in at least one tissue (median TPM >0.1), these HS lncRNAs and their target transcripts form 198,876 pairs in all tissues, and 45% of pairs show a significant expression correlation in specific tissues (Spearman's |rho| >0.3 and FDR <0.05). To assess the likelihood that these correlations could be generated by chance, we randomly sampled 10,000 pairs of lncRNAs and protein-coding transcripts genome-wide and found that only 2.3% of pairs showed significant expression correlation (Spearman's |rho| >0.3 and FDR <0.05). Moreover, a higher percentage (56%) of HS lncRNA–target transcript pairs with significant correlation was detected in at least one brain region, indicating more extensive gene expression regulation in the brain (*Figure 2A*).

To further obtain supporting evidence on the transcriptome level, we analyzed nine experimental datasets (Appendix 7). First, we analyzed two datasets of epigenetic studies: one examining H3K27ac and H3K4me2 profiling in human, macaque, and mouse corticogenesis, and the other examining gene expression and H3K27ac modification in eight brain regions in humans and four other primates (*Reilly et al., 2015*; *Xu et al., 2018*). Compared with the genome-wide background, 84% and 73% of genes in the two datasets have DBSs for HS lncRNAs, indicating significantly higher enrichment in HS lncRNA targets (p = 1.21e−21 and 1.2e−56, two-sided Fisher's exact test). When chimpanzee gene expression was used as the control, 1851 genes showed human-specific transcriptomic differences in

one or multiple brain regions, whereas only 240 genes showed chimpanzee-specific transcriptomic differences. Second, we analyzed two datasets of PsychENCODE studies: one examining spatiotemporally differentially expressed genes and spatiotemporally differentially methylated sites across 16 brain regions, and the other examining the spatiotemporal transcriptomic divergence between human and macaque brain development (*Li et al., 2018*; *Zhu et al., 2018*). In the first dataset, 65 HS lncRNAs are expressed in all 16 brain regions, 109 transcripts have spatiotemporally differentially methylated sites in their promoters, and 56 of the 109 transcripts have DBSs for HS lncRNAs. In the second dataset, 8951 genes show differential expression between the human and macaque brains, and 72% of differentially expressed genes in the human brain have DBSs for HS lncRNAs. Thus, both datasets show significant enrichment for HS lncRNA regulation. Third, three studies identified genes critically regulating cortical expansion (*Florio et al., 2018*; *Johnson et al., 2018*; *Suzuki et al., 2018*). Of the 40 reported protein-coding genes, 29 have DBSs of HS lncRNAs in promoter regions (*Supplementary file 1O*). Thus, these genes are enriched with DBSs of HS lncRNAs compared to the genome-wide background ($p < 0.01$, two-sided Fisher's exact test). Fourth, we analyzed two datasets of brain organoid studies. By establishing and comparing cerebral organoids between humans, chimpanzees, and macaques, *Pollen et al., 2019* identified 261 human-specific gene expression changes. By fusing human and chimpanzee iPS cells and differentiating the hybrid iPS cells into hybrid cortical spheroids, *Agoglia et al., 2021* generated a panel of tetraploid human–chimpanzee hybrid iPS cells and identified thousands of genes with divergent expression between humans and chimpanzees. We found that 261 and 1102 genes in the two datasets are enriched for DBSs of HS lncRNAs compared to the genome-wide background ($p = 1.2e-16$ and $3.4e-74$).

Next, we examined the evolution of the impact of HS lncRNAs and their DBSs on gene expression using the GTEx data. For each DBS in an HS lncRNA–target transcript pair that shows correlated expression in a GTEx tissue, we computed the sequence distances of this DBS from the modern humans to the three archaic humans. Then, we compared the distribution of DBS sequence distances in each tissue with that in all tissues as the background (one-sided two-sample Kolmogorov–Smirnov test). DBSs in HS lncRNA–target transcript pairs with correlated expression in brain regions have significantly changed sequence distances since the Altai Neanderthals and Denisovans (*Figure 2B*), supporting that gene expression regulation by HS lncRNAs in the brain has undergone more significant evolution (*Leon et al., 2021*).

To substantiate the above conclusion, we further examined whether HS lncRNAs have contributed more to human evolution than HS TFs. Based on the 'hg38-panTro6' gene sets reported by *Kirilenko et al., 2023* and the human TF lists reported by previous studies (*Bahrami et al., 2015*; *Lambert et al., 2018*), five HS TFs were identified. We predicted their DBSs in the 5000 bp promoter regions of the same 179128 Ensembl-annotated transcripts (release 79) using the *FIMO* and *CellOracle* programs (*Grant et al., 2011*; *Kamimoto et al., 2023*), identified counterparts of these HS TF DBSs in archaic humans and chimpanzees, calculated DBS sequence distances from modern humans to archaic humans and chimpanzees (*Supplementary file 1P*), computed the Pearson correlation of HS TFs and their target transcripts across GTEx tissues, and calculated sequence distances of DBSs in HS TF–target transcript pairs across GTEx tissues from modern humans to archaic humans. Highly correlated HS TF–target transcript pairs are distributed across many GTEx tissues rather than being confined to the brain, and significantly changed HS TF DBSs do not occur densely in the brain (*Appendix 8—figures 1 and 2*). These results further support that HS lncRNAs have promoted human brain evolution more than HS TFs.

## HS lncRNAs mediate human-specific correlated gene expression in the brain

Finally, we further examined whether the gene expression pattern in the brain is human-specific. Based on the GTEx data from the human frontal cortex (BA9) and anterior cingulate cortex (BA24) ($n = 101$ and $n = 83$, respectively) (*GTEx Consortium, 2017*), and the gene expression data for the same brain regions in macaques ($n = 22$ and $n = 25$, respectively) (*Zhu et al., 2018*), we identified transcriptional regulation modules using the *eGRAM* program. In the two human brain regions, HS lncRNAs' target genes form distinct modules, are characterized by highly correlated gene expression (Pearson's $r > 0.8$), and are enriched for KEGG pathways related to neurodevelopment (hypergeometric distribution test, FDR <0.05). In the same macaque brain regions, the orthologs of these human genes lack

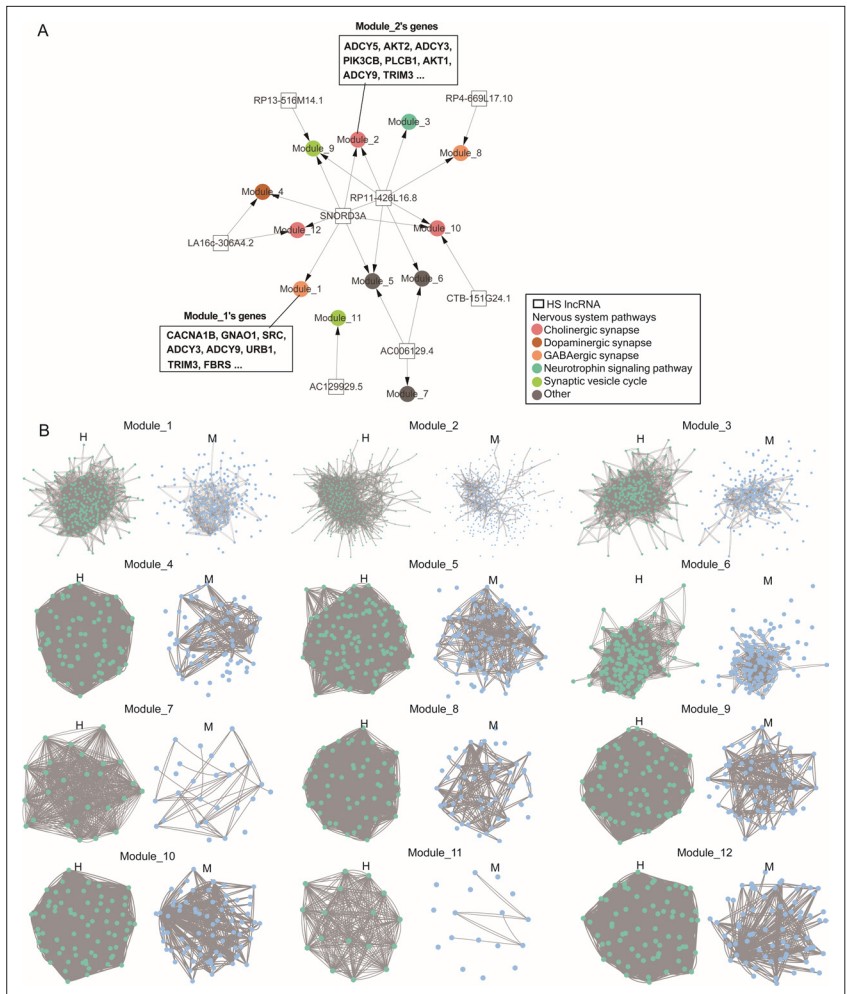

**Figure 3.** Human-specifically reshaped gene expression by HS lncRNAs in the frontal cortex (BA9). (**A**) Genes expressed in the human frontal cortex are enriched for HS lncRNAs' target genes and neurodevelopment-related pathways. Squares, dots, and colors indicate HS lncRNAs, gene modules (Module_1 and Module_2 are illustrated), and enriched KEGG pathways, respectively. (**B**) Comparison of modules and genes in humans (indicated by H) and macaques (indicated by M). In each pair of modules, green and blue dots denote human genes and their orthologs, and lines between dots indicate correlated expression. Many orthologous genes in macaques (displayed at the corresponding positions) are not in the modules, and correlated expression is more prominent in humans than in macaques.

these features (*Figure 3*; *Appendix 9—figure 1*, *Supplementary file 1Q*). The differences in gene modules suggest that the gene expression pattern in the human brain is highly human-specific and, to a great extent, can be attributed to the regulation by HS lncRNAs.

## Discussion

The limited genomic but substantial phenotypic and behavioral differences between modern humans, archaic humans, and apes make 'what genomic differences critically determine modern humans' a profound question. This question has been addressed by many studies using various methods. Especially, studies examining protein-coding genes and HARs reported that HS protein-coding genes promote rapid neocortex expansion (*Fiddes et al., 2018*; *Florio et al., 2018*; *Pinson et al., 2022*) and that HARs are significantly enriched in genomic regions important for human-specific 3D genome organization (*Keough et al., 2023*). Many studies also examined lncRNAs in humans and mice. However, reports on HS lncRNAs and their DBSs have been rare. Similar to gene expression reshaping or rewiring by lineage-specific TFs and their DBSs (*Johnson, 2017*), gene expression can also be

substantially rewired by lineage-specific lncRNAs and their DBSs. This study examined the postulation that HS lncRNAs and their DBSs have significantly reshaped gene expression during human evolution.

Due to the nature of large-scale multi-omics data analysis, our main results are unlikely to be generated by chance, although differences in parameters, thresholds, and significance levels can make the results somewhat different. We used multiple methods, including DBD KO following differential gene expression analysis, to validate DBS prediction due to its importance. When predicting and analyzing DBSs, we examined DBS in all transcripts; thus, transcripts (including those in the brain) whose DBSs have had significant human-specific evolution can be identified. When running programs including *Infernal*, *LongTarget*, and g:*Profiler* and performing statistical tests that detect selection signals, we used the default parameters that suit most situations and the widely used significance levels. When comparing target genes of HS lncRNAs and their counterparts and orthologs in archaic humans and chimpanzees, we used multiple ORA-based methods and explored the ASE genes in tetraploid hybrid human–chimpanzee stem cells (*Agoglia et al., 2021*). Since weak DBSs reflect recent evolution but may be less reliable, we classified DBSs into strong/weak and old/young classes, from which distinct features were identified. Since not all detected signals reliably indicate positive selection, we analyzed both SNPs and selection signals.

A few notes on several findings. First, in line with the finding that Neanderthal-inherited sequences have measurable effects on gene expression in modern humans, but these effects are least detectable in the brain (*McCoy et al., 2017*), we find that HS lncRNAs and their DBSs have influenced gene expression in the brain more significantly than in other tissues. Second, the evolution of gene regulation by HS lncRNAs may be rapid, and notable examples include those that enable humans to adapt to high-sugar intake. Third, regulated genes are enriched for different functions at different periods of human evolution, as evidenced by genes with young/old DBSs and with DBSs that contain 'old' and 'new' SNPs. For example, DBSs containing 'old' and 'new' SNPs are in genes regulating neural development and glucose metabolism, respectively. Fourth, newly emerged regulations (indicated by young weak DBSs) may have promoted recent human evolution.

Our findings also raise new questions. First, since many mouse lncRNAs are rodent-specific (*Yue et al., 2014*), two questions are how these lncRNAs specifically rewire gene expression in mice, and to what extent human- and mouse-specific lncRNAs cause differences in cross-species transcriptional regulation (*Breschi et al., 2017*; *Hodge et al., 2019*; *Zhu et al., 2018*). Second, whether mice and other mammals have species-specific lncRNAs like RP11-423H2.3 in humans that regulate the expression of substantial genes. Third, whether lineage-specific lncRNAs would make many evolutionary novelties preordained.

# Materials and methods
## Data resources

The sequences of human (GRCh38/hg38, GRCh37/hg19) and chimpanzee genomes (panTro5) were obtained from the UCSC Genome Browser (http://genome.UCSC.edu). Three high-quality archaic human genomes were obtained from the Max Planck Institute for Evolutionary Anthropology (https://www.eva.mpg.de/genetics/index.html), which include an Altai Neanderthal that lived approximately 122 thousand years ago (kya) (*Prüfer et al., 2014*), a Denisovan (an Asian relative of Neanderthals) that lived approximately 72 kya (*Meyer et al., 2012*), and a Vindija Neanderthal that lived approximately 52 kya (*Prüfer et al., 2017*). The ancestral sequences for the human genome (GRCh37) (which were generated upon the EPO alignments of six primate species human, chimpanzee, gorilla, orangutan, macaque, and marmoset) were downloaded from the EBI website (here). The SNP data of modern humans were obtained from the 1000 Genomes Project (phase I) (*The 1000 Genomes Project Consortium, 2012*). The three modern human populations are CEU (Utah residents with Northern and Western European ancestry), CHB (Han Chinese in Beijing, China), and YRI (Yoruba in Ibadan, Nigeria), which contain 100, 99, and 97 individuals, respectively. The genetic map files of modern humans were downloaded from the 1000 Genomes Project website (here). The 1000 Genomes phased genotype data in VCF format were downloaded from the International Genome Sample Resource website (http://www.internationalgenome.org/). Annotated human genes and transcripts were obtained from the Ensembl website (http://www.Ensembl.org). FASTA-format nucleotide sequences were converted from VCFs using the *vcf-consensus* program in the *VCFtools* package when necessary (*Danecek et al.,*

*2011*). Data in the GeneCards Human Gene Database (https://www.genecards.org) were used to annotate genes and transcripts.

RNA-sequencing (RNA-seq) data (transcripts per million [TPM]) and eQTL data of human tissues were obtained from the Genotype-Tissue Expression project (GTEx, v8) website (https://gtexportal.org/) (*GTEx Consortium, 2017*). Histone modification and DNA methylation signals in multiple cell lines and ENCODE Candidate Cis-Regulatory Elements (cCRE) were obtained from the UCSC Genome Browser (http://genome.UCSC.edu). The genome-wide DBSs of lncRNA NEAT1, MALAT1, and MEG3 were obtained from experimental studies (*Mondal et al., 2015*; *West et al., 2014*). The predicted and experimentally detected DBSs in target genes of MALAT1, NEAT1, and MEG3 are given in the supplementary file (*Supplementary file 1R*).

## Identifying HS lncRNA genes

We used the *Infernal* program (*Nawrocki et al., 2009*), which searches orthologous RNA sequences upon sequence and structure alignment, to identify orthologous exons in 16 mammalian genomes for each exon in each of the 13,562 GENCODE (v18)-annotated human lncRNA genes (*Lin et al., 2019*). The 16 mammals were chimpanzee (CSAC 2.1.4/panTro4), macaque (BGI CR_1.0/rheMac3), marmoset (WUGSC 3.2/calJac3), tarsier (Broad/tarSyr1), mouse lemur (Broad/micMur1), tree shrew (Broad/tupBel1), mouse (GRCm38/mm10), rat (Baylor3.4/rn4, RGSC6.0/rn6), guinea pig (Broad/cavPor3), rabbit (Broad/oryCun2), dog (Broad CanFam3.1/canFam3), cow (Baylor Btau_4.6.1/bosTau7), elephant (Broad/loxAfr3), hedgehog (EriEur2.0/eriEur2), opossum (Broad/monDom5), and platypus (WUGSC 5.0.1/ornAna1) (http://genome.UCSC.edu). If the number of orthologous exons of a human lncRNA gene in a genome exceeded half the exon number of the human lncRNA gene, these orthologous exons were assumed to form an orthologous lncRNA gene. If a human lncRNA gene had no orthologous gene in all of the 16 mammals, it was assumed to be human-specific.

## Identifying DBSs of HS lncRNAs

LncRNAs bind to DNA sequences by forming RNA:DNA triplexes. Each triplex comprises triplex-forming oligonucleotides (TFO) in the lncRNA and a triplex-targeting site (TTS) in the DNA sequence. We used the *LongTarget* program to predict HS lncRNAs' DBDs and DBSs with the default parameters (Ruleset = all, *TT penalty* = –1000, *CC penalty* = 0, Offset = 15, Identity ≥60, Nt ≥50) (*Lin et al., 2019*; *Wen et al., 2022*). The *LongTarget* program simultaneously predicts DBDs and DBSs, where a DBD comprises a set of densely overlapping TFOs and a DBS comprises a set of densely overlapping TTSs. For each HS lncRNA, we predicted its DBSs in the 5000 bp promoter regions (–3500 bp upstream and +1500 bp downstream the transcription start site) of the 179,128 Ensembl-annotated transcripts (release 79). For each DBS, its binding affinity is the product of DBS length and the averaged *Identity* score of all TTSs (the *Identity* score is the percentage of paired nucleotides). Strong and weak DBSs were classified based on binding affinity ≥60 and <60. A transcript whose promoter region contains a strong DBS of an HS lncRNA was assumed to be a target transcript of the HS lncRNA, and the gene containing this transcript was assumed to be a target gene of the HS lncRNA. As the 1000 Genomes Project (phase I) data and the archaic human genomes are based on GRCh37/hg19, the DBS coordinates were converted from GRCh38/hg38 to GRCh37/hg19 using the *liftover* program from the UCSC Genome Browser (*Kuhn et al., 2013*).

## Experimentally validating DBS prediction

A 157 bp sequence (chr17:80252565–80252721, hg19) containing the DBD of RP13-516M14.1, a 202 bp sequence (chr1:113392603–113392804, hg19) containing the DBD of RP11-426L16.8, and a 198 bp sequence (chr17:19460524–19460721, hg19) containing the DBD of SNORA59B, were knocked out in the HeLa cell line, RKO cell line, and SK-MES-1 cell line, respectively. Two sequences (chr1:156643524–156643684, chr10:52445649–52445740, hg38) containing the DBD of two wrongly transcribed noncoding sequences were knocked out in the HCT-116 and A549 cell lines, respectively. The seven knockouts were performed by UBIGENE, Guangzhou, China (http://www.ubigene.com) using CRISPR-U, a revised version of CRISPR/Cas9 technology. Before and after the seven DBD knock-outs, RNA sequencing (RNA-seq) was performed by Novogene, Beijing, China (https://cn.novogene.com) and HaploX, Shenzhen, China (https://www.haplox.cn/). The reads were aligned to the human GRCh38 genome using the *Hiasat2* program (*Kim et al., 2019*), and the resulting SAM files were

converted to BAM files using *Samtools* (*Li et al., 2009*). The *Stringtie* program was used to quantify gene expression levels (*Pertea et al., 2015*). Fold change of gene expression was computed using the *edgeR* package (*Robinson et al., 2010*), and significant up- and downregulation of target genes after DBD knockout was determined upon |log$_2$(fold change)| >1 with FDR <0.1.

Genome-wide DBSs of NEAT1, MALAT1, and MEG3 were experimentally detected (*Mondal et al., 2015*; *West et al., 2014*). We also used these data to validate DBS prediction by predicting DBSs of the three lncRNAs and checking the overlap between predicted and experimentally detected DBSs (*Supplementary file 1R*).

## Mapping DBSs of HS lncRNAs in the chimpanzee and archaic human genomes

We used the *liftover* program from the UCSC Genome Browser to map DBS loci from the human genome (hg38) to the chimpanzee genome (Pan_tro 3.0, panTro5). The mapping results were verified by inspecting the human–chimpanzee pairwise alignment in the UCSC Genome Browser. This initial screening identified 2248 DBSs (residing in 429 genes) that could not be mapped to the chimpanzee genome. To definitively determine whether these unmapped DBSs represent human-specific gains or chimpanzee-specific losses, we analyzed their sequences using the UCSC Multiz Alignments of 100 Vertebrates. This comparative genomics analysis confirmed that all 2248 DBSs are present in the human genome but are absent from the chimpanzee genome and all other aligned vertebrate genomes. Therefore, we classified these DBSs as human-specific gains.

We used *vcf-consensus* in the *VCFtools* package to extract the DBSs of HS lncRNAs from the VCF files of Altai Neanderthals, Denisovans, and Vindija Neanderthals. The variant with the highest quality score was selected whenever multiple variant calls were observed at a given locus. The obtained DBS sequences in chimpanzees and three archaic humans are called counterparts of DBSs in these genomes.

## Estimating sequence distances of DBSs between different genomes

We first aligned DBS sequences in the genomes of humans, chimpanzees, Altai Neanderthals, Denisovans, and Vindija Neanderthals using the *MAFFT7* program to measure sequence distances from modern humans to chimpanzees and archaic humans (*Katoh and Standley, 2013*). We then computed sequence distances using the *dnadist* program with the Kimura 2-parameter model in the *PHYLIP* (3.6) package (http://evolution.genetics.washington.edu/phylip.html) and the *Tamura-Nei model* in the *MEGA7* package (*Kumar et al., 2016*). The two methods generated equivalent results. The largest distance between DBSs in humans and their chimpanzee counterparts is 5.383. Since 2248 DBSs in 429 human genes lack counterparts in chimpanzees, we assumed that these DBSs have a sequence distance of 10.0 between humans and chimpanzees.

We determined human ancestral sequences of DBSs using the human ancestor sequences from the EBI website, which were generated from the EPO alignments of six primate species. We used the above-mentioned methods to calculate the sequence distances DBSs from the human ancestor to chimpanzees, archaic humans, and modern humans. We found that when the human–chimpanzee ancestral sequence has the ancestral sequence (which means the inference of ancestral allele is of high confidence), DBS distances from the human ancestor to modern humans are larger than to archaic humans, but this situation accounts for only about 63.8%. For many DBSs, the distances from the human ancestor to modern humans are smaller than to archaic humans (especially Neanderthals and Denisovans) and even to chimpanzees. This defect may be caused by the absence of archaic humans in building the human ancestral sequence.

## Detecting positive selection signals in HS lncRNA genes and DBSs

We used multiple tests to detect positive selection signals in HS lncRNA genes. First, we used the XP-CLR test (parameters = -w1 0.001 300 100 -p0 0.95, window size = 0.1 cM, grid size = 100 bp) to perform six pairwise genome-wide scans (i.e., CEU–CHB, CEU–YRI, CHB–CEU, CHB–YRI, YRI–CEU, and YRI–CHB) (*Chen et al., 2010*). The upper 1% of scores across the entire genome in each pairwise scan was 34.6 in the CEU–YRI scan, 16.8 in the CEU–CHB scan, 45.0 in the CHB–YRI scan, 26.9 in the CHB–CEU scan, 14.1 in the YRI–CEU scan, and 14.1 in the YRI–CHB scan. These scores were used as the thresholds of positive selection signals in these populations. Second, we used the *iSAFE* program

to scan each genomic region containing an HS lncRNA gene and its 500 kb upstream and downstream sequences (*Akbari et al., 2018*). Strongly selected loci were detected only in CEU and CHB. Third, we used the *VCFtools* program to calculate Tajima's *D* values for each HS lncRNA gene in CEU, CHB, and YRI (*Danecek et al., 2011*). The calculation was performed using a 1500-bp non-overlapping sliding window because the lengths of these genes exceed 1500 bp. To generate a background reference for assessing significant increases or decreases in Tajima's *D* for HS lncRNA genes in a population, we calculated Tajima's *D* across the whole genome using a sliding window of 1500 bp. As the values of Tajima's *D* were compared with the background reference, significant *D* < 0 and *D* > 0 indicate positive (or directional) selection and balancing selection, respectively, rather than population demography dynamics (*Tajima, 1989*). Fourth, we used the integrated Fst to detect positive selection signals in HS lncRNA genes. The Fst of each HS lncRNA gene was computed for three comparisons, that is, CEU–YRI, CHB–YRI, and CHB–CEU. Extreme Fst values of SNPs were detected in HS lncRNA genes in the comparisons of CEU–YRI and CHB–YRI. Since allele frequencies for different loci vary across a genome and genetic drift may have different effects at different loci, we used a sliding window of 1500 to compare Fst values of HS lncRNA genes with the genome-wide background. Extreme Fst values indicate positive selection. Finally, we applied LD analysis to each HS lncRNA gene. We computed the pairwise LD ($r^2$) in CEU, CHB, and YRI for common SNPs (with MAF ≥0.05 in at least one population) in HS lncRNA genes and DBSs using the *PLINK* program (*Purcell et al., 2007*). Significantly increased LD was detected in SNPs in HS lncRNA genes in CEU and CHB. The LD patterns were portrayed using the *Haploview* program (*Barrett et al., 2005*).

Next, we used the above tests to detect positive selection signals in DBSs. First, the 100 bp grid size of the XP-CLR test also allowed the detection of selection signals in DBSs. Second, we performed Tajima's *D* and Fay–Wu's *H* tests (*Fay and Wu, 2000*; *Tajima, 1989*). We calculated Tajima's *D* values for each DBS in CEU, CHB, and YRI using the *VCFtools* program (*Danecek et al., 2011*), with a sliding window of 147 bp (the mean length of strong DBSs). To generate a background reference for judging the significant increase or decrease of Tajima's *D* in a population, we calculated Tajima's *D* values across the whole genome using the same sliding window. When Tajima's *D* values were compared with the background reference, significant *D* < 0 and *D* > 0 indicate positive (or directional) selection and balancing selection, respectively. Fay–Wu's *H* values were calculated similarly using the *VariScan* program (*Vilella et al., 2005*). Calculating Fay–Wu's *H* demands the ancestral sequences as the outgroup. We extracted the ancestral sequences of DBSs from the human ancestral sequence, which was generated upon the EPO alignments of six primate species (human, chimpanzee, gorilla, orangutan, macaque, and marmoset) (here). Third, we computed the Fst to measure the frequency differences of alleles in DBSs between populations. For the CEU–YRI, CHB–YRI, and CHB–CEU pairwise comparisons, we used the revised *VCFtools* program to compute the weighted Fst for all SNPs in each DBS (*Weir and Cockerham, 1984*). Fourth, we integrated the weighted Fst values in the three populations into an 'integrated Fst' which indicated whether the DBS locus was under selection in a certain population (*Nielsen and Slatkin, 2013*). We used sliding windows of 147 bp for comparing Fst values of DBSs with the genome-wide background. We empirically defined the upper 10% of integrated Fst scores across the entire genome as statistically significant. To determine positive selection more reliably, we used Tajima's *D* and the integrated Fst to jointly determine if a DBS was under positive selection in a population. The thresholds that determined the upper 10% of Tajima's *D* values across the entire genome in CEU, CHB, and YRI were −0.97, −0.96, and −0.97, respectively, and the threshold that determined the upper 10% of integrated Fst values across the entire genome in the three populations was 0.22. For example, a DBS was assumed to be under positive selection in CEU if (1) the DBS had a Tajima's *D* <−0.97 in CEU and Tajima's *D* >0.0 in the two other populations and (2) the DBS had an integrated Fst >0.22. Analogously, a DBS was assumed to be under positive selection in both CEU and CHB if the DBS had a Tajima's *D* <−0.97 in CEU, <−0.96 in CHB, and >0.0 in YRI, and had an integrated Fst >0.22.

## Functional enrichment analysis of genes

We used the *g:Profiler* program (with the parameter settings: Organism = *Homo sapiens*, Ordered query = No, Significance threshold = Benjamini–Hochberg FDR, User threshold = 0.05, 50 < terms size < 1000) and the GO database to perform ORA (*Raudvere et al., 2019*). This analysis determines which pre-defined gene sets (GO terms) are more prevalent (over-represented) in a list of 'interesting'

genes than would be expected by chance. The lists of genes included genes with strong and weak DBSs, genes with large and small DBS distances from humans to chimpanzees, and genes with large and small DBS distances from humans to archaic humans (*Table 2*). Strong DBSs have top affinity, and genes with weak DBSs not only have DBS affinity ≤40 but also have DBSs of ≥5 HS lncRNAs to help ensure that these genes are likely HS lncRNAs' targets.

## Analyzing SNP frequencies in human populations

The frequencies of common SNPs (MAF ≥0.05) in DBSs across the three modern human populations were computed using the *VCFtools* package (*Danecek et al., 2011*). The ancestral/derived states of SNPs were inferred from the human ancestor sequences and were used to determine DAFs.

## Analyzing the cis-effect of SNPs in DBSs on target gene expression

SNPs with MAF >0.1 in DBSs in any of the three modern human populations and absolute values of cis-effect size >0.5 (FDR <0.05) in any of the GTEx tissues were examined for an influence on the expression of the target genes (*GTEx Consortium, 2017*). SNPs that are eQTLs in the GTEx tissues and have biased DAFs in the three modern human populations were examined to estimate whether the eQTL is population-specific.

## Examining the tissue-specific impact of HS lncRNA-regulated gene expression

First, we examined the expression of HS lncRNA genes across the GTEx tissues. HS lncRNA genes with a median TPM value >0.1 in a tissue were considered robustly expressed in that tissue. Upon this criterion, 40 HS lncRNA genes were expressed in at least one tissue and were used to examine the impact of HS lncRNA regulation on gene and transcript expression (other HS lncRNAs may function in the cytoplasm). Since an HS lncRNA gene may have multiple transcripts, we selected the transcript containing the predicted DBD and with the highest TPM as the representative transcript of the HS lncRNA. We calculated the pairwise Spearman's correlation coefficient between the expression of an HS lncRNA (the representative transcript) and the expression of each of its target transcripts using the *scipy.stats.spearmanr* program in the *scipy* package. The expression of an HS lncRNA and a target transcript was considered to be significantly correlated if the |Spearman's rho| >0.3, with Benjamini–Hochberg FDR <0.05. We examined the percentage distribution of correlated HS lncRNA–target transcript pairs across GTEx tissues and organs (*Figure 2A*).

We examined all GTEx tissues to determine which tissues may have exhibited significant changes in HS lncRNA-regulated gene expression from archaic to modern humans. GTEx data include both gene expression matrices and transcript expression matrices; we used the latter to examine changes in HS lncRNA–target transcript pairs from modern humans to archaic humans. If a pair of HS lncRNA and target transcript is robustly expressed in a tissue and their expression shows a significant correlation (|Spearman's rho| >0.3, with Benjamini–Hochberg FDR <0.05) in the tissue, we computed the sequence distance of the HS lncRNA's DBS in the transcript from the three archaic humans to modern humans. We compared the sequence distances of all DBSs in each tissue with the sequence distances of all DBSs in all GTEx tissues (as the background). A one-sided two-sample Kolmogorov–Smirnov test was used to examine whether the sequence distances of all DBSs in a specific tissue deviate from the background distribution (which reflects the 'neutral evolution' of gene expression). For each tissue, if the Benjamini–Hochberg FDR was <0.001, the tissue was considered to have significantly altered gene expression regulated by the HS lncRNA. We used different colors to mark tissues with significantly changed gene expression regulation since Altai Neanderthals, Denisovans, and Vindija Neanderthals, and used 'D', 'A.D.', and 'ADV' to indicate changes since Denisovans, since Altai Neanderthals and Denisovans, and since Altai Neanderthals, Denisovans, and Vindija Neanderthals, respectively (*Figure 2B*).

## Examining enrichment of favored and hitchhiking mutations in DBSs

Using the deep learning network *DeepFavored*, which integrates multiple statistical tests for identifying favored mutations (*Tang et al., 2022*), we identified 13,339 favored mutations and 244,098 hitchhiking mutations in 17 human populations (*Tang et al., 2023*). In this study, we classified DBSs in two ways: into strong ones (affinity >60) and weak ones (36 < affinity < 60), and into old ones

(Human–Chimp distance >0.034 and Human–Altai Neanderthals distance = 0), young ones (Human–Altai Neanderthals distance >0.034 or Human–Denisovan distance >0.034), and others. We then examined the number of favored and hitchhiking mutations in each class of DBSs. The weak young DBSs have the largest proportion of favored and hitchhiking mutations.

## The analysis of HS TFs and their DBSs

*Kirilenko et al., 2023* recently identified orthologous genes in hundreds of placental mammals and birds and organized genes into pairwise datasets using humans and mice as the references (e.g., 'hg38-panTro6', 'hg38-mm10', and 'mm10-hg38'). Based on the 'many2zero' and 'one2zero' gene lists (which contain 0 and 147 genes, respectively) in the hg38-panTro6 dataset, which indicate multiple human genes and a single human gene that have no orthologs in chimpanzees, we identified HS protein-coding genes. Further, based on three human TF lists reported by two studies and used in the *SCENIC* package (*Bahrami et al., 2015*; *Lambert et al., 2018*), we identified HS TFs. Using the JASPAR database and the CellOracle program (with default parameters), we predicted DBSs for HS TFs. Then, we repeated the steps of our HS lncRNA analyses (Appendix 8; *Supplementary file 1P*), including computing sequence distances of DBSs and examining the impact of HS TF–target transcript pairs on gene expression in GTEx tissues and organs.

## Identifying and analyzing transcriptional regulatory modules

Most clustering algorithms classify genes into disjoint modules based on expression correlation without considering regulatory relationships (*Saelens et al., 2018*). The *GRAM* program identifies gene regulatory modules based on correlation and TF–TFBS binding (*Bar-Joseph et al., 2003*). LncRNAs transcriptionally regulate genes based on lncRNA–DBS binding and correlate gene expression. We developed the *eGRAM* program to identify gene modules based on correlated expression, TF–TFBS interaction, and lncRNA–DBS interaction. In this study, we used *eGRAM* to identify gene modules in the same regions of the human and macaque brains and enriched KEGG pathways using reported RNA-seq datasets ($n$ = 101 and 83 for frontal cortex and $n$ = 22 and 25 for anterior cingulate cortex) (*GTEx Consortium, 2017*; *Zhu et al., 2018*). The default parameters, DBS-binding affinity = 60, Pearson correlation = 0.5, module size = 50, and FDR = 0.01 (hypergeometric distribution test), were used. The key steps of the program are as follows. (a1) Identify each lncRNA's correlated lncRNAs, which may form a set of co-regulators. (a2, optional) Identify each TF's correlated TFs, which may form a set of co-regulators. (b1) Compute the correlation between each lncRNA and all genes. (b2, optional) Compute the correlation between each TF and all genes. (c1) Identify each lncRNA's target genes. (c2, optional) Identify each TF's target genes. (d1) Identify each lncRNA set's target module upon correlation and targeting relationships. (d2, optional) Identify each TF set's target module upon correlation and targeting relationships. (e) Check whether TFs' modules contain lncRNAs' targets and whether lncRNAs' modules contain TFs' targets, which reveal genes co-regulated by TFs and lncRNAs and genes independently regulated by TFs and ncRNAs. (f) Performs pathway enrichment analysis for all modules.

## Acknowledgements

This work was supported by the National Natural Science Foundation of China (31771456 to HZ) and the China Postdoctoral Science Foundation (2020M682788 to JL).

## Additional information

### Funding

| Funder | Grant reference number | Author |
|---|---|---|
| National Natural Science Foundation of China | 31771456 | Hao Zhu |
| China Postdoctoral Science Foundation | 2020M682788 | Jie Lin |

| Funder | Grant reference number | Author |
|---|---|---|

The funders had no role in study design, data collection, and interpretation, or the decision to submit the work for publication.

## Author contributions

Jie Lin, Software, Formal analysis, Funding acquisition, Investigation, Methodology; Yujian Wen, Software, Formal analysis, Investigation, Methodology; Ji Tang, Formal analysis, Investigation, Methodology; Xuecong Zhang, Huanlin Zhang, Formal analysis, Investigation; Hao Zhu, Conceptualization, Formal analysis, Supervision, Funding acquisition, Writing – original draft, Project administration, Writing – review and editing

## Author ORCIDs

Jie Lin https://orcid.org/0000-0001-9158-9866
Hao Zhu https://orcid.org/0000-0001-7384-3840

Joint Public Review: https://doi.org/10.7554/eLife.89001.6.sa1
Author response https://doi.org/10.7554/eLife.89001.6.sa2

# Additional files

## Supplementary files

MDAR checklist

Supplementary file 1. All supplementary results from this study (including 18 tables).

## Data availability

Data is provided in the *Supplementary file 1* and at the following locations: GENCODE human lncRNAs are freely available at the website (https://www.gencodegenes.org/human/). The Long-Target program and human lncRNAs' orthologs in 16 mammals are freely available at the Long-Target website (http://www.gaemons.net/), which consists of the program, the lncRNA database LongMan, the full sequences of many mammmalain genomes, and a cluster of servers (also available at Zenodo). Fasim-LongTarget is a fast, simple, and standalone version of LongTarget and is freely available at Zenodo. The eGRAM program with examples is freely available the at Zenodo and on the GitHub website (https://github.com/LinjieCodes/eGRAMv2R1 copy archived at *Lin, 2026*). The data on favored and hitchhiking mutations are available at Zenodo. RNA-seq data of cell lines before and after DBD KO of lncRNAs were deposited to the NCBI GEO database (https://www.ncbi.nlm.nih.gov/geo), with two accession numbers (accession number GSE213231; accession number is GSE229846).

The following datasets were generated:

| Author(s) | Year | Dataset title | Dataset URL | Database and Identifier |
|---|---|---|---|---|
| Zhu H | 2022 | Knockout of the DNA binding domains in human-specific lncRNAs causes changed expression of target genes | https://www.ncbi.nlm.nih.gov/geo/query/acc.cgi?acc=GSE213231 | NCBI Gene Expression Omnibus, GSE213231 |
| Zhu H | 2024 | A multi-cancer CRISPR/Cas9 reveals distinct gene dysregulation by noncoding MSTRG transcripts | https://www.ncbi.nlm.nih.gov/geo/query/acc.cgi?acc=GSE229846 | NCBI Gene Expression Omnibus, GSE229846 |
| Lin J | 2026 | Human-specific lncRNAs contributed critically to human evolution by distinctly regulating gene expression | https://doi.org/10.5281/zenodo.18919871 | Zenodo, 10.5281/zenodo.18919871 |

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

## Appendix 1

## Supporting evidence for DBS prediction

We used multiple methods and datasets to validate DBS prediction. First, since lncRNAs can recruit epigenetic regulatory enzymes to lncRNAs' DNA-binding sites, we uploaded predicted DBSs of HS lncRNAs to the UCSC Genome Browser as a custom track, compared these DBSs with tracks such as 'ENCODE DNA Methylation tracks' (*Meissner et al., 2008*) and 'ENCODE Histone Modification tracks' (*Ernst et al., 2011*). We found that predicted DBSs overlap well with experimentally identified DNA methylation and histone modification signals in multiple cell lines (*Appendix 1—figure 1*).

Second, since lncRNAs' DBSs can be detected genome-wide by experiments such as ChIRP-seq, we predicted the DBSs of the lncRNAs MALAT1, NEAT1, and MEG3 and compared predicted DBSs with the experimentally identified DNA-binding sites (*Mondal et al., 2015*; *West et al., 2014*). We found predicted DBSs agree well with experimentally identified DNA-binding sites (*Appendix 1—figure 2*).

Third, we found that many DBSs also co-localize with ENCODE Candidate Cis-Regulatory Elements (cCREs) in promoter regions of genes (*Appendix 1—figure 3*). cCREs are a subset of representative DNase hypersensitive sites across ENCODE and Roadmap Epigenomics samples supported by either histone modifications (H3K4me3 and H3K27ac) or CTCF-binding data.

Fourth, we used the CRISPR/Cas9 technique to knock out the sequences containing the DBD of seven lncRNAs (three HS lncRNAs and four wrongly transcribed long noncoding RNAs) in multiple cell lines and performed RNA-seq before and after the knockouts. The deleted sequences were just 100–200 bp, and gene expression analysis revealed that the |fold change| of target genes was significantly larger than the |fold change| of non-target genes (one-sided Mann–Whitney test, p = 3.1e−72, 1.49e−114, and 1.12e−206 for HS lncRNA RP13-516M14.1, RP11-426L16.8, and SNORA59B, and p = 2.58e−09, 6.49e−41, 0.034, and 5.23e−07 for the four wrongly transcribed lncRNAs in cancer cell lines) (*Appendix 1—figure 4*). These results suggest that the deletion of DBD causes the changed expression of target genes (*Appendix 1—figure 5*).

Fifth, the *LongTarget* program uses a variant of the Smith–Waterman algorithm to identify all local alignments in each lncRNA/DNA pair. Each alignment is an RNA:DNA triplex that consists of a TFO (triplex-forming oligonucleotides) and a TTS (triplex-targeting sites). A DBS is defined based on a set of densely overlapping TTSs, and a DBD is defined based on both TFOs and the DBS (*He et al., 2015*). *LongTarget* can more robustly and accurately identify DBS than some popular methods (*Wen et al., 2022*). To examine the likelihood that a local alignment of 147 bp (the average length of strong DBSs) can be generated by chance in a 5000-bp DNA sequence, we randomly simulated two sequences seqA and seqB (which were 5000 and 147 bp), and aligned seqB to seqA using the EMBOSS *Water* (https://www.ebi.ac.uk/Tools/psa/emboss_water/) local alignment program with alignment identity controlled to 60% (because the default *Identity* parameter of the *LongTarget* program was 60%). The equation $p = 1 - \exp\left(-Kmne^{-\lambda S}\right)$ was used to calculate a p-value to estimate the likelihood, in which $m$ and $n$ were the length of seqA and seqB, $K$ was a small adjusting constant (which was 0.1), $\lambda$ was the normalization coefficient of the score (which was 0.3), and $S$ was the alignment score. We repeated the simulation and calculation process 10,000 times and determined that the maximal and minimal p values were between 8.2e−19 and 1.5e−48. Thus, a DBS of 147 bp is extremely unlikely to be generated by chance. This result also supports that the changed expression of target genes is caused by the knockout of DBDs in the CRISPR/Cas9 experiments.

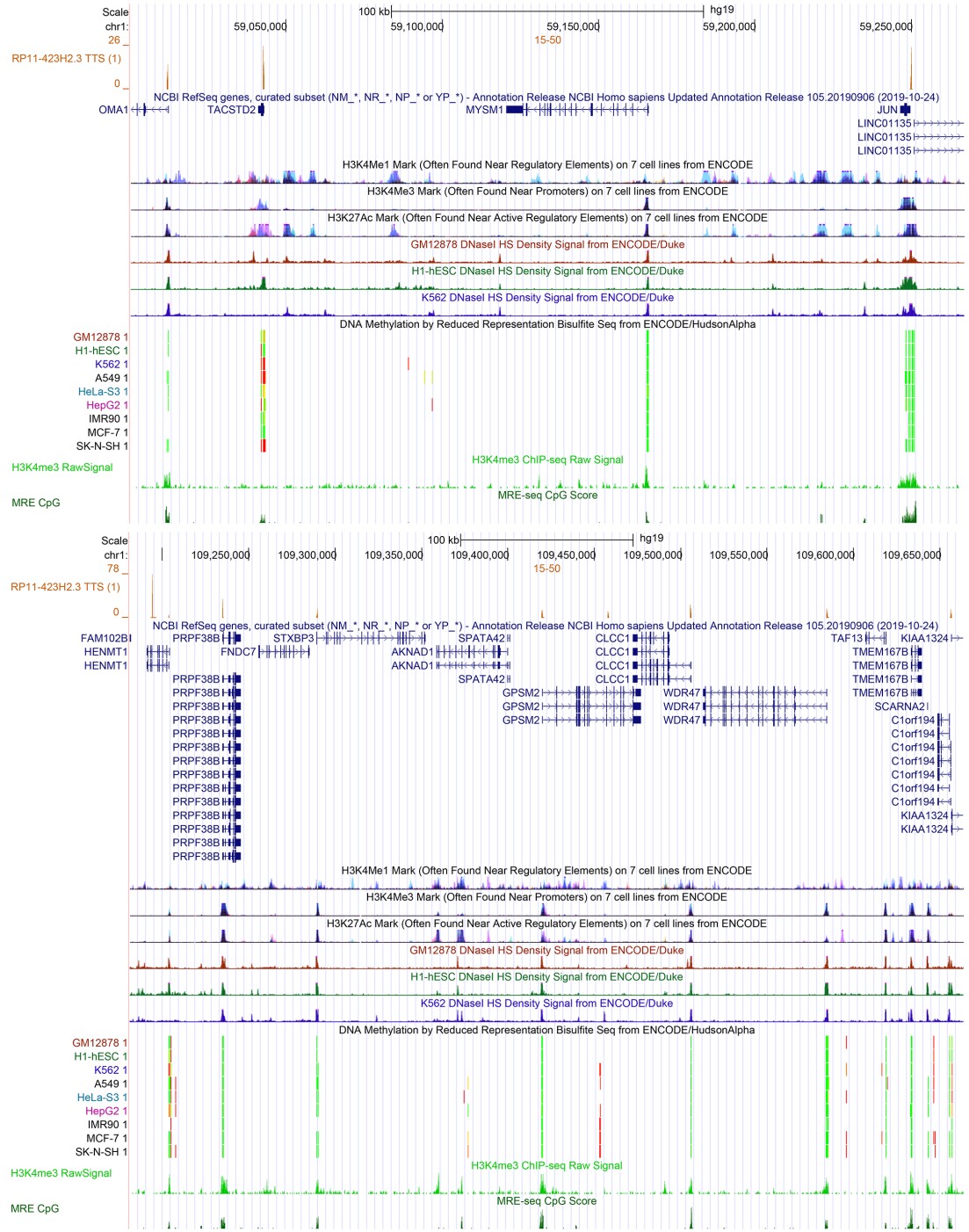

**Appendix 1—figure 1.** DBSs of the HS lncRNA RP11-423H2.3 in two genomic regions. In the upper and lower panels that display two genomic regions, the tracks from top to bottom are DBSs (orange peaks), gene annotation, histone modification signals in cell lines, DNA methylation signals in cell lines, H3K4me3 RawSignal, and MRE CpG signals. DBSs overlap very well with DNA methylation and histone modification signals in multiple cell lines.

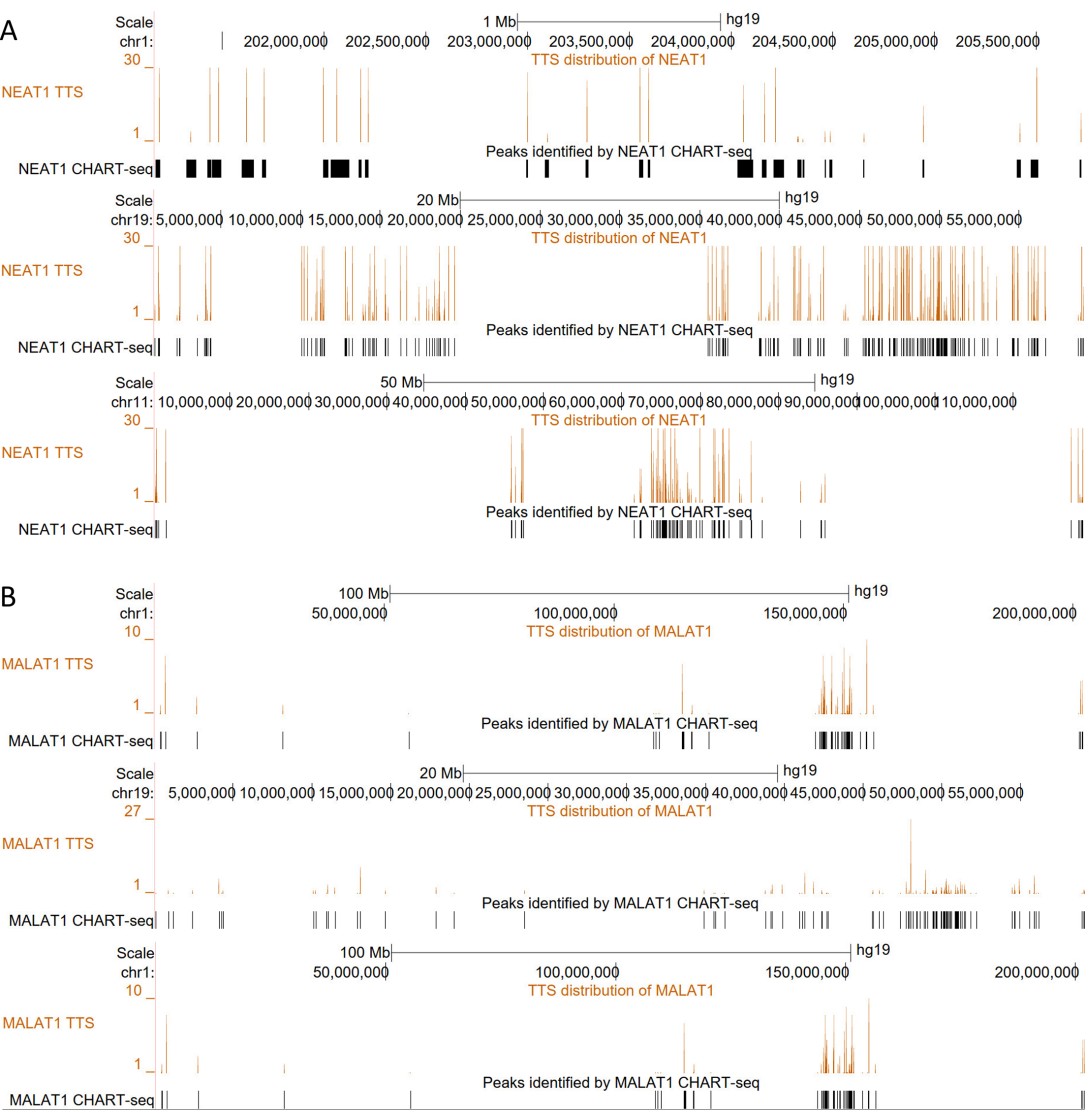

**Appendix 1—figure 2.** Predicted DBSs and experimentally identified (by CHART-seq) DNA-binding sites of NEAT1 and MALAT1 in two cell lines (**West et al., 2014**). DBSs were predicted using the DNA sequences of CHART-seq peaks. 99% and 87% of experimentally identified DNA-binding sites of NEAT1 and MALAT1 overlap with predicted DBSs. (**A**) Predicted DBSs and experimentally identified DNA-binding sites of NEAT1 in three genomic regions. (**B**) Predicted DBSs and experimentally identified DNA-binding sites of MALAT1 in three genomic regions.

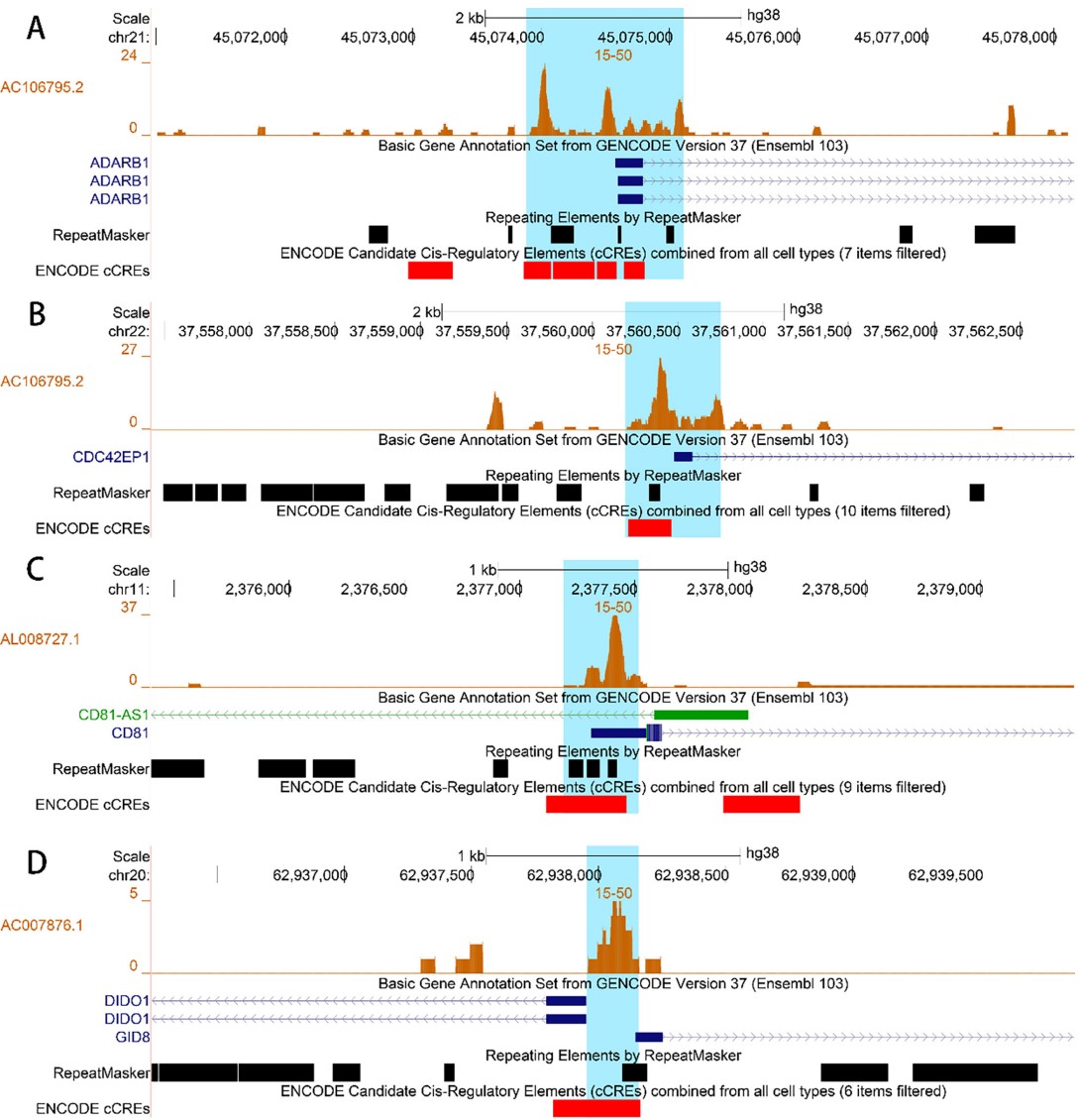

**Appendix 1—figure 3.** Examples of co-localization of DBSs, TEs, and cCREs in the promoter regions of genes. (**A**) The DBSs of AC106795.2 in the promoter region of *ADARB1*. (**B**) The DBSs of AC106795.2 in the promoter region of *CDC42EP1*. (**C**) The DBS of AL008727.1 in the promoter region of *CD81*. (**D**) The DBS of AC007876.1 in the promoter region of *DIDO1* and *GID8*.

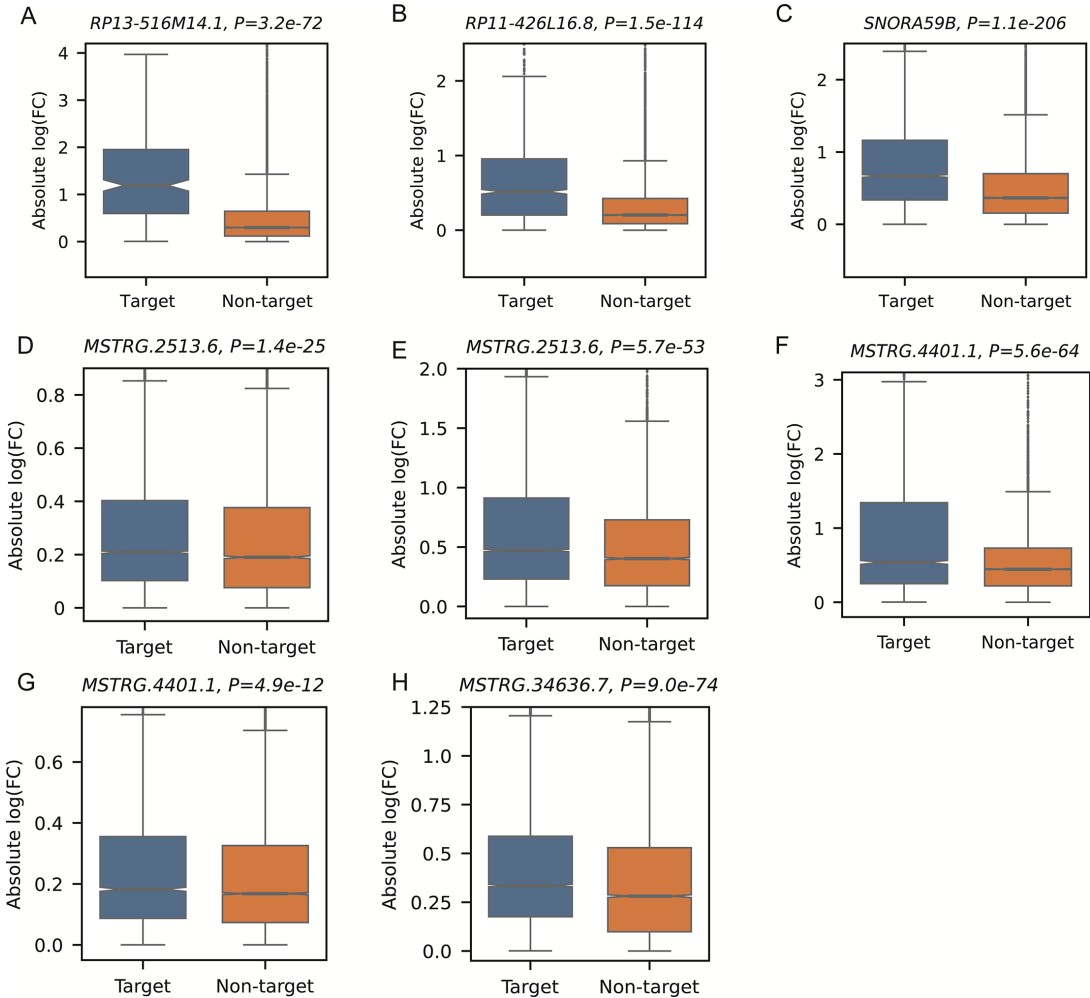

**Appendix 1—figure 4.** The expression change of target genes was significantly larger than that of non-target genes after DBD knockout. The fold change of gene expression was computed using the edgeR package. The |fold change| distribution of target genes was compared with the |fold change| distribution of non-target genes (one-sided Mann–Whitney test). (**A**) The knockout of a 157-bp sequence (chr17:80252565–80252721) which contains the DBD of RP13-516M14.1, in the HeLa cell line. (**B**) The knockout of a 202-bp sequence (chr1:113392603–113392804) which contains the DBD of RP11-426L16.8, in the RKO cell line. (**C**) The knockout of a 198-bp sequence (chr17:19460524–19460721) which contains the DBD of SNORA59B, in the SK-MES-1 cell line. (**D, E**) The knockout of the DBD of a wrongly transcribed long noncoding RNA (chr1:156641670–156661464) in the A549 cell line and the HCT116 cell line. (**F, G**) The knockout of the DBD of a wrongly transcribed long noncoding RNA (chr10:52443915–52455313) in the A549 cell line and the HCT116 cell line. These wrongly transcribed long noncoding RNAs are labeled as 'MSTRG' transcripts by the *Stringtie* package. (**H**) The knockout of the DBD of a third wrongly transcribed long noncoding RNA in the HCT116 cell line.

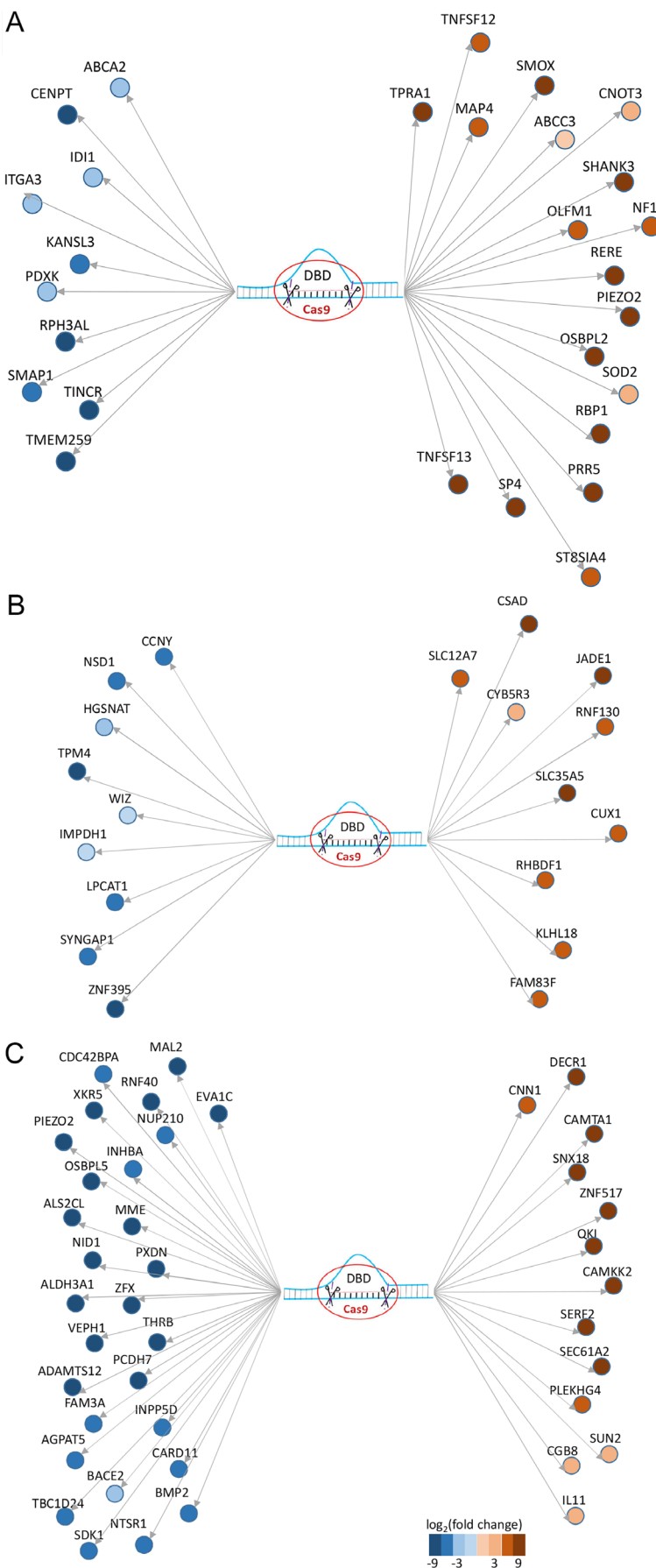

**Appendix 1—figure 5.** Significant up- and downregulation ($|\log_2(\text{fold change})| > 1$, FDR <0.1) of target genes after DBD knockout. (**A**) RP13-516M14.1. (**B**) RP11-426L16.8. (**C**) SNORA59B.

## Appendix 2

### Features of HS lncRNA-mediated gene expression regulation

HS lncRNA-mediated gene expression regulation shows multiple features. First, HS lncRNAs themselves form complex targeting relationships (*Appendix 2—figure 1*), suggesting networks and cascades of regulation. Second, some DBSs are human-specific sequences (*Appendix 2—figure 2*), suggesting that human-specific sequences were explored by HS lncRNAs for regulating gene expression. Third, many genes and transcripts have DBSs for multiple HS lncRNAs (*Appendix 2—figure 3*), suggesting co-regulation of functionally related genes. Fourth, certain genes and transcripts have multiple DBSs for one HS lncRNA, suggesting tissue-specific regulation. Fifth, selection signals were detected in some DBSs in specific populations (*Appendix 2—figure 4*). Sixth, generally, HS lncRNAs on the Y chromosome have longer DBSs than HS lncRNAs on the autosomes (*Appendix 2—figure 5*). Finally, SNPs in the DBSs of an HS lncRNA in multiple genes on a chromosome show linkage disequilibrium (LD), suggesting an association between these DBSs (*Appendix 2—figure 6*).

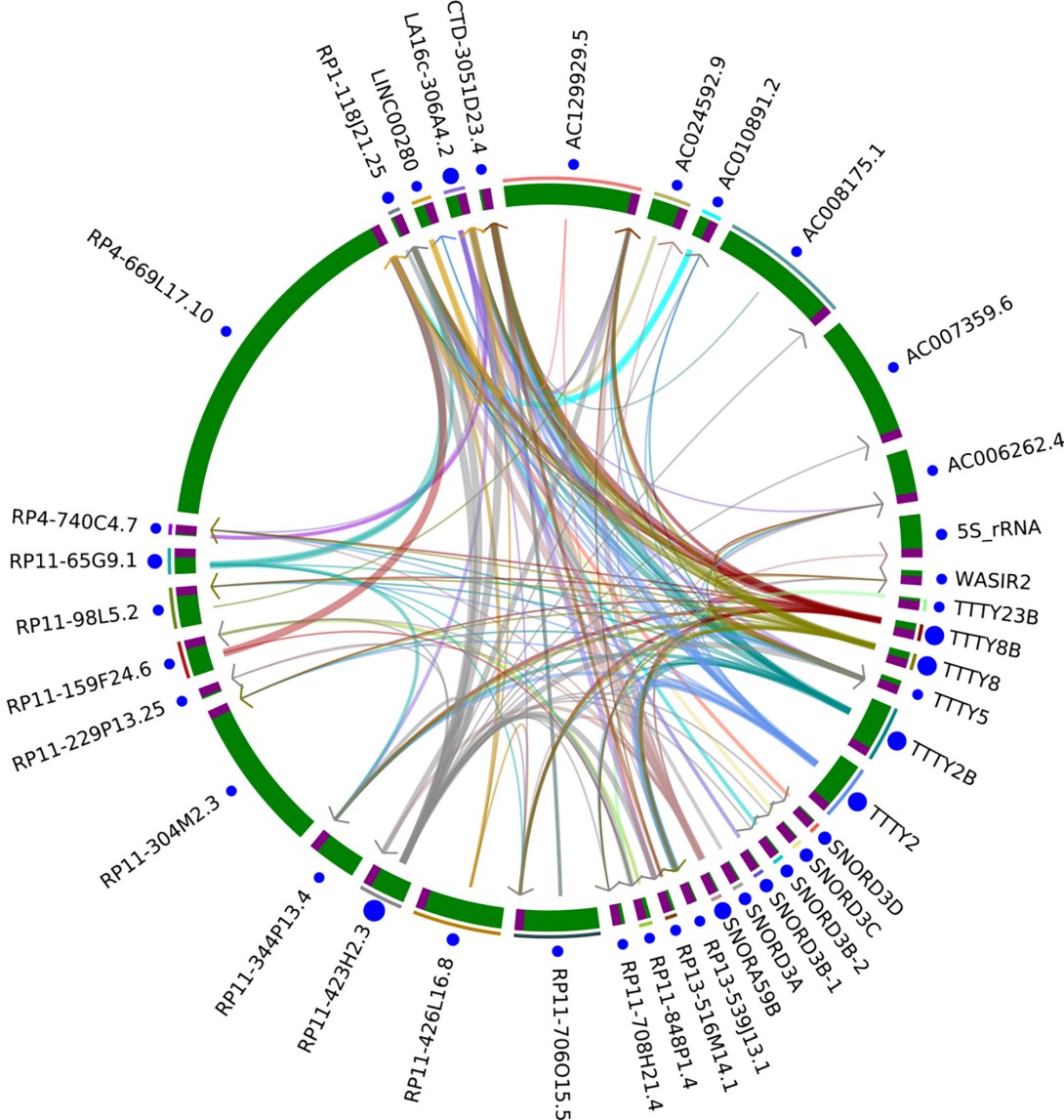

**Appendix 2—figure 1.** Potential targeting regulation between HS lncRNAs. The circle's brown and green regions indicate promoter and gene body regions. Arrows indicate the direction from the gene body to the promoter regions. The width of the arrows indicates the binding affinity of DBSs, and the sizes of blue dots indicate the number of DBSs of the lncRNA in the genome.

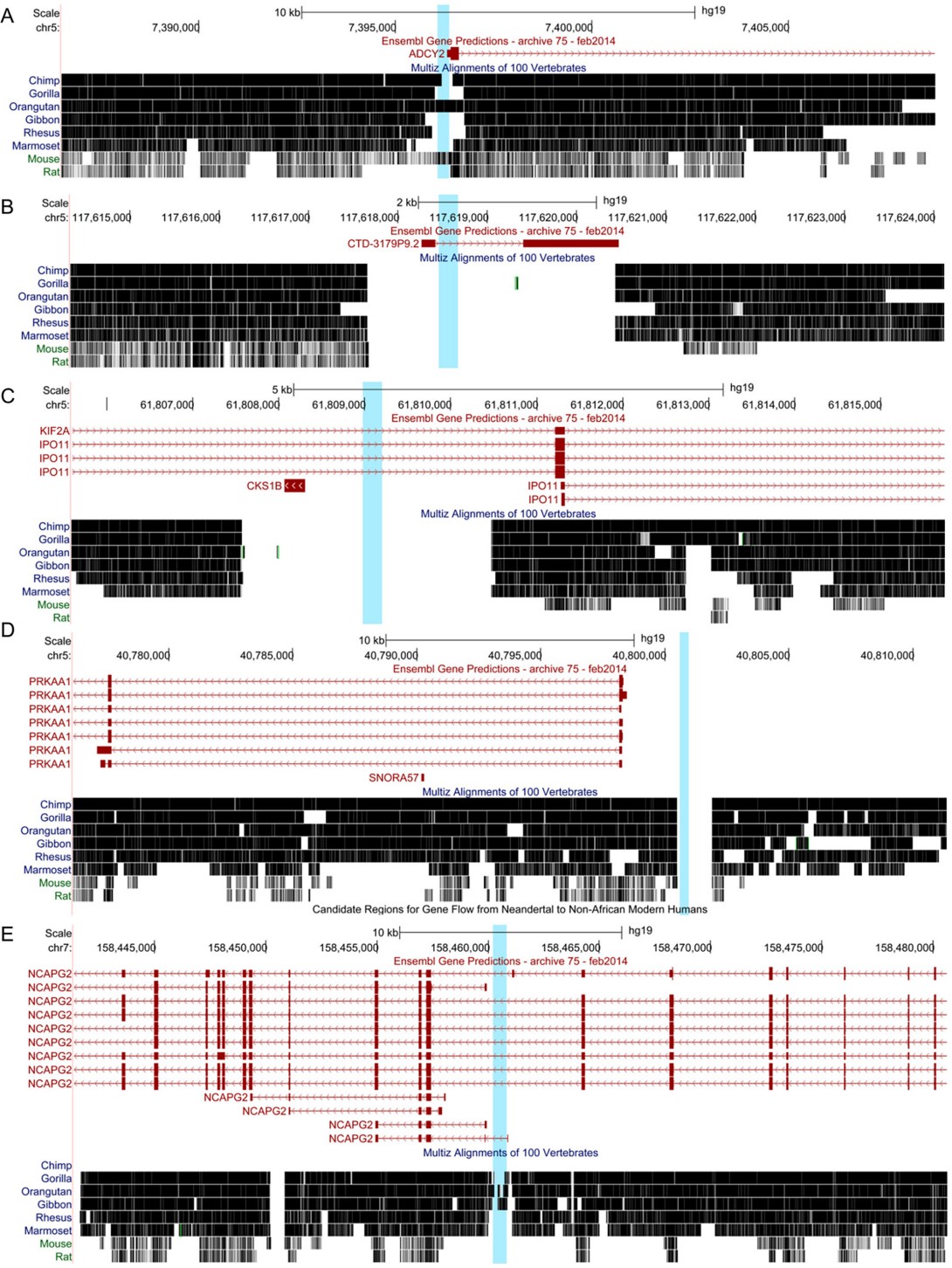

**Appendix 2—figure 2.** Some DBSs (indicated by blue bars) are in human-specific genome sequences. (**A–D**) The DBSs of RP11-848P1.4 in the genes *ADCY2*, *CTD-3179P9.2*, *IPO11*, and *PRKAA1*. (**E**) The DBS of RP11-598D14.1 in the gene *NCAPG2*.

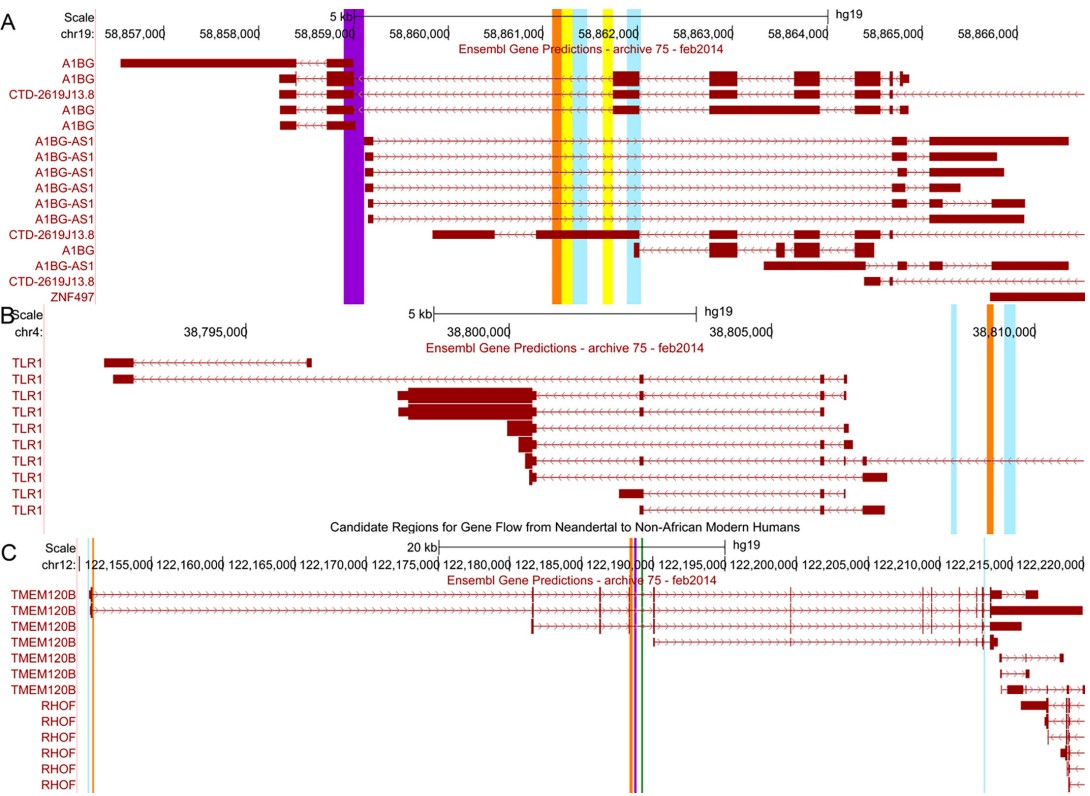

**Appendix 2—figure 3.** Many genes and transcripts contain DBSs for multiple HS lncRNAs. (**A**) Left to right: the DBSs of RP11-65G9.1, LA16c-306A4.2, RP13-516M14.1, SNORA59B, RP11-423H2.3, and TTTY8/8B in the *A1BG*. (**B**) Left to right: the DBSs of TTTY8/8B, RP4-669L17.10, and RP11-423H2.3 in *TLR1*. (**C**) Left to right: the DBSs of LA16c-306A4.2, RP11-423H2.3, RP11-423H2.3, RP1-118J21.25, RP11-706O15.5, and SNORA59B in *TMEM210B*.

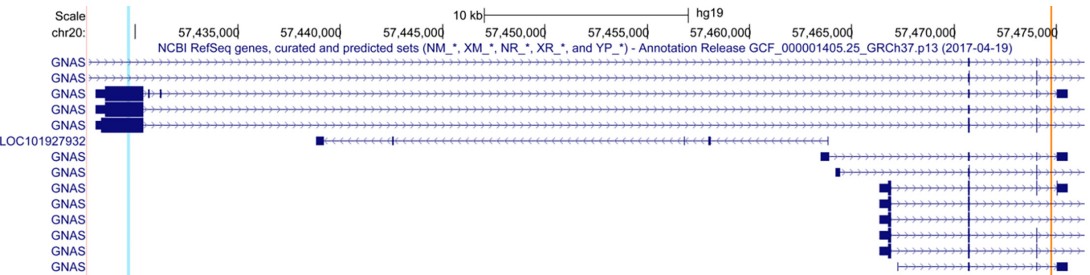

**Appendix 2—figure 4.** In the *GNAS* region, RP11-423H2.3 has a DBS (indicated by the blue bar) wherein a selection signal was detected in CEU and CHB (Tajima's *D* = −0.99/−1.13/1.86 in CEU/CHB/YRI, integrated Fst = 0.22), and has another DBS (indicated by the orange bar) wherein a selection signal was detected in YRI (Tajima's *D* = 0.25/1.09/−1.17 in CEU/CHB/YRI, integrated Fst = 0.33).

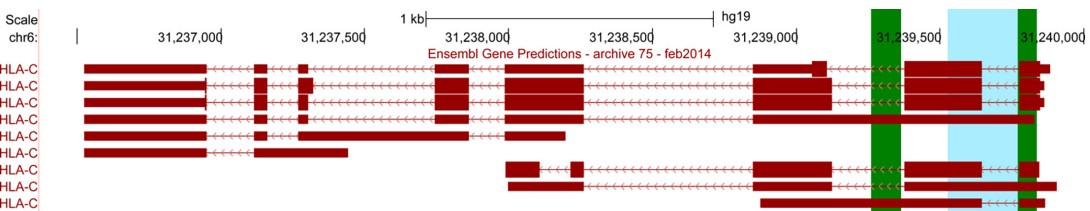

**Appendix 2—figure 5.** HS lncRNAs on the Y chromosome often have longer DBSs than HS lncRNAs on the autosomes. The top panel shows that the DBS of TTTY2/2B in *HLA-C* (indicated by the blue bar) is longer than the two DBSs of RP11-423H2.3 (indicated by the green bars). The bottom panel shows that the DBS of TTTY8/8B in *IFNAR1* (indicated by the blue bar) is longer than the DBS of LINC00279 (indicated by the green bar).

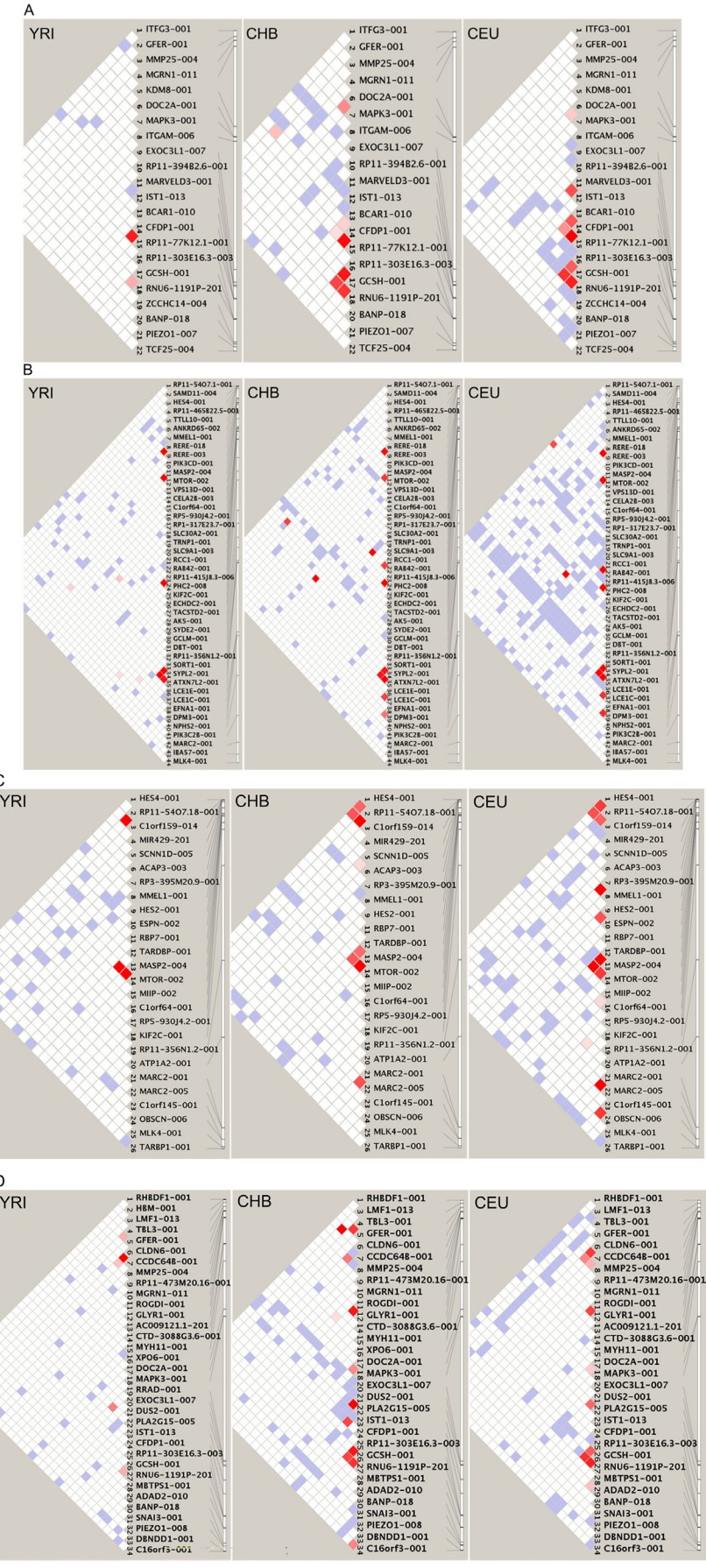

**Appendix 2—figure 6.** The linkage disequilibrium (LD) of the key SNP in DBSs of HS lncRNAs in genes on some chromosomes. (**A**) The LD of the key SNP in the DBSs of LA16c-306A4.2 in some genes on chromosome 16. (**B**) The LD of the key SNP in the DBSs of RP11-423H2.3 in some genes on chromosome 1. (**C**) The LD of the key SNP in the DBSs of SNORA59B in some genes on chromosome 1. (**D**) The LD of the key SNP in the DBSs of TTTY8B in some genes on chromosome 16.

## Appendix 3

### Target genes with specific DBS features are enriched in specific functions

Strong and weak DBSs may indicate well-established and recently occurred epigenetic regulation by HS lncRNAs. To examine if genes with DBSs of different features are enriched in different biological functions, we performed over-representation analyses (ORA) using the *g:Profiler* program (Organism = *Homo sapiens*, Ordered query = No, Significance threshold = Benjamini–Hochberg FDR, User threshold = 0.05, 50 < terms size <1000) and the Gene Ontology (GO) database multiple times.

First, we examined the top 2000 genes and bottom 2000 genes with the strongest and weakest DBSs. ORA results of the two gene sets share many GO terms, and it is notable that genes with weak DBSs are enriched in slightly more GO terms (*Appendix 3—table 1*). Next, we computed DBS sequence distances (per-base distances) in two ways. The first were the distances from the reconstructed human ancestor to chimpanzees, archaic humans, and modern humans; the second were the distances from modern humans to chimpanzees and archaic humans. We found that, when the human–chimpanzee ancestral sequence has the ancestral sequence (which means the inference of ancestral allele is of high confidence), more DBSs have a distance >0.015 from the human ancestor to archaic humans than to modern humans (*Appendix 3—figure 1*). According to the file 'homo_sapiens_ancestor_GRCh37_e71.README' and the paper '1000 Genomes Project Consortium. A global reference for human genetic variation. Nature 2015, a high-confidence call in the human ancestor sequence is made when all three sequences – the ancestral (the common ancestor of humans and chimpanzees), the sister (chimpanzees), and the ancestral of the ancestral sequences – agree. We found that only about 64% of calls in the human ancestor sequence are high-confidence calls, and that only DBSs in human ancestral loci with high confidence have correct distances to the five leaf nodes. We therefore computed DBS distances from modern humans to archaic humans and chimpanzees.

To examine whether DBS distances computed using the second method reflect genetic changes in the human lineage or the chimpanzee lineage, we also computed DBS sequence distances with the most changed sequences between humans and gorillas. We found that DBS distances between humans and chimpanzees were significantly correlated with those between humans and gorillas (Spearman's rho = 0.57, p = 0.0) and that these DBSs had larger distances between humans and gorillas. These results suggest that the sequence differences of the most changed DBSs between humans and chimpanzees are determined mainly by genetic changes occurring in the human lineage, but not in the chimpanzee lineage. The same results were observed when the archaic humans were examined (*Appendix 3—figure 2*).

When distances were computed using the second method, for DBSs without chimpanzee counterparts, we assumed their human–chimpanzee distances were 10.0. Then, ORA was performed in two ways, based on genes sorted by human–chimpanzee DBS distance and human-Altai Neanderthal DBS distance, respectively. First, we used the top 25% and bottom 25% of genes to examine whether genes with large DBS distances are enriched for more human evolution-related GO terms than genes with small DBS distances (*Supplementary file 1F, G*). Second, we used the top 50% and bottom 50% of genes, intersected with ASE genes reported by *Agoglia et al., 2021*; *Supplementary file 1F, H*; *Agoglia et al., 2021*. For genes intersected with significant ASE genes (p-adj <0.01), those with large DBS distances are enriched with more GO terms and also more human evolution-related GO terms than genes with small DBS distances (*Appendix 3—table 2*).

Finally, we classified strong and weak DBSs with 'mostly changed' sequence distances into the following six classes. Since only about 20% of genes have human–chimpanzee DBS distances ≥0.034, and much fewer genes have human-Altai Neanderthal DBS distances exceeding this value, 0.034 is a stringent (reliable) threshold for defining 'mostly changed'. DBSs with 'Human–Chimp distance >0.034 and Human–Altai Neanderthals distance = 0' (little distance change occurred since Altai Neanderthals) were defined as old ones, and DBSs with 'Human–Altai Neanderthals distance >0.034 or Human–Denisovan distance >0.034' ('mostly changed' since Altai Neanderthals or Denisovans) were defined as young ones. These six classes – strong old, strong young, strong others, weak old, weak young, and weak others – therefore reflect early and late, and strong and weak, DBS sequence changes in human evolution. Based on favored and hitchhiking mutations in

17 human populations (*Tang et al., 2022*; *Tang et al., 2023*), we examined the number of favored and hitchhiking mutations in each class. Favored and hitchhiking mutations are most enriched in the weak young class (*Appendix 3—table 3*).

**Appendix 3—table 1.** The enriched GO terms for the top 2000 and bottom 2000 genes with largest and smallest binding affinity.
Upper (black): Top 30 GO terms of the top 2000 genes (left) and the bottom 2000 genes (right).
Lower (blue): Bottom 13 GO terms of the top 2000 genes (left) and 30 of bottom GO terms of the bottom 2000 genes (right).

| GO terms (genes with strongest DBS) | Term_id | Adjusted_p | GO terms (genes with weakest DBS) | Term_id | Adjusted_p |
|---|---|---|---|---|---|
| Small GTPase mediated signal transduction | GO:0007264 | 8.55E−17 | Neuron projection development | GO:0031175 | 3.79E−11 |
| Neuron projection development | GO:0031175 | 2.36E−16 | Cell morphogenesis involved in differentiation | GO:0000904 | 5.36E−11 |
| Cell projection morphogenesis | GO:0048858 | 5.53E−16 | Cellular component morphogenesis | GO:0032989 | 6.81E−11 |
| Neuron projection morphogenesis | GO:0048812 | 8.56E−16 | Regulation of plasma membrane bounded cell projection organization | GO:0120035 | 2.58E−10 |
| Plasma membrane bounded cell projection morphogenesis | GO:0120039 | 8.56E−16 | Plasma membrane bounded cell projection morphogenesis | GO:0120039 | 3.44E−10 |
| Cell junction organization | GO:0034330 | 1.96E−15 | Regulation of anatomical structure morphogenesis | GO:0022603 | 4.34E−10 |
| Cell part morphogenesis | GO:0032990 | 5.24E−15 | Cell projection morphogenesis | GO:0048858 | 4.88E−10 |
| Synaptic signaling | GO:0099536 | 1.38E−14 | Cell part morphogenesis | GO:0032990 | 6.25E−10 |
| Cellular component morphogenesis | GO:0032989 | 3.76E−14 | Regulation of cell projection organization | GO:0031344 | 1.13E−09 |
| Trans-synaptic signaling | GO:0099537 | 1.88E−13 | Neuron projection morphogenesis | GO:0048812 | 1.64E−09 |
| Cell morphogenesis involved in differentiation | GO:0000904 | 3.16E−13 | Actin filament-based process | GO:0030029 | 1.65E−09 |
| Regulation of small GTPase mediated signal transduction | GO:0051056 | 4.03E−13 | Actin cytoskeleton organization | GO:0030036 | 1.99E−09 |
| Chemical synaptic transmission | GO:0007268 | 4.18E−13 | Organophosphate metabolic process | GO:0019637 | 2.05E−09 |
| Anterograde trans-synaptic signaling | GO:0098916 | 4.18E−13 | Cell morphogenesis involved in neuron differentiation | GO:0048667 | 2.47E−09 |
| Regulation of plasma membrane bounded cell projection organization | GO:0120035 | 1.72E−12 | Regulation of cellular component biogenesis | GO:0044087 | 8.54E−09 |
| Regulation of cell projection organization | GO:0031344 | 1.74E−12 | Regulation of neuron projection development | GO:0010975 | 1.22E−08 |
| Cell morphogenesis involved in neuron differentiation | GO:0048667 | 3.83E−12 | Cell junction organization | GO:0034330 | 3.32E−08 |

*Appendix 3—table 1 Continued on next page*

*Appendix 3—table 1 Continued*

| GO terms (genes with strongest DBS) | Term_id | Adjusted_p | GO terms (genes with weakest DBS) | Term_id | Adjusted_p |
|---|---|---|---|---|---|
| Dendrite development | GO:0016358 | 4.71E−11 | Organophosphate biosynthetic process | GO:0090407 | 3.82E−08 |
| Enzyme-linked receptor protein signaling pathway | GO:0007167 | 4.71E−11 | Growth | GO:0040007 | 3.85E−08 |
| Cell surface receptor signaling pathway involved in cell–cell signaling | GO:1905114 | 4.86E−11 | Developmental growth | GO:0048589 | 5.54E−08 |
| Actin filament-based process | GO:0030029 | 7.42E−11 | Positive regulation of protein modification process | GO:0031401 | 9.39E−08 |
| Regulation of transmembrane transport | GO:0034762 | 8.88E−11 | Regulation of cell morphogenesis | GO:0022604 | 1.08E−07 |
| Synapse organization | GO:0050808 | 1.07E−10 | Negative regulation of cellular component organization | GO:0051129 | 1.64E−07 |
| Regulation of cellular component biogenesis | GO:0044087 | 1.17E−10 | Lipid biosynthetic process | GO:0008610 | 2.44E−07 |
| Metal ion transport | GO:0030001 | 4.76E−10 | Positive regulation of transport | GO:0051050 | 3.52E−07 |
| Ras protein signal transduction | GO:0007265 | 5.80E−10 | Regulation of locomotion | GO:0040012 | 4.21E−07 |
| Regulation of ion transport | GO:0043269 | 7.96E−10 | Organelle assembly | GO:0070925 | 4.42E−07 |
| Modulation of chemical synaptic transmission | GO:0050804 | 8.08E−10 | Regulation of cell migration | GO:0030334 | 4.90E−07 |
| Regulation of trans-synaptic signaling | GO:0099177 | 8.80E−10 | Mitotic cell cycle | GO:0000278 | 6.42E−07 |
| Cation transmembrane transport | GO:0098655 | 1.65E−09 | Synapse organization | GO:0050808 | 6.74E−07 |
| Absent speech | HP:0001344 | 1.54E−02 | Glycolipid metabolic process | GO:0006664 | 3.80E−02 |
| Abnormal aggressive, impulsive, or violent behavior | HP:0006919 | 1.54E−02 | Carboxylic acid catabolic process | GO:0046395 | 3.80E−02 |
| Autistic behavior | HP:0000729 | 1.54E−02 | Regulation of epithelial cell proliferation | GO:0050678 | 3.83E−02 |
| Absent toe | HP:0010760 | 1.89E−02 | Response to radiation | GO:0009314 | 3.85E−02 |
| Abnormality of calvarial morphology | HP:0002648 | 1.89E−02 | Protein methylation | GO:0006479 | 3.86E−02 |
| Aplasia/hypoplasia of toe | HP:0001991 | 1.89E−02 | Protein alkylation | GO:0008213 | 3.86E−02 |
| Tall stature | HP:0000098 | 1.89E−02 | Golgi organization | GO:0007030 | 3.88E−02 |
| Short philtrum | HP:0000322 | 1.89E−02 | Membrane depolarization | GO:0051899 | 3.97E−02 |

*Appendix 3—table 1 Continued*

| GO terms (genes with strongest DBS) | Term_id | Adjusted_p | GO terms (genes with weakest DBS) | Term_id | Adjusted_p |
|---|---|---|---|---|---|
| Motor stereotypy | HP:0000733 | 3.37E−02 | Skeletal system morphogenesis | GO:0048705 | 3.98E−02 |
| Slender finger | HP:0001238 | 3.37E−02 | Positive chemotaxis | GO:0050918 | 3.98E−02 |
| Asymmetric growth | HP:0100555 | 3.37E−02 | Development of primary sexual characteristics | GO:0045137 | 3.98E−02 |
| Abnormal upper limb bone morphology | HP:0040070 | 3.37E−02 | Metaphase/anaphase transition of cell cycle | GO:0044784 | 3.98E−02 |
| Long fingers | HP:0100807 | 4.09E−02 | Non-motile cilium assembly | GO:1905515 | 3.98E−02 |
| | | | Muscle cell differentiation | GO:0042692 | 4.55E−02 |
| | | | Cell activation involved in immune response | GO:0002263 | 4.73E−02 |
| | | | Regulation of exocytosis | GO:0017157 | 4.74E−02 |
| | | | Negative regulation of chromosome organization | GO:2001251 | 4.76E−02 |
| | | | ADP metabolic process | GO:0046031 | 4.76E−02 |
| | | | Cytoskeleton-dependent cytokinesis | GO:0061640 | 4.76E−02 |
| | | | Regulation of canonical Wnt signaling pathway | GO:0060828 | 4.77E−02 |
| | | | Olefinic compound metabolic process | GO:0120254 | 4.77E−02 |
| | | | DNA geometric change | GO:0032392 | 4.77E−02 |
| | | | Gonad development | GO:0008406 | 4.77E−02 |
| | | | Reproductive system development | GO:0061458 | 4.77E−02 |
| | | | Vasculature development | GO:0001944 | 4.77E−02 |
| | | | Response to insulin | GO:0032868 | 4.79E−02 |
| | | | Ribonucleotide biosynthetic process | GO:0009260 | 4.79E−02 |
| | | | Organic acid biosynthetic process | GO:0016053 | 4.82E−02 |
| | | | Vacuole organization | GO:0007033 | 4.84E−02 |
| | | | Import across plasma membrane | GO:0098739 | 4.95E−02 |

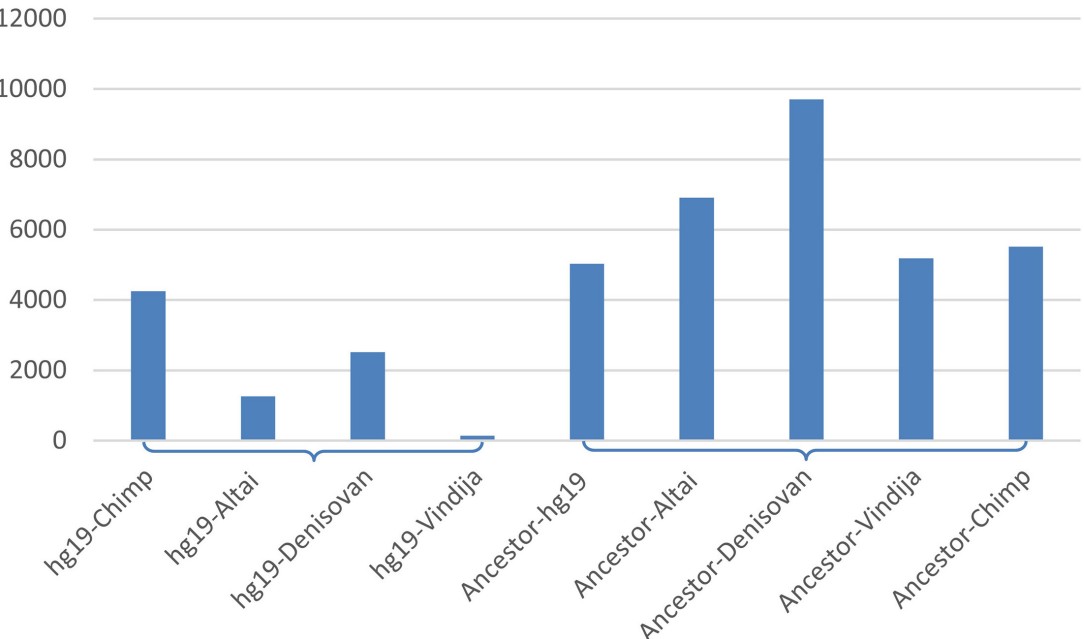

**Appendix 3—figure 1.** Numbers of DBSs with large distances from modern humans to archaic humans and chimpanzees, and from the human ancestor to chimpanzees, archaic humans, and modern humans. Left: DBSs in 4248, 1256, 2513, and 134 genes have distances >0.034 from modern humans to chimpanzees, Altai Neanderthals, Denisovans, and Vindija Neanderthals. Right: DBSs in 5033, 6908, 9707, 5189, and 5521 genes have distances >0.015 from the ancestor to modern humans, Altai Neanderthals, Denisovans, Vindija Neanderthals, and chimpanzees.

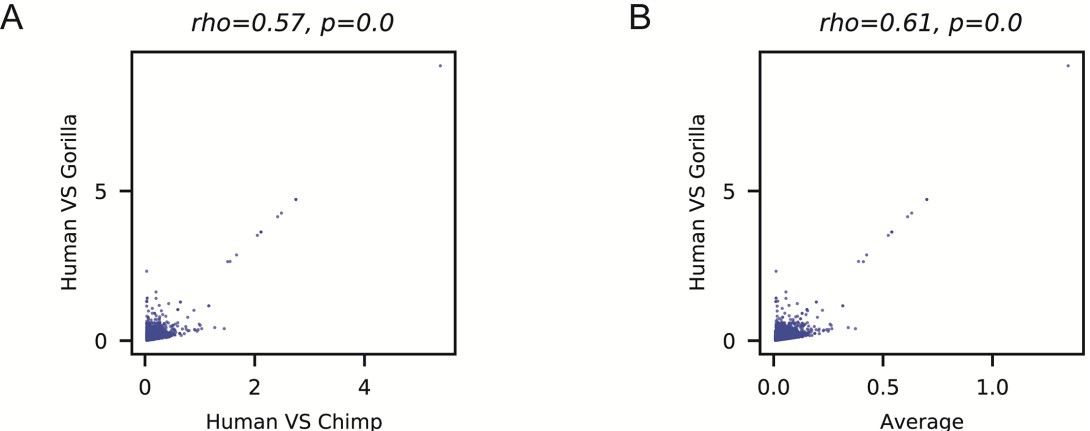

**Appendix 3—figure 2.** The most changed DBSs also have large sequence distances between humans and gorillas. (**A**) Scatter plot showing the sequence distances between humans and chimpanzees and between humans and gorillas. (**B**) The scatter plot shows the average sequence distances between humans and chimpanzees, the three archaic humans, and between humans and gorillas. The rho and p values were estimated using the Spearman correlation test.

**Appendix 3—table 2.** Enriched GO terms of different sets of genes with large and small DBS distances from humans to chimpanzees and Altai Neanderthals.

Shown are the presence and absence of GO terms highly related to human evolution. The intersections of genes sorted by DBS distance from humans to chimpanzees and to Altai Neanderthals, respectively, and genes showing significant ASE (adj-p <0.01 and |LFC| >0.5).

| Intersection of top 50% of genes (sorted by DBS distance from humans to chimpanzees) and ASE genes (adj-p <0.01) | term_ID | adj_p | Intersection of bottom 50% of genes (sorted by DBS distance from humans to chimpanzees) and ASE genes (adj-p <0.01) | term_ID | adj_p |
|---|---|---|---|---|---|
| Cellular pigmentation | GO:0033059 | 2.41E−06 | Brain development | GO:0007420 | 7.80E−03 |
| Behavior | GO:0007610 | 5.78E−05 | Forebrain development | GO:0030900 | 4.34E−02 |
| Pigmentation | GO:0043473 | 7.68E−05 | | | |
| Learning | GO:0007612 | 3.60E−04 | | | |
| Associative learning | GO:0008306 | 2.08E−03 | | | |
| Adaptive thermogenesis | GO:1990845 | 2.28E−03 | | | |
| Sensory system development | GO:0048880 | 3.19E−03 | | | |
| Cold-induced thermogenesis | GO:0106106 | 3.78E−03 | | | |
| Digestive system development | GO:0055123 | 3.94E−03 | | | |
| Glucose metabolic process | GO:0006006 | 3.97E−03 | | | |
| Learning or memory | GO:0007611 | 4.69E−03 | | | |
| Cognition | GO:0050890 | 6.32E−03 | | | |
| Regulation of cold-induced thermogenesis | GO:0120161 | 6.33E−03 | | | |
| Memory | GO:0007613 | 4.42E−02 | | | |
| Alcohol metabolic process | GO:0006066 | 4.82E−02 | | | |
| **Intersection of top 50% of genes (sorted by DBS distance from humans to Altai Neanderthals) and ASE genes (adj-p <0.01)** | **term_ID** | **adj_p** | **Intersection of bottom 50% of genes (sorted by DBS distance from humans to Altai Neanderthals) and ASE genes (adj-p <0.01)** | **term_ID** | **adj_p** |
| Behavior | GO:0007610 | 2.52E−07 | Pigmentation | GO:0043473 | 4.39E−04 |
| Glucose metabolic process | GO:0006006 | 9.29E−04 | Cellular pigmentation | GO:0033059 | 2.74E−03 |
| Sensory system development | GO:0048880 | 1.05E−03 | Brain development | GO:0007420 | 4.96E−02 |
| Learning | GO:0007612 | 1.77E−03 | | | |
| Learning or memory | GO:0007611 | 3.62E−03 | | | |
| Cognition | GO:0050890 | 4.42E−03 | | | |
| Associative learning | GO:0008306 | 6.95E−03 | | | |
| Digestive system development | GO:0055123 | 8.54E−03 | | | |

*Continued on next page*

*Continued*

| Intersection of top 50% of genes (sorted by DBS distance from humans to Altai Neanderthals) and ASE genes (adj-p <0.01) | term_ID | adj_p | Intersection of bottom 50% of genes (sorted by DBS distance from humans to Altai Neanderthals) and ASE genes (adj-p <0.01) | term_ID | adj_p |
|---|---|---|---|---|---|
| Cold-induced thermogenesis | GO:0106106 | 1.43E−02 | | | |
| Adaptive thermogenesis | GO:1990845 | 1.50E−02 | | | |
| Brain development | GO:0007420 | 1.85E−02 | | | |
| Forebrain development | GO:0030900 | 2.04E−02 | | | |
| Regulation of cold-induced thermogenesis | GO:0120161 | 2.28E−02 | | | |
| Alcohol metabolic process | GO:0006066 | 2.38E−02 | | | |
| Memory | GO:0007613 | 2.72E−02 | | | |
| Visual behavior | GO:0007632 | 4.57E−02 | | | |

**Appendix 3—table 3.** Numbers of favored and hitchhiking mutations in different classes of DBSs.

| Hitchhiking SNPs | Strong old | Strong young | Strong others | Weak old | Weak young | Weak others |
|---|---|---|---|---|---|---|
| | 3/15,685 | 11/163,007 | 78/170,389 | 10/180,505 | 44/47,251 | 57/168,692 |

| Favored SNPs | Strong old | Strong young | Strong others | Weak old | Weak young | Weak others |
|---|---|---|---|---|---|---|
| | 0/10,216 | 1/16,040 | 4/92,153 | 0/26,532 | 5/31,242 | 2/108,014 |

## Appendix 4

### Positive selection signals in HS lncRNA genes

We used multiple tests, including XP-CLR (*Chen et al., 2010*), iSAFE (*Akbari et al., 2018*), Tajima's *D* (*Tajima, 1989*), the fixation index (Fst) (*Weir and Cockerham, 1984*), and LD (*Slatkin, 2008*), to detect positive selection signals in HS lncRNA genes.

First, we used the XP-CLR program to scan the genome regions that contain HS lncRNAs and their 500 kb upstream and downstream sequences. Six pairwise comparisons were applied to the three human populations (CEU–CHB, CEU–YRI, CHB–CEU, CHB–YRI, YRI–CEU, and YRI–CHB). Selective sweeps were detected in the genome regions containing RP11-848P1.4 and RP11-598D14.1 in the CEU–YRI and CHB–YRI comparisons (*Appendix 4—figure 1*). Abundant SNPs in these regions have low derived allele frequency (DAF) in YRI, but are nearly fixed in CEU and CHB. Examples (with DAF in YRI, CEU, and CHB) include rs7208589 (DAF = 0.162, 0.990, and 0.995), rs8073226 (DAF = 0.148, 0.990, and 0.995), and rs9915124 (DAF = 0.181, 0.990, and 0.995) in the region containing RP11-848P1.4, and rs4690648 (DAF = 0.185, 0.939, and 0.917), rs11722101 (DAF = 0.269, 0.970, and 0.917), and rs11730933 (DAF = 0.292, 0.970, and 0.917) in the region containing RP11-598D14.1.

Second, we used the iSAFE program to detect favored mutations in HS lncRNA genes in these populations. Because selective sweeps caused by favored mutations may have varied lengths, we ran iSAFE four times with genome regions of 80, 500, 1000, and 2500 kb, which centered at the HS lncRNA. Selective sweeps and favored mutations were detected robustly in genome regions containing RP11-598D14.1, RP11-848P1.4, AC006129.1, AC006129.4, CTD-2291D10.1, CTD-2291D10.2, and LA16c-306A4.2 in CEU and CHB, and in the genome region containing CTD-3051D23.4 in CHB (*Appendix 4—figure 2*). Mutations with top iSAFE scores were located in the gene body regions of HS lncRNAs, and their DAF values were low in YRI but high in CEU and CHB. In addition, all of the detected favored mutations have high LD ($r^2$) scores.

Third, we used Tajima's *D* and integrated Fst to detect positive selection signals in gene body regions of HS lncRNAs. Positive selection signals were detected in RP11-598D14.1, RP11-848P1.4, RP11-344P13.4, and RP11-426L16.8 in CEU and CHB, in AC129929.5 and RP11-423H2.3 in CEU, and in CTB-151G24.1 in CHB (*Appendix 4—figure 3*). Since Tajima's *D* values were referenced with the genome-wide background, $D < 0$ and $D > 0$ indicate positive (or directional selection) and balancing selection, respectively, instead of population demography dynamics. The Fst of each HS lncRNA gene, also referenced with the genome-wide background, was computed for the CEU–YRI, CHB–YRI, and CHB–CEU comparisons. Extreme Fst values of SNPs were detected in RP11-598D14.1 and AC129929.5 in the comparisons of CEU–YRI and CHB–YRI. Since Fst values were referenced with the genome-wide background, extreme Fst values indicate positive selection.

Finally, we applied LD analysis to each HS lncRNA gene. Significantly increased LD was detected in SNPs in AC024592.9, AC129929.5, RP11-157B13.7, RP11-277P12.10, CTD-2142D14.1, and CTD-2291D10.1 in CEU and CHB (*Appendix 4—figure 4*). Taken together, the above results indicate that HS lncRNA genes may have undergone more significant adaptive evolution in CEU and CHB than in YRI.

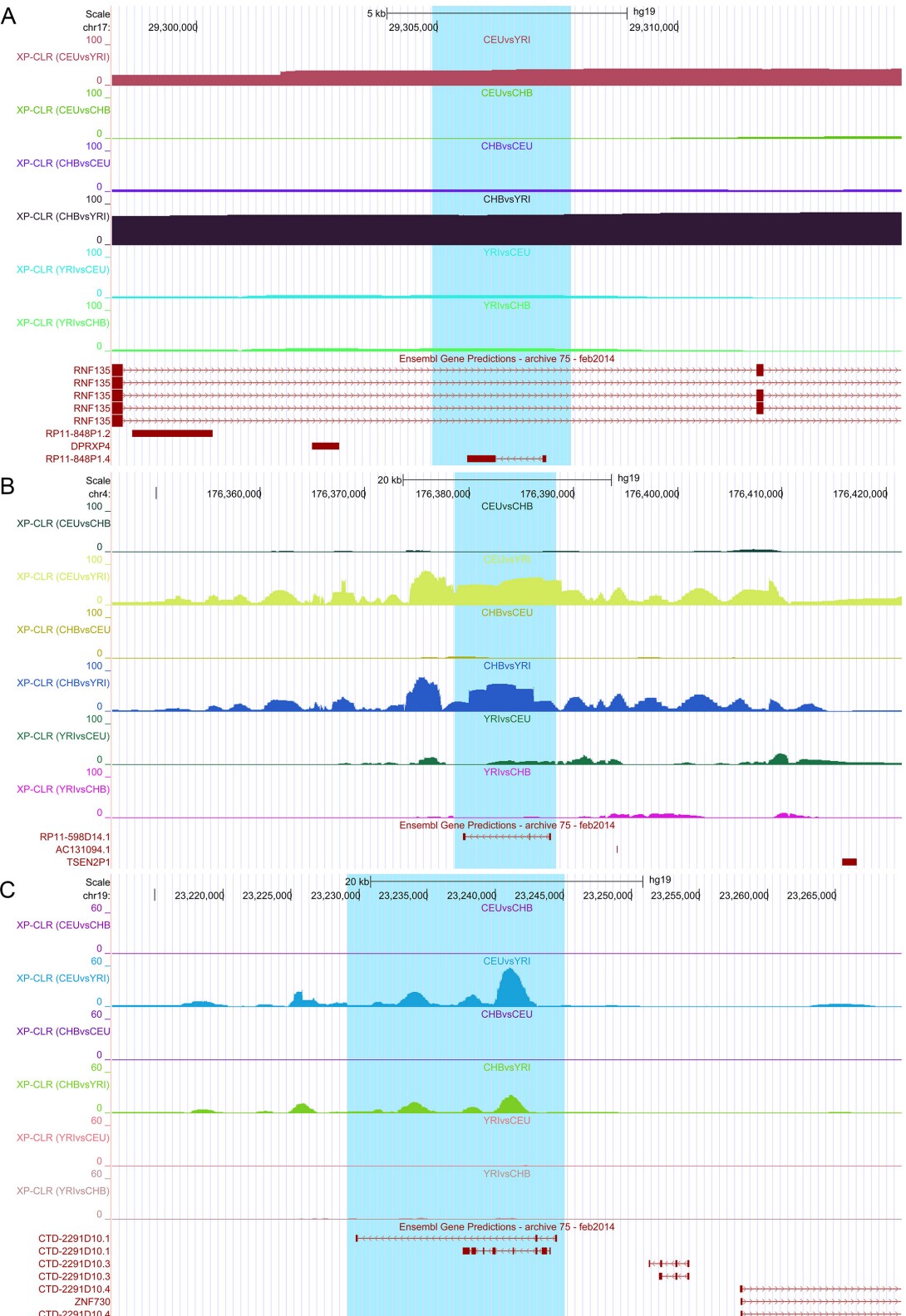

**Appendix 4—figure 1.** Positive selection signals detected by the XP-CLR program in (**A**) RP11-848P1.4, (**B**) RP11-598D14.1, and (**C**) CTD-2291D10.1 in CEU and CHB.

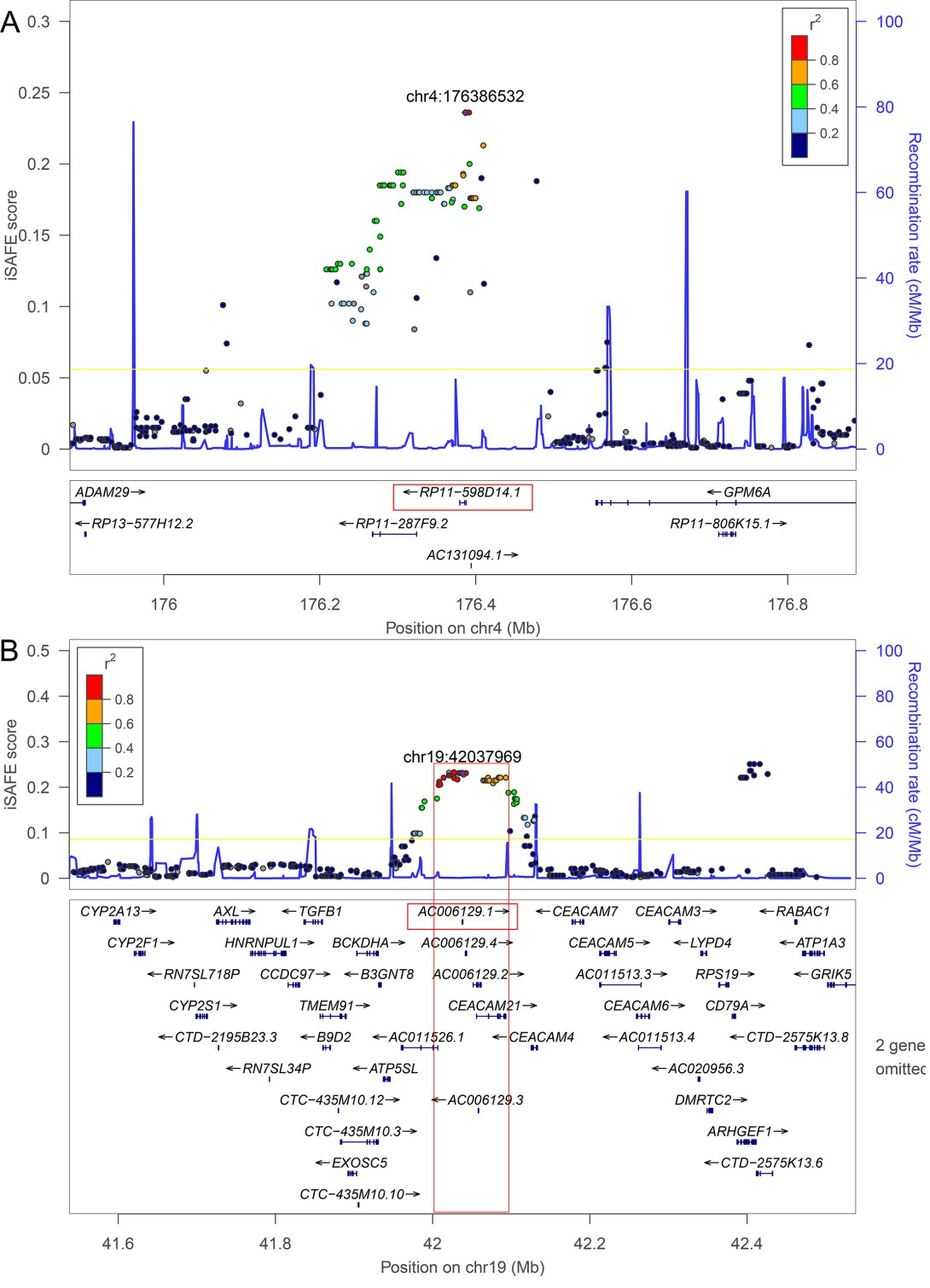

**Appendix 4—figure 2.** Favored mutations detected by iSAFE. Left and right vertical axes indicate iSAFE scores and recombination rate. The purple diamond marks the top-scored mutation. Colors mark linkage disequilibrium (LD) ($r^2$) between the top-scoring mutation and others. The yellow line indicates that mutations above it have a probability of p = 1e−6 to be neutral. The blue curve indicates the position-specific recombination rates. (**A**) SNPs in *RP11-598D14.1*. The top-scoring SNP has DAFs of 0.125/0.960/0.922 in YRI/CEU/CHB. (**B**) SNPs in *AC006129.1*. The top-scoring SNP has DAFs of 0.134/0.717/0.587 in YRI/CEU/CHB.

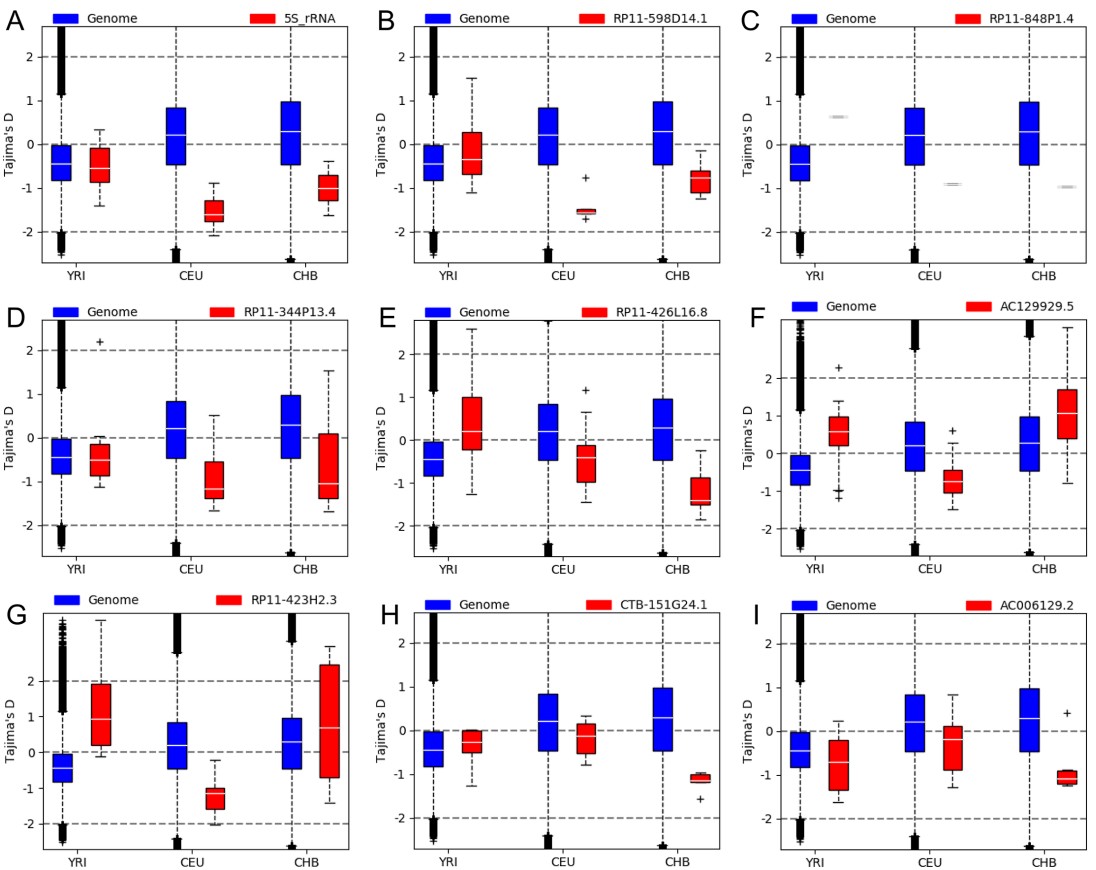

**Appendix 4—figure 3.** HS lncRNA genes with significantly changed Tajima's *D* in CEU, CHB, and YRI. Negative and positive Tajima's *D* scores, which are significantly smaller or larger than the genome-wide background in a population, indicate the signature of positive selection or balancing selection, respectively, in the population.

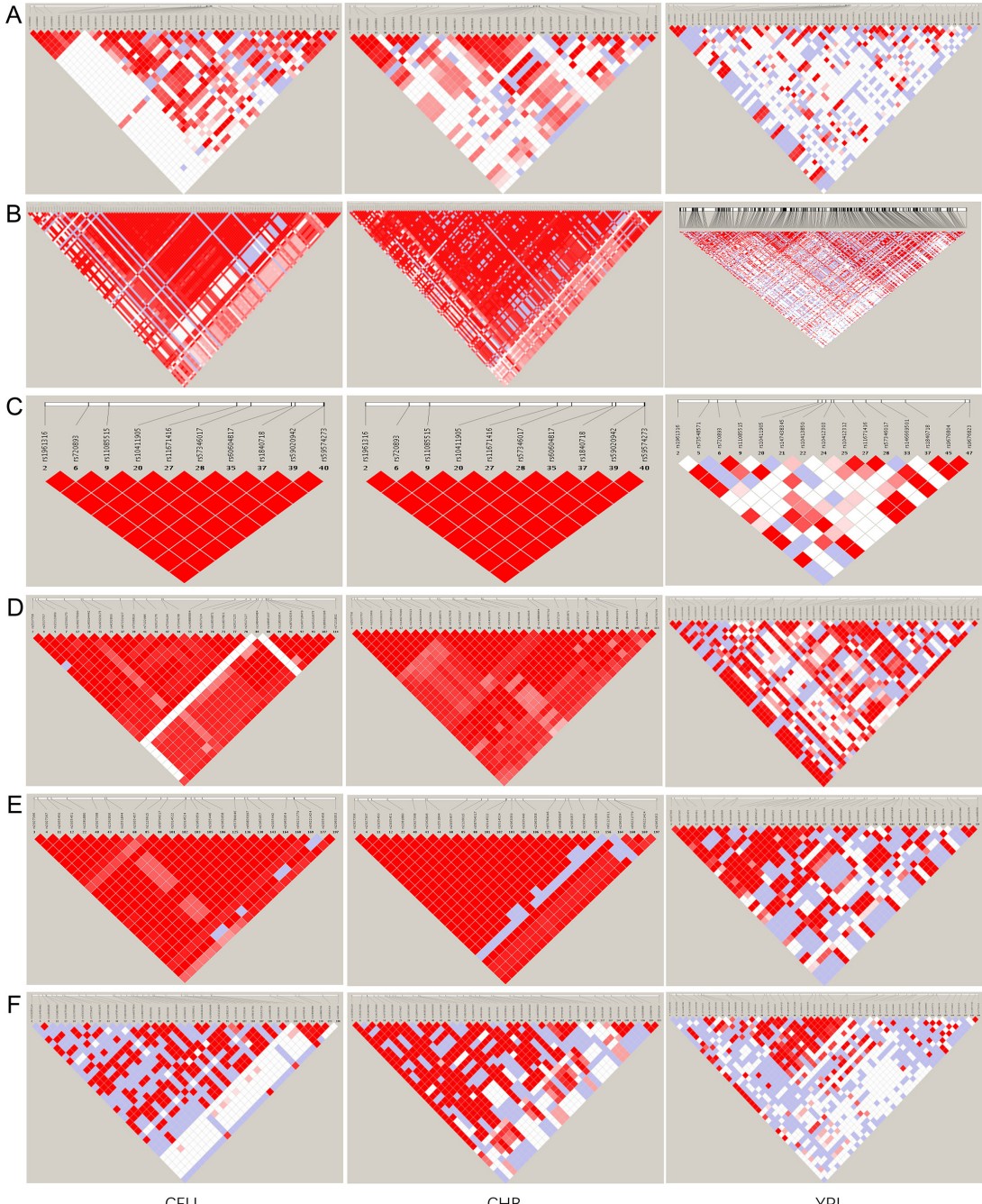

**Appendix 4—figure 4.** The linkage disequilibrium (LD) of SNPs in HS lncRNA genes in CEU, CHB, and YRI. The red color indicates high LD values. These panels show that LD between SNPs in CEU and CHB in these lncRNA genes is stronger than LD between SNPs in YRI. (**A**) AC024592.9, (**B**) AC129929.5, (**C**) RP11-157B13.7, (**D**) RP11-277P12.10, (**E**) CTD-2291D10.1, and (**F**) CTD-2142D14.1.

## Appendix 5

### Positive selection signals in DBSs

We used the tests mentioned above, including Tajima's *D* (**Tajima, 1989**), Fay and Wu's *H* (**Fay and Wu, 2000**), integrated Fst (**Weir and Cockerham, 1984**), and LD (**Slatkin, 2008**), to detect positive selection signals in DBSs in each population. These tests reveal positive selection signals in DBSs in specific target genes in specific populations (**Supplementary file 1I–L**).

First, we calculated Tajima's *D*, Fst, and integrated Fst for polymorphic DBSs in CEU, CHB, and YRI. Positive selection signals were detected in DBSs in more genes in CEU and CHB than in YRI. These genes include the two pigmentation-related genes *MC1R* and *MFSD12*, the two odor reception-related genes *OR6C1* and *TAS1R3*, and the immune-related gene *TLR1*. These target genes may suggest adaptations in gene expression regulation in CEU and CHB in response to changes in diet and the environment. In YRI, positive selection signals were detected in DBSs in genes such as *SLCO4A1*, which encodes a protein mediating the Na-independent transport of organic anions (e.g., the thyroid hormones T3 and T4). DBSs in different transcripts of *GNAS* (a gene important for genomic imprinting) contain SNPs selected explicitly in different populations (**Appendix 5—figure 1**).

Next, we calculated Fay-Wu's *H* and integrated Fst for each polymorphic DBS in CEU, CHB, and YRI. Strong negative values ($H < -2$), together with significantly large integrated Fst ($>0.22$), were obtained mainly in DBSs in CEU and CHB. These genes fall into two classes. The first class includes *PASK*, *CPT1A*, and *EXOC7*, which are important for glucose and lipid metabolism. *PASK* encodes a protein that plays a role in the regulation of insulin gene expression. *CPT1A* encodes a protein that exerts an important role in triglyceride metabolism. *EXOC7* encodes a protein that plays a crucial role in targeting SLC2A4 vesicles to the plasma membrane in response to insulin in adipocytes. The second class includes *COMT*, *TAS1R3*, and *ALMS1*, which are important for neural development. *COMT* is involved in the metabolism of adrenaline and noradrenaline. *TAS1R3* encodes a protein important for recognizing diverse natural and synthetic sweeteners. Mutations in *ALMS1* are associated with Alström syndrome.

Third, to examine whether gene expression regulation by HS lncRNAs is coordinated, we examined the LD between SNPs in DBSs of the same HS lncRNA chromosome-wide. When more than one SNP in a DBS has a minimal allele frequency (MAF) $\geq 0.05$, we chose the SNP that had the strongest LD ($r^2$). Despite genetic recombination, LD between DBSs was detected on some chromosomes in CEU and CHB (**Appendix 5—figure 2**).

Fourth, we computed the distances between DBSs in modern humans and their counterparts in archaic humans and compared them with the distances between annotated promoters and their counterparts in archaic humans. A considerable portion of DBSs (note that DBSs are within promoter regions) have larger distances than promoters between modern and archaic humans (**Appendix 5—figure 1**). We also computed the frequency distribution of SNPs ($>0.05$) in DBSs and found that SNPs are enriched at low and high frequencies (**Appendix 5—figure 2**), indicating positive selection.

Finally, we analyzed two experimental datasets. It was reported that the expression of a set of genes in T cell activation in response to pathogens shows significant variations in 348 healthy European, African, and Asian individuals (**Ye et al., 2014**). To examine whether HS lncRNAs contribute to these variations, we examined whether DBSs in these genes exhibit different binding affinities across populations. We found that DBSs in *IFITM3*, *IL2RA*, *IL17F*, *MXRA7*, *CCL22*, and *FADS2* contain SNPs with biased frequencies in CEU, CHB, and YRI. A typical example is the DBSs in *FADS2*, a gene expressed differentially in Europeans and Africans. The two DBSs in *FADS2-003* and *FADS2-010*, respectively, contain four SNPs – rs71046746 has DAF 0.96/0.99/0.88 in CEU/CHB/YRI, but the other three unannotated SNPs have DAFs 0.24/0.22/0.01, 0.24/0.22/0.04, and 0.24/0.22/0.02 in CEU/CHB/YRI.

Another study examined genome-wide patterns of selection in 230 West Eurasians who lived between 6500 and 300 BC and identified selection signals in multiple genes that are associated with diet, pigmentation, and immunity (**Mathieson et al., 2015**). We detected signatures of positive selection in DBSs in *LCT*, *TLR1*, *TLR6*, *TLR10*, *SLC45A2*, *SLC22A4*, *MHC*, *ZKSCAN3*, *FADS1*, *FADS2*, *DHCR7*, *GRM5*, *ATXN2*, and *HERC2*. Reliable population-specific selection signals were identified in the DBS of RP11-423H2.3 in *HERC2* (the Tajima's *D* in CEU/CHB/YRI are −0.19/1.82/−1.12 and

the integrated Fst is 0.27), in the DBSs of RP11-423H2.3 in *TLR1* and *TLR6* (the Tajima's *D* in CEU/CHB/YRI are −1.26/−1.2/1.38 and the integrated Fst is 0.24), and in the DBS of SNORA59B in *TLR1* and *TLR6* (the Tajima's *D* in CEU/CHB/YRI are −1.73/−0.9/1.76 and the integrated Fst is 0.24). Taken together, the above analyses suggest that population-specific selection signals in DBSs may help explain the phenotypic and physiological differences between human populations.

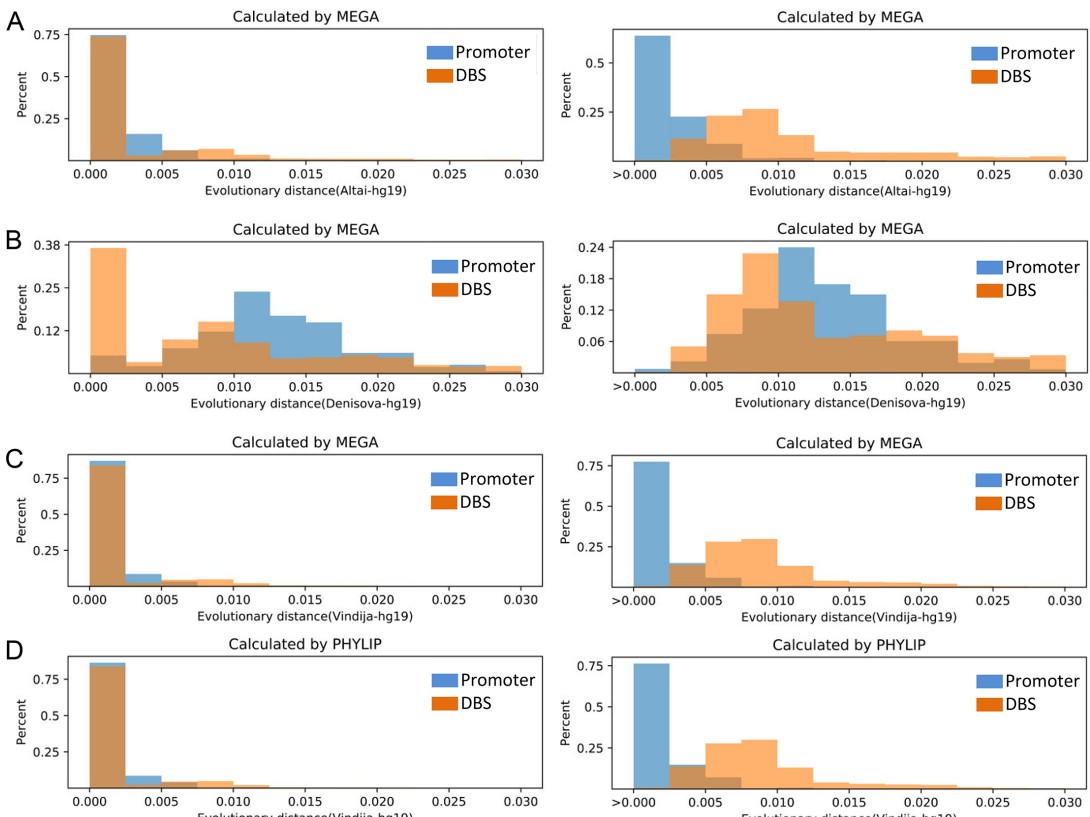

**Appendix 5—figure 1.** The distributions of DBS sequence distances and promoter sequence distances from modern to archaic humans (right-hand panels illustrating distances >0.005). A fraction of DBSs has larger distances than promoters.

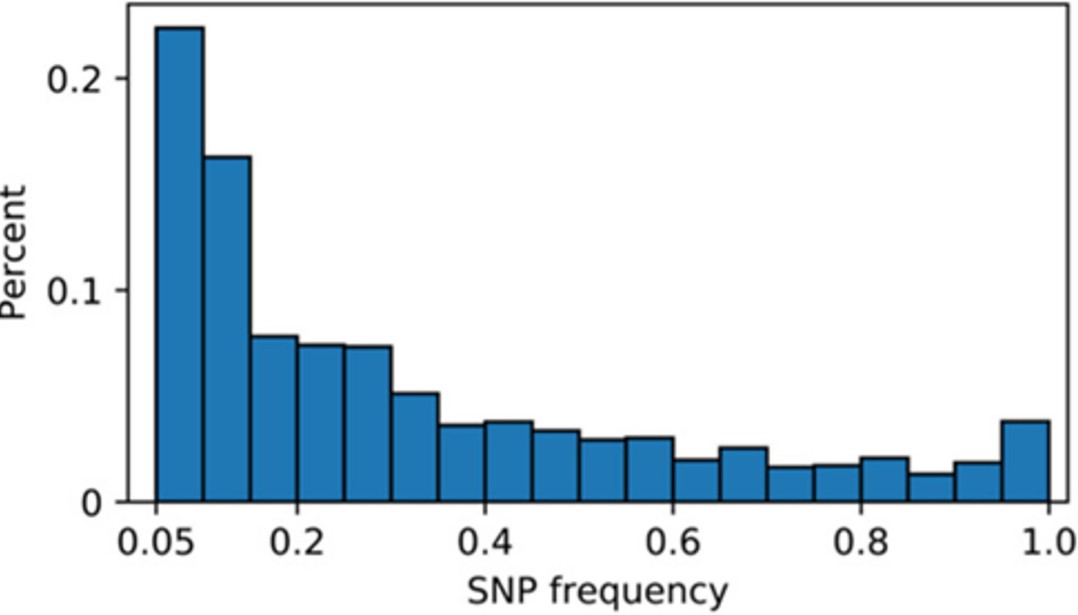

**Appendix 5—figure 2.** The distribution of SNP frequencies (MAF >0.05) in DBSs.

## Appendix 6

### Considerable SNPs in DBSs have a cis-effect on gene expression in specific tissues and populations

To find supporting evidence for DBS influencing gene expression in tissues and organs, we first examined whether SNPs in DBSs have a cis-effect on gene expression using the data of the Genotype-Tissue Expression (GTEx) project (*GTEx Consortium, 2017*). The expression of some genes in specific tissues is of interest. For example, of the 21 SNPs that are eQTLs exclusively in the GTEx tissue Thyroid, 14 have high DAF in YRI (*Appendix 6—table 1*). Correspondingly, in Africans, positive selection signals were found in genes involved in energy metabolism (*Fan et al., 2019*). Second, we computed the eQTL density for each DBS and each Ensembl-annotated promoter and compared the eQTL density of the two kinds of sequences. We found that eQTLs were more enriched in DBSs than in promoters (one-sided Mann–Whitney test, p = 0.0) (*Appendix 6—figure 1*). Third, for DBS harboring eQTLs in specific tissues, 94% of the corresponding HS lncRNA–target transcript pairs exhibit expression correlation (|Spearman's rho| >0.3 and FDR <0.05) in the eQTLs' tissues.

Finally, we examined how many SNPs DBSs are DNA methylation QTL (mQTL) and histone modification QTL (haQTL). One study analyzed DNA methylation data from three populations (10 Caucasians, 10 African Americans, and 10 Japanese) and identified *RGS6*, *CLEC2L*, *ABCF1*, *MIR3678*, and *EP400* as having differential methylation in African Americans compared with Japanese and Caucasians (*Giri et al., 2017*). Based on this study, we found that DBSs of several HS lncRNAs in some transcripts of *RGS6*, *CLEC2L*, and *EP400* (also *EP400NL*) had significantly different Tajima's *D* and integrated Fst in YRI. Another study applied QTL analysis to a multi-omics dataset and identified a set of SNPs significantly associated with levels of gene expression, DNA methylation, and histone modification in the human dorsolateral prefrontal cortex (*Ng et al., 2017*). Ng et al. called SNPs with cis-effect (including eQTL, mQTLs, and haQTLs) xQTL and found that xQTL SNPs were enriched close to transcription start sites. We re-examined these xQTL SNPs and identified 4735 in the DBSs of HS lncRNAs (*Supplementary file 1MN*). Notably, 4319 out of the 4735 xQTLs were meQTL SNPs, many of which had biased frequencies in CEU, CHB, and YRI.

**Appendix 6—table 1.** The 14 SNPs have high DAF in YRI and are eQTLs exclusively in the GTEx tissue Thyroid.

| SNP ID | CEU-frequency | CHB-frequency | YRI-frequency |
|---|---|---|---|
| rs75508216 | 0.01 | 0.05 | 0.1 |
| rs114086993 | 0.01 | 0.05 | 0.1 |
| rs201187971 | 0.01 | 0 | 0.1 |
| rs73677017 | 0 | 0 | 0.14 |
| rs11944829 | 0 | 0 | 0.14 |
| rs114884549 | 0 | 0 | 0.15 |
| rs77133472 | 0 | 0 | 0.15 |
| rs115688283 | 0 | 0 | 0.17 |
| rs113131895 | 0 | 0 | 0.17 |
| rs142522981 | 0 | 0.02 | 0.19 |
| rs112731299 | 0 | 0 | 0.2 |
| rs4565803 | 0.01 | 0 | 0.24 |
| rs4604779 | 0.01 | 0 | 0.24 |
| rs76612433 | 0.02 | 0.19 | 0.24 |

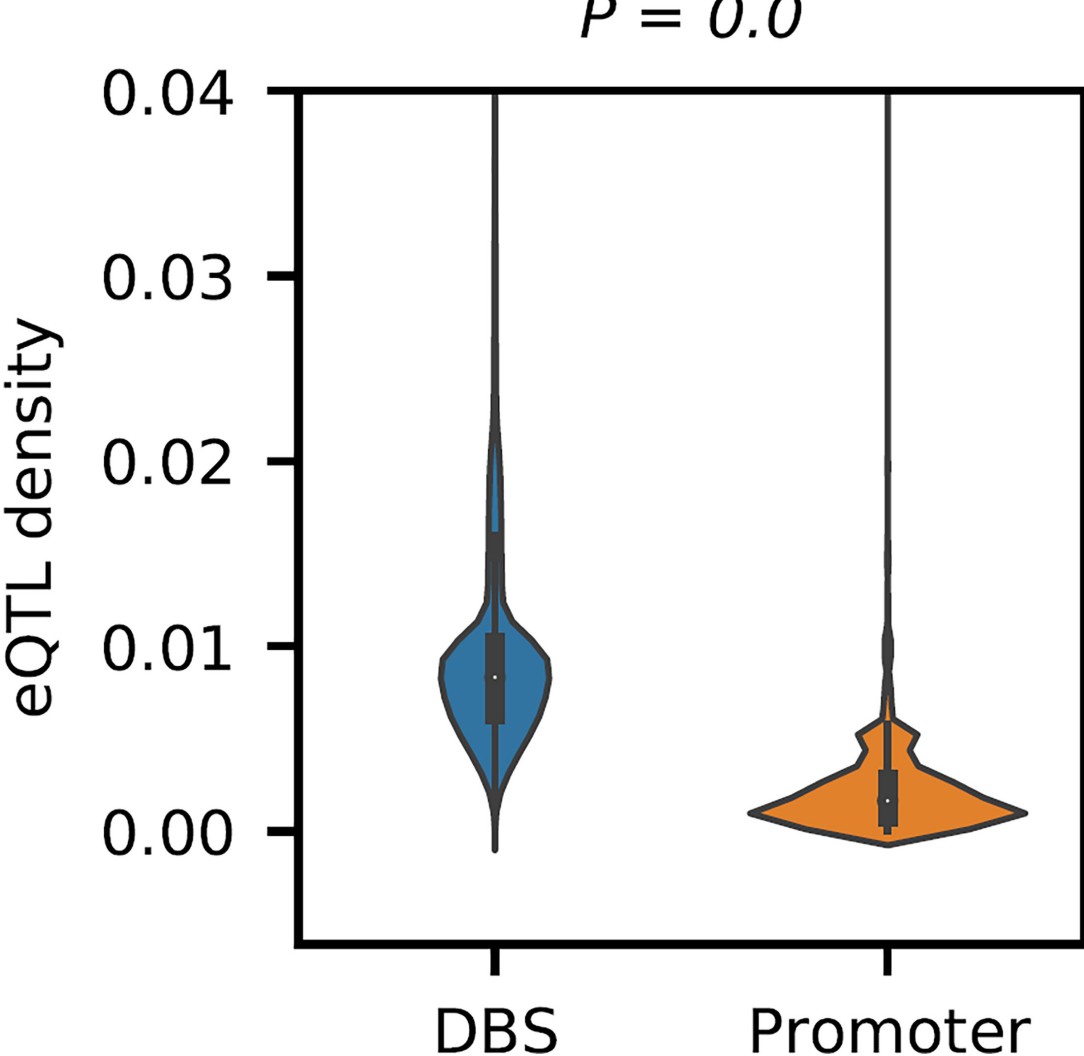

**Appendix 6—figure 1.** DBSs have significantly higher eQTL density than promoters. DBSs and promoters harboring at least one eQTL were used to compute eQTL density and make the comparison. A one-sided Mann–Whitney test was used to compute the p-value.

## Appendix 7

### HS lncRNAs critically regulate genes important for brain development

The enlarged brain is one of the most prominent features of modern humans. We analyzed multiple experimental datasets to examine the impact of HS lncRNAs on brain development. First, we analyzed two datasets of epigenetic studies. A comparative H3K27ac and H3K4me2 profiling of human, macaque, and mouse corticogenesis revealed that strongly concordant epigenetic gains are enriched in promoters of 301 genes that are active during human corticogenesis (*Reilly et al., 2015*). Of these genes (257 are annotated in hg38), 84% have DBSs for at least one HS lncRNA, and 56% have DBSs for ≥5 HS lncRNAs. Thus, these 301 genes are enriched significantly for regulation by HS lncRNAs compared with genes without DBSs of HS lncRNAs (p = 1.32e−26 for the situation of 56% genes, and p = 1.21e−21 for the situation of 84% genes, two-sided Fisher's exact test). Another study examined gene expression and H3K27ac modification in eight brain regions in humans and four other primates and revealed 1851 genes (1687 genes annotated in hg38) with human-specific transcriptome differences and 240 genes with chimpanzee-specific transcriptome differences in at least one brain region (*Xu et al., 2018*). Of these genes, 73% have DBSs for at least one HS lncRNA, and 46% have DBSs for ≥5 HS lncRNA. These data again indicate that genes related to brain development are significantly enriched in HS lncRNA targets (p = 1.1e−75 for the situation of 46% genes and p = 1.2e−56 for the situation of 73% genes, two-sided Fisher's exact test).

Second, we analyzed the data from two studies conducted by the PsychENCODE consortium. The data from one study cover 16 brain regions and 9 time windows and include spatiotemporally differentially expressed genes, spatiotemporally differentially methylated sites, spatiotemporal variations in H3K27ac enrichment, and cell-type marker genes (*Li et al., 2018*). Of the 66 HS lncRNAs, 65 were expressed in all of the 16 brain regions and in the 9 time windows, and 7 HS lncRNAs (RP11-423H2.3, RP11-706O15.5, RP13-539J13.1, SNORD3B-1, SNORD3B-2, RP4-740C4.7, and RP13-516M14.1) exhibited spatial or temporal expression differences. To assess the contribution of these 7 HS lncRNAs to the spatiotemporal changes of gene expression, we mapped the spatiotemporally differentially methylated sites to the promoters of annotated transcripts and identified 109 transcripts that had these sites in their promoters. We then predicted the DBSs of the 7 HS lncRNAs in the promoter regions of the 109 transcripts and found that 56 transcripts had DBSs for at least one of the 7 HS lncRNAs. In addition, 61% of 770 cell type-specific marker genes had at least one DBS of the s7 HS lncRNAs. These results suggest that HS lncRNAs regulate spatiotemporal gene expression and determine cell fate during brain development, probably by regulating DNA methylation in promoter regions. The other PsychENCODE study examined the spatiotemporal transcriptomic divergence across human and macaque brain development (*Zhu et al., 2018*). We examined the potential relationships between the HS lncRNAs and the 8951 genes showing differential expression between human and macaque brains during brain development. We identified DBSs of at least one HS lncRNA in the promoter regions of 72% of differentially expressed genes in the human brain. Moreover, DBSs of at least five HS lncRNAs were identified in the promoter regions of 44% of differentially expressed genes. Compared with genes without DBS for HS lncRNAs, these differentially expressed genes are significantly enriched for regulation by HS lncRNAs (p = 0, Chi-square test). Of the seven transcription factor genes differentially expressed between humans and macaques, four had DBSs of HS lncRNAs. These results support that gene expression in the human brain is highly regulated by HS lncRNAs.

Third, three recent studies identified several genes, including *NOTCH2NL* and *Aspm*, which are critical for regulating cortical expansion in the human brain (*Florio et al., 2018*; *Johnson et al., 2018*; *Suzuki et al., 2018*). *NOTCH2NL* is highly expressed in radial glia and activates Notch signaling to promote the clonal expansion of human cortical progenitors (*Florio et al., 2018*; *Suzuki et al., 2018*). *Aspm* is expressed in mice but shows no contribution to mouse corticogenesis. Nevertheless, an *Aspm* knockout greatly influenced corticogenesis in ferrets (*Johnson et al., 2018*). We examined whether HS lncRNAs potentially regulate these genes. Of the 40 protein-coding genes reported by the three studies, 14 have DBSs of ≥5 HS lncRNAs, and 29 have DBSs of ≥1 HS lncRNAs in their promoter regions (*Supplementary file 1O*). Compared with the background situation (22,562 protein-coding genes' promoter regions contain, but 20,944 protein-coding genes' promoter regions

do not contain, DBSs of HS lncRNAs), the 40 protein-coding genes involved in cortical expansion are highly enriched for regulation by HS lncRNAs (p < 0.01, two-sided Fisher's exact test).

Recently, brain organoids have emerged as an important approach to studying primate neural development in vitro. By establishing and comparing cerebral organoids between humans, chimpanzees, and macaques, *Pollen et al., 2019* identified 261 human-specific gene expression changes. In another study, *Agoglia et al., 2021* generated a panel of tetraploid human–chimpanzee hybrid iPS cells (i.e., hybrid induced pluripotent stem cells) by fusing human and chimpanzee iPS cells and differentiated the hybrid iPS cells into hybrid cortical spheroids. By allele-specific expression (ASE) analysis, Agoglia et al. identified thousands of genes with divergent expressions between humans and chimpanzees. We obtained the 261 genes from the first study and the 1102 genes with |logFC| >1 and adjusted p < 0.05 from the second study. Compared with the background situation mentioned above, the two sets of genes were significantly enriched with DBSs of HS lncRNAs (p = 1.2e−16 and 3.4e−74, respectively).

## Appendix 8

### The analysis of HS TFs and their DBSs

We lastly examined the contribution of HS TFs to gene expression in GTEx tissues and organs. *Kirilenko et al., 2023* recently identified orthologous genes in hundreds of placental mammals and birds, organized genes into pairwise datasets using humans and mice as the references (e.g., 'hg38-panTro6', 'hg38-mm10', and 'mm10-hg38'). In the hg38-panTro6 dataset, the many2zero and one2zero lists (which contain 0 and 147 genes, respectively) indicate the multiple human genes and the one human gene that have no orthologs in chimpanzees. Two studies and the *SCENIC* package reported three human TF lists (*Bahrami et al., 2015*; *Lambert et al., 2018*). Based on these data, we identified FOXO4, ZNF41, ZNF843, ZNF717, and TIMM8A as HS TFs (but note that not all of these orthologs are compatible with the Ensembl-annotated ones).

Multiple programs have been developed to predict TF DBSs, which has been a challenging task because TFs have complex 3D structures, TF DBSs are very short (typically around 10 bp), and TF-DNA binding involves diverse co-factors (*Bianchi et al., 2025*). Two popular programs are *FIMO* and *CellOracle* (*Grant et al., 2011*; *Kamimoto et al., 2023*). For each DBS, *FIMO* reports the start and end coordinates, but *CellOracle* reports just a position. First, for the above 5 potential HS TFs, we used *FIMO* and *CellOracle* to predict their DBSs in the 5000 bp promoter regions of the 179,128 Ensembl-annotated transcripts (release 79). Since TF DBSs predicted by *FIMO* have been incorporated into the JASPAR database (*Rauluseviciute et al., 2024*), we directly extracted the related DBSs from the 'JASPAR Transcription Factor' track in the UCSC Genome Browser (GRCh38/hg38) with the default cutoff. For *CellOracle*, we used the author-recommended score cutoff of 10 and extended 5 bp on both sides from each reported position. The intersection of *FIMO* and *CellOracle* results includes 171208 DBSs of FOXO4 and ZNF41 (*Supplementary file 1P*). Second, we identified counterparts of HS TF DBSs in archaic humans and chimpanzees and computed sequence distances of these DBSs from modern humans to archaic humans and chimpanzees (as we did for HS lncRNA DBS analyses). A small portion of HS TF DBSs lacks chimpanzee counterparts, and we assumed their distance = 10 (as we did for HS lncRNA DBSs). Since detecting selection signals in sequences as short as 10 bp is challenging, we simply used the DeepFavored system to scan HS lncRNA DBSs and HS TF DBSs and compared the results (*Tang et al., 2022*). The number of favored mutations is about 4 times higher in HS lncRNA DBSs than in HS TF DBSs (the number/the total length). Third, we computed the Pearson correlation between HS TFs and their target transcripts in GTEx tissues to identify correlated expression (|Spearman rho| >0.3 and FDR <0.05). Of all HS TF–target transcript pairs, 95% show correlated expression in at least one tissue. However, unlike the high percentages of correlated HS lncRNA–target transcript pairs in the brain, correlated HS TF–target transcript pairs are distributed across many tissues and organs (*Appendix 8—figure 1*). Finally, for each DBS in an HS TF–target transcript pair that shows correlated expression in a GTEx tissue, we computed the sequence distances of this DBS from modern humans to the three archaic humans, and then compared the distribution of DBS sequence distances in each tissue with the distribution of DBS sequence distances in all tissues (one-sided two-sample Kolmogorov–Smirnov test). Unlike DBSs in HS lncRNA–target transcript pairs (*Figure 3*), the significantly changed DBSs (in terms of sequence distance) in HS TF–target transcript pairs across GTEx tissues and organs do not show dense distribution in the brain (*Appendix 8—figure 2*). Taken together, these results suggest that HS lncRNAs may have contributed more significantly to human evolution than HS TFs by regulating gene expression.

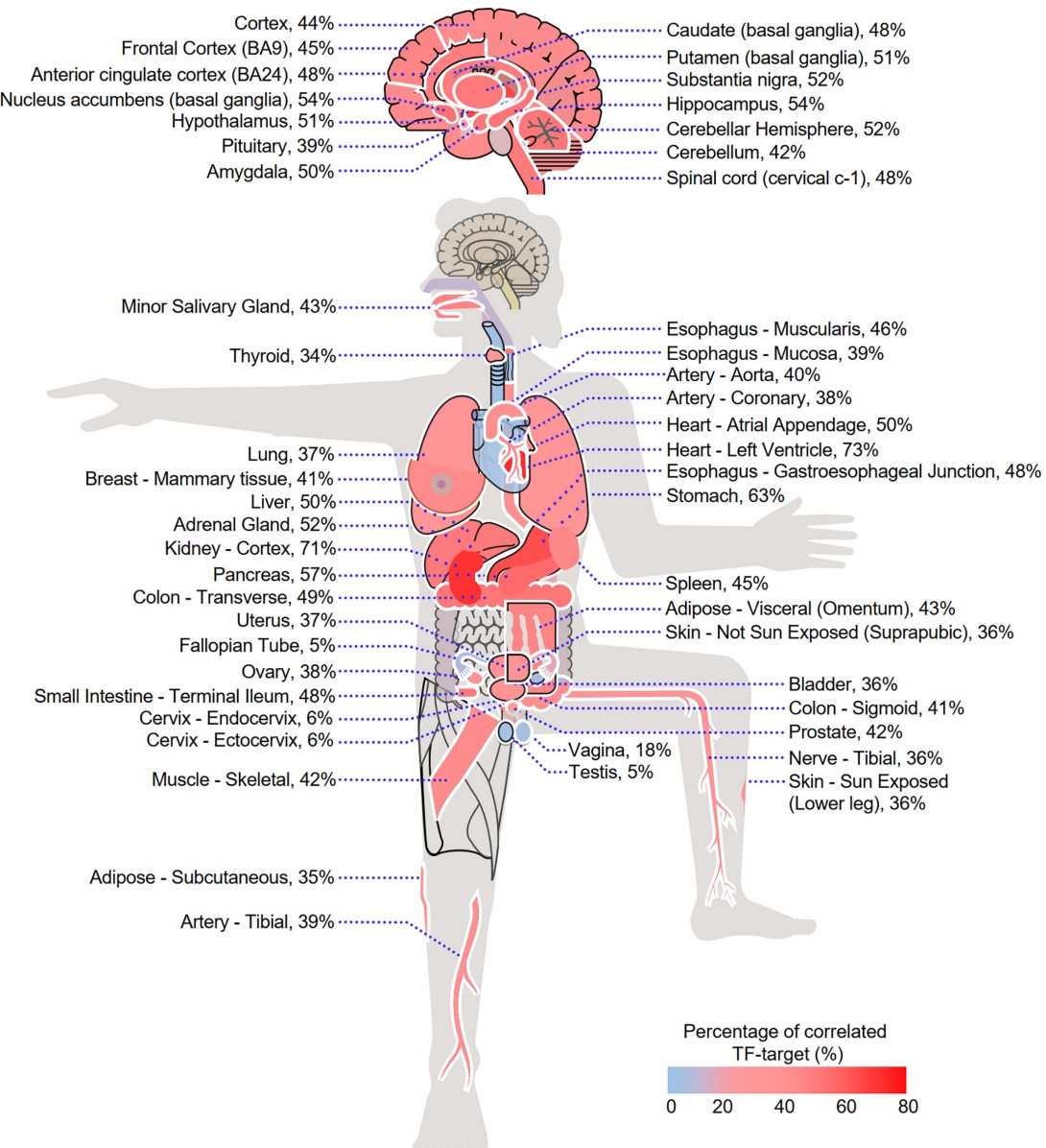

**Appendix 8—figure 1.** The distribution of the percentage of HS TF–target transcript pairs with correlated expression across GTEx tissues and organs (see *Figure 3A*).

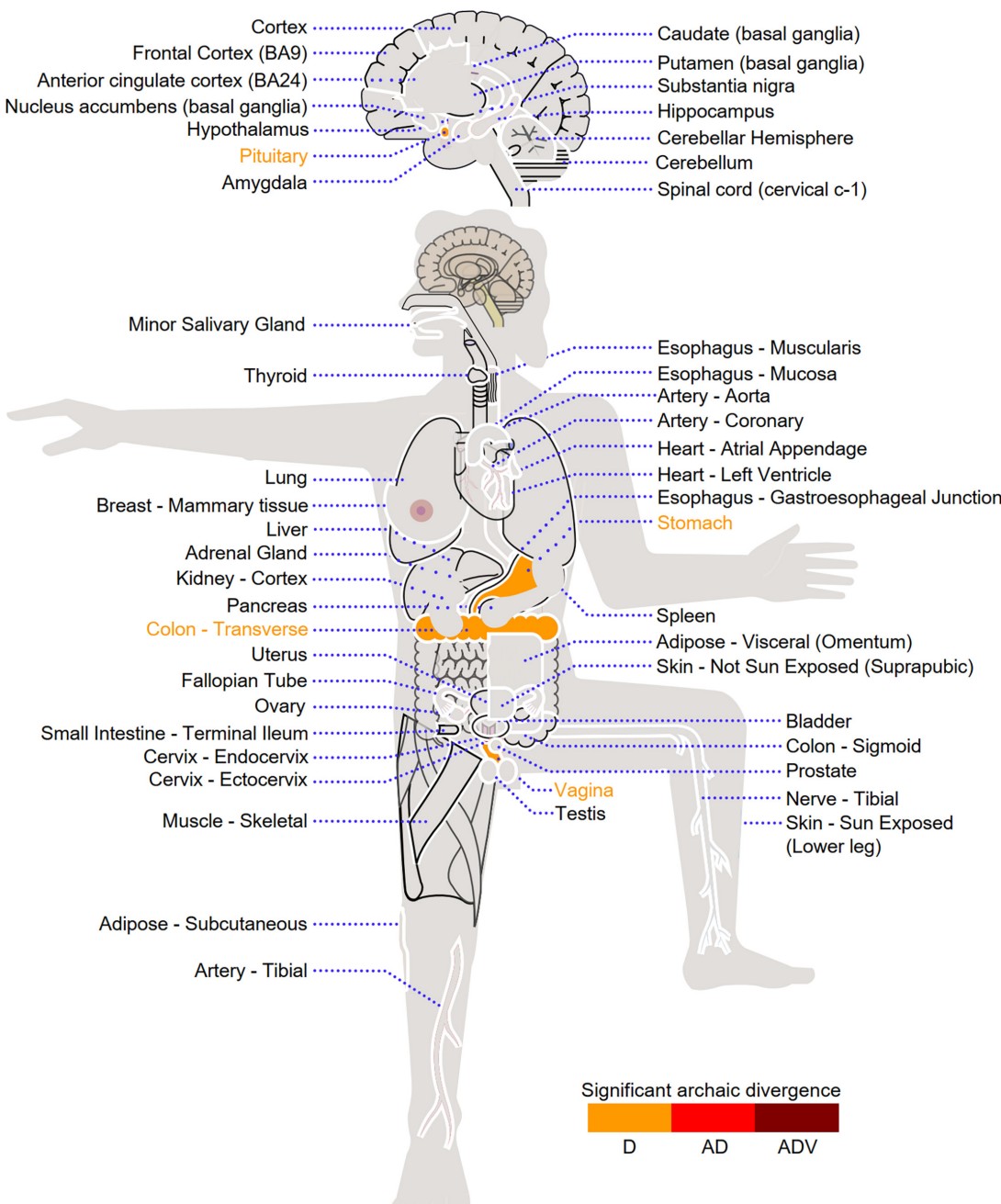

**Appendix 8—figure 2.** The distribution of significantly changed DBSs (in terms of sequence distance) in HS TF–target transcript pairs across GTEx tissues and organs between archaic and modern humans. As in *Figure 3B*, orange, red, and dark red indicate significant changes from Denisovans (**D**), Altai Neanderthals and Denisovans (**AD**), and all three archaic humans (**ADV**).

## Appendix 9

### Human-specifically rewiring of gene expression in the brain

Human-specific rewiring of gene expression should result in distinct correlations. To identify whether the pattern exists in the brain, we analyzed the transcriptomic data from two brain regions – frontal cortex (BA9) and anterior cingulate cortex (BA24) – in the human and macaque brain (*GTEx Consortium, 2017*; *Zhu et al., 2018*) ($n$ = 101 and 83, and $n$ = 22 and 25, respectively). We used the *eGRAM* to identify modules of genes with expression correlation and regulatory relationship (*Appendix 9—figure 1*). Orthologous genes (displayed at the same positions in each panel) show much more correlations in humans than in macaques and are significantly enriched for genes in neurodevelopment-related KEGG pathways. This result supports that gene expression is substantially rewired in the human brain by HS lncRNAs compared with in the macaque brain, and the correlations and modules provide novel information for interpreting the mechanisms of human cognition and behavior.

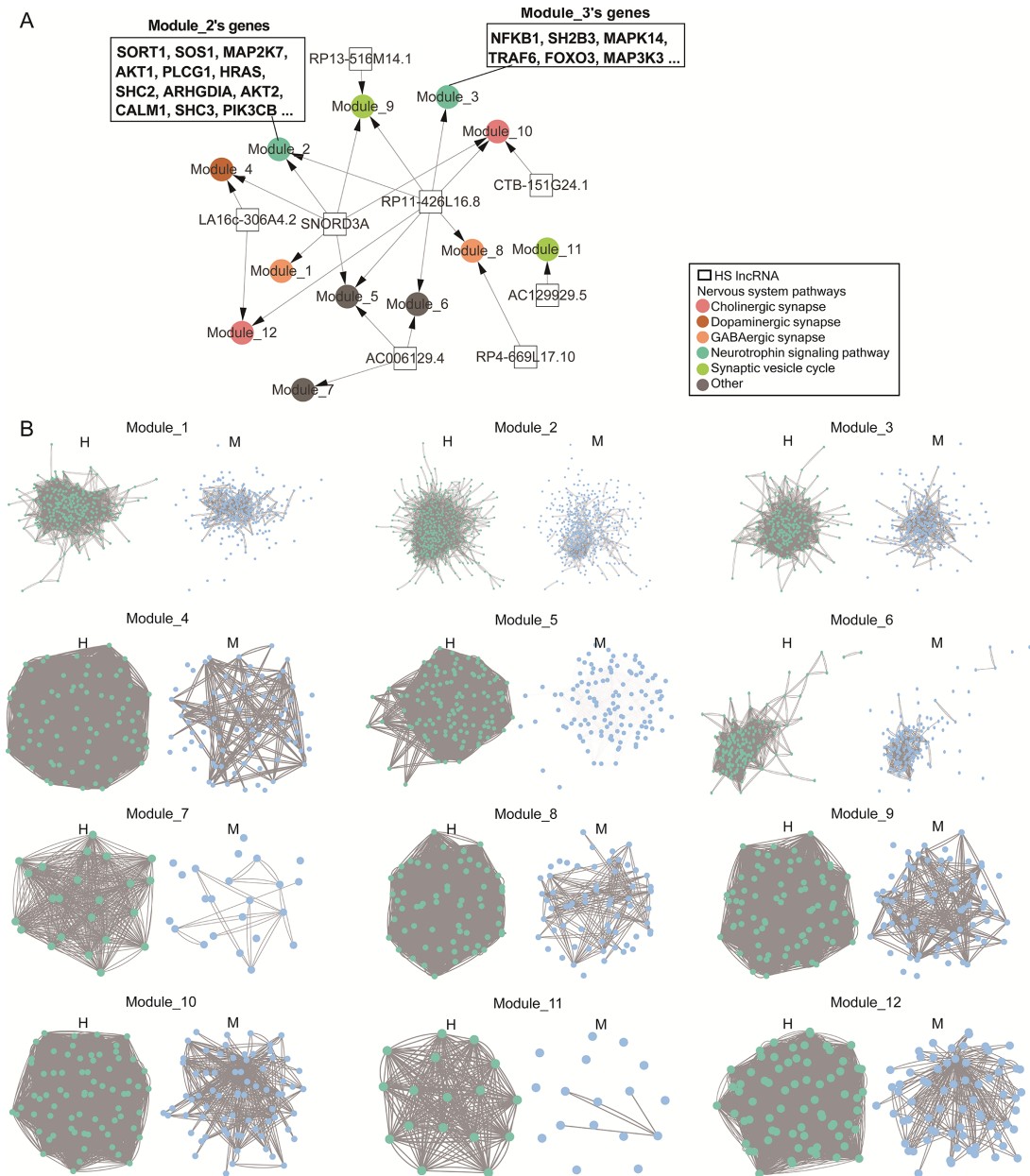

**Appendix 9—figure 1.** Human-specifically rewired gene expression by HS lncRNAs in the anterior cingulate cortex (BA24). (**A**) Genes expressed in the anterior cingulate cortex are enriched for HS lncRNAs' target genes and neurodevelopment-related pathways. Squares, dots, and colors indicate HS lncRNAs, gene modules, and enriched KEGG pathways, respectively. (**B**) Comparison of modules and genes in humans (indicated by H) and macaques (indicated by M).

