## [Editor Report · eLife Assessment]

This **valuable** study uses tools of population and functional genomics to examine long non-coding RNAs (lncRNAs) in the context of human evolution. Analyses of computationally predicted human-specific lncRNAs and their genomic targets lead to the development of hypotheses regarding the potential roles of these genetic elements in human biology. Compared to previous versions, the conclusions regarding evolutionary acceleration and adaptation have become more **solid** by more fully taking data and literature on human/chimpanzee genetics and functional genomics into account.

---

## [Referee Report · Joint Public Review]

While DNA sequence divergence, differential expression and differential methylation analysis have been conducted between humans and the great apes to study changes that "make us human", the role of lncRNAs and their impact on the human genome and biology has not been fully explored. In this study the authors computationally predict HSlncRNAs as well as their DNA Binding sites using a method they have developed previously and then examine these predicted regions with different types of enrichment analyses. Broadly the analysis are straightforward and after identifying these regions/HSlncRNAs they examined their effects using different external datasets.

Comments on the latest version from Reviewer #2:

I think this is as good as it is going to get, and I do appreciate that the authors are still engaging in good faith after all these rounds of revision, so I am happy to stop here! I do think the paper is significantly improved from the last time around, and the conclusions have been tempered significantly.

---

## [Author Response]

The following is the authors’ response to the previous reviews

**Public Reviews:**

**Reviewer #2 (Public review):**
In this valuable manuscript, Lin et al attempt to examine the role of long non coding RNAs (lncRNAs) in human evolution, through a set of population genetics and functional genomics analyses that leverage existing datasets and tools. Although the methods are incomplete and at times inadequate, the results nonetheless point towards a possible contribution of long non coding RNAs to shaping humans, and suggest clear directions for future, more rigorous study.Comments on revisions:I thank the authors for their revision and changes in response to previous rounds of comments. As before, I appreciate the changes made in response to my comments, and I think everyone is approaching this in the spirit of arriving at the best possible manuscript, but we still have some deep disagreements on the nature of the relevant statistical approach and defining adequate controls. I highlight a couple of places that I think are particularly relevant, but note that given the authors disagree with my interpretation, they should feel free to not respond!(1) On the subject of the 0.034 threshold, I had previously stated: "I do not agree with the rationale for this claim, and do not agree that it supports the cutoff of 0.034 used below."In their reply to me, the authors state:"What we need is a gene number, which (a) indicates genes that effectively differentiate humans from chimpanzees, (b) can be used to set a DBS sequence distance cutoff. Since this study is the first to systematically examine DBSs in humans and chimpanzees, we must estimate this gene number based on studies that identify differentially expressed genes in humans and chimpanzees. We choose Song et al. 2021 (Song et al. Genetic studies of human-chimpanzee divergence using stem cell fusions. PNAS 2021), which identified 5984 differentially expressed genes, including 4377 genes whose differential expression is due to trans-acting differences between humans and chimpanzees. To the best of our knowledge, this is the only published data on trans-acting differences between humans and chimpanzees, and most HS lncRNAs and their DBSs/targets have trans-acting relationships (see Supplementary Table 2). Based on these numbers, we chose a DBS sequence distance cutoff of 0.034, which corresponds to 4248 genes (the top 20%), slightly fewer than 4377."I have some notes here. First, Agoglia et al, Nature, 2021, also examined the nature of cis vs trans regulatory differences between human and chimps using a very similar set up to Song et al; their Supplementary Table 4 enables the discovery of genes with cis vs trans effects although admittedly this is less straightforward than the Song et al data. Second, I can't actually tell how the 4377 number is arrived at. From Song et al, "Of 4,671 genes with regulatory changes between human-only and chimpanzee-only iPSC lines, 44.4% (2,073 genes) were regulated primarily in cis, 31.4% (1,465 genes) were regulated primarily in trans, and the remaining 1,133 genes were regulated both in cis and in trans (Fig. 2C). This final category was further broken down into a cis+trans category (cis- and transregulatory changes acting in the same direction) and a cis-trans category (cis- and trans-regulatory changes acting in opposite directions)." Even when combining trans-only and cis&trans genes that gives 2,598 genes with evidence for some trans regulation. I cannot find 4,377 in the main text of the Song et al paper.Elsewhere in their response, the authors respond to my comment that 0.034 is an arbitrary threshold by repeating the analyses using a cutoff of 0.035. I appreciate the sentiment here, but I would not expect this to make any great difference, given how similar those numbers are! A better approach, and what I had in mind when I mentioned this, would be to test multiple thresholds, ranging from, eg,0.05 to 0.01 <DBS dist = 0.01 -> 0.034 -> 0.05> at some well-defined step size.

(1) We sincerely thank the reviewer for this critical point. Our initial purpose, based on DBS distances from the human genome to chimpanzee genome and archaic genomes, was that genes with large DBS distances may have contributed more to human evolution. However, our ORA (overrepresentation analysis) explored only genes with large DBS distances (the legend of old Figure 2 was “1256 target genes whose DBSs have the largest distances from modern humans to chimpanzees and Altai Neanderthals are enriched in different Biological Processes GO terms”), with the use of the cutoff (threshold) of 0.034 for defining large distance. The cutoff is not totally unreasonable (as our new results and the following sensitivity analysis indicate), but this approach was indirect and flawed.

(2) We have now performed ORA using two methods. The first uses only DBS distances. Instead of using a cutoff, we now sort genes by DBS distance (human-chimpanzee distances and human-Altai Neanderthal distance, respectively, see Supplementary Table 5) and use the top 25% and bottom 25% of genes to perform ORA. This directly examines whether DBS distances along indicate that genes with large DBS distances contribute more to human evolution than genes with small DBS distances. The second also explores the ASE genes (allele-specific expression, genes undergoing human/chimpanzee-specific regulation in the tetraploid human–chimpanzee hybrid iPS) reported by Agoglia et al. 2021. We select the top 50% and bottom 50% of genes with large and small DBS distances, intersect them with ASE genes from Agoglia et al. 2021 (their Supplementary Table 4), and apply ORA to the intersections. Both the results are that: (a) more GO terms are obtained from genes with large DBS distances, (b) more human evolution-related GO terms are obtained from genes with large DBS distances (Supplementary Table 5,6,7; Figure 2; Supplementary Fig. 15). These results directly suggest that genes with large DBS distances contribute more to human evolution than genes with small DBS distances, which is a key theme of the study.

(3) Regarding Song et al 2021, the statement of “we differentiated…allotetraploid (H1C1a, H1C1b, H2C2a, H2C2b) lines into ectoderm, mesoderm, and endoderm” made us assume that their differentiated hybrid cell lines cover more tissue types than those of Agoglia et al. 2021. Now, upon re-examining Supplementary Table 5 of Song et al. and Supplementary Table 4 of Agoglia et al. 2021, we find that the latter more clearly indicates significant ASE genes (p-adj<0.01 and |LFC>0.5| in GRCh38 and PanTro5).

(4) We have also performed two additional analyses in response to the suggestion of “test multiple thresholds, ranging from, eg, 0.05 to 0.01 <DBS dist = 0.01 -> 0.034 -> 0.05> at some well-defined step size”. First, we performed a multi-threshold sensitivity analysis using a spectrum of cutoffs (0.03, 0.034, 0.04, 0.05), and tracked the number of genes identified and the enrichment significance of key GO terms (e.g., "neuron projection development," "behavior") across these thresholds. The result confirms that while the absolute number of genes varies with the cutoffs, the core biological conclusion (specifically, the significant enrichment of target genes in neurodevelopmental and cognitive functions) remains stable and significant. For instance, "behavior" maintains strong statistical significance (FDR<0.01) in both the human-chimpanzee and human-Altai Neanderthal comparisons across all tested cutoffs, and "Neuron projection development" also remains significant across three (0.03, 0.034, 0.04) of the four cutoffs in the Altai comparison. This pattern suggests that our core findings regarding neurodevelopmental functions are robust across a range of cutoffs. Nevertheless, we did not extend the analysis to smaller cutoffs (e.g., 0.01 or 0.02) because such values would identify an excessively large number of genes (>10000) for ORA, which would render the GOterm enrichment analysis less meaningful due to a loss of specificity.

Second, we have performed an additional validation to directly evaluate whether the 0.034 cutoff itself represents a stringent and biologically meaningful value. We sought to empirically determine how often a DBS sequence distance of 0.034 or greater might occur by chance in promoter regions, thereby testing its significance as a marker of potential evolutionary divergence. We randomly sampled 10,000 windows from annotated promoter regions across the hg38 genome, each with a size matching the average length of DBSs (147 bp). We then calculated the per-base sequence distances for these random windows between modern humans and chimpanzees, as well as between modern humans and the three archaic humans (Altai, Denisovan, Vindija). The analysis reveals that a distance of ≥0.034 is a rare event in random promoter sequences: for Human-Chimp, Human-Altai, HumanDenisovan, and Human-Vindija, 5.49% (549/10000), 0.31% (31/10000), 4.47% (447/10000), and0.03% (3/10000) of random windows reach this distance. This empirical evidence suggests that 0.034 is a sufficiently strong cutoff for defining large DBS distance, it would occur very unlikely in a random genomic background (P<0.1 for Chimpanzee and P<0.05 for the archaic humans), and DBSs exceeding this cutoff are significantly enriched for sequences that have undergone substantial evolutionary change instead of being random neutral variations.

(5) We present new Figure 2, Supplementary Table 5,6,7, and Supplementary Fig. 15. We have substantially revised section 2.3, related sections in Results, Supplementary Note 3, and Supplementary Table 8. We have removed related descriptions and explanations in the main text and Supplementary Notes. The results of the above two analyses are presented here as two Author response images.

**Author response table 1. sa2table1:** Sensitivity analysis of GO-term enrichment across different DBS sequence distance cutoffs. The table shows the numbers of target genes identified and the false discovery rates (FDR) for the enrichment of three selected GO terms at four different distance cutoffs. Note that, unlike in the old Figure 2, the results for chimpanzees and Altai Neanderthals are not directly comparable here, as the numbers of target genes used for the enrichment analysis differ between them at each cutoff.

	Cutoff = 0.03		Cutoff = 0.034		Cutoff = 0.04		Cutoff = 0.05	
	Chimp	Altai	Chimp	Altai	Chimp	Altai	Chimp	Altai
Target gene with distance > cutoff	7087	1817	4248	1256	3789	1036	3223	745
Behavior (FDR)	7.06E-05	8.56E-03	7.10E-07	1.32E-06	2.31E-05	4.65E-05	5.17E-05	0.00741
Neuron projection development (FDR)	2.91E-05	1.77 E -02	1.41 E -08	9.91E-05	4.52E-07	0.01887	4.22E-05	NS
Synaptic signaling (FDR)	1.86E-08	4.99E-03	4.60E-07	2.34E-05	4.32 E -07	6.31 E -05	9.11E-07	0.001891

**Author response image 1. sa2fig1:** Distribution of per-base sequence distances for DBS size-matched random genomic windows in Ensembl-annotated promoter regions, calculated between modern humans and (A) chimpanzee, (B) Altai Neanderthal, (C) Denisovan, and (D) Vindija Neanderthal genomes.

(2) The authors have introduced a new TFBS section, as a control for their lncRNAs - this is welcome, though again I would ask for caution when interpreting results. For instance, in their reply to me the authors state: "The number of HS TFs and HS lncRNAs (5 vs 66) alone lends strong evidence suggesting that HS lncRNAs have contributed more significantly to human evolution than HS TFs (note that 5 is the union of three intersections between <many2zero + one2zero> and the three)."But this assumes the denominator is the same! There are 35899 lncRNAs according to the current GENCOVE build; 66/35899 = 0.0018, so, 0.18% of lncRNAs are HS. The authors compare this to 5 TFs. There are 19433 protein coding genes in the current GENCOVE build, which naively (5/19433) gives a big depletion (0.026%) relative to the lnc number. However, this assumes all protein coding genes are TFs, which is not the case. A quick search suggests that ~2000 protein coding genes are TFs (see, eg, https://pubmed.ncbi.nlm.nih.gov/34755879/); which gives an enrichment (although I doubt it is a statistically significant one!) of HS TFs over HS lncRNAs (5/2000 = 0.0025). Hence my emphasis on needing to be sure the controls are robust and valid throughout!

We thank the reviewer for this comment. While 5 vs 66 reveals a difference, a direct comparison is too simplified. The real take-home message of the new TFBS section is not the numbers but the distributions of HS TFs’ targets and HS lncRNAs’ targets across GTEx organs and tissues (Figure 3 and Supplementary Figures 24, 25) - correlated HS lncRNA-target transcript pairs are highly enriched in brain regions, but correlated HS TF-target transcript pairs are distributed broadly across GTEx tissues and organs. We have now removed the simple comparison of “5 vs 66” and more carefully explained our comparison in section 2.6.

(3) In my original review I said: line 187: "Notably, 97.81% of the 105141 strong DBSs have counterparts in chimpanzees, suggesting that these DBSs are similar to HARs in evolution and have undergone human-specific evolution." I do not see any support for the inference here. Identifying HARs and acceleration relies on a far more thorough methodology than what's being presented here. Even generously, pairwise comparison between two taxa only cannot polarise the direction of differences; inferring human-specific change requires outgroups beyond chimpanzee.In their reply to me, the authors state:Here, we actually made an analogy but not an inference; therefore, we used such words as "suggesting" and "similar" instead of using more confirmatory words. We have revised the latter half sentence, saying "raising the possibility that these sequences have evolved considerably during human evolution".Is the aim here to draw attention to the ~2.2% of DBS that do not have a counterpart? In that case, it would be better to rewrite the sentence to emphasise those, not the ones that are shared between the two species? I do appreciate the revised wording, though.

(1) Our original phrasing may be misleading, and we agree entirely that “pairwise comparison between two taxa only cannot polarise the direction of differences; inferring human-specific change requires outgroups beyond chimpanzee”. As explained in that reply, we know and think that DBSs and HARs are two different classes of sequences, and indeed, identifying HARs and acceleration relies on a far more thorough methodology. Yet, three factors prompted us to compare them. First, both suggest the importance of sequences outside genes. Second, both are quite “old” sequences and have undergone considerable evolution recently (although the references are different). Third, both have contributed greatly to human brain evolution.

(2) Here, our stress is 97.81% but not 2.2%, and we have made this analogy more clearly and cautiously. Relevant revisions have been made in the Results, Discussion, and Methods sections.

(3) We also have further determined whether the 2.2% DBSs are human-specific gains by analyzing them using the UCSC Multiz Alignments of 100 Vertebrates. The result confirms that all 2248 DBSs are present in the human genome but are absent from the chimpanzee genome and all other aligned vertebrate genomes. We add this result into the manuscript.

(4) Finally, Line 408: "Ensembl-annotated transcripts (release 79)" Release 79 is dated to March 2015, which is quite a few releases and genome builds ago. Is this a typo? Both the human and the chimpanzee genome have been significantly improved since then!

(1) We thank the reviewer for this comment, which prompts us to provide further explanation and additional data. First, we began predicting HS lncRNAs’ DBSs when Ensembl release 79 was available, but did not re-predict DBSs when new Ensembl releases were published because (a) these new Ensembl releases are based also on hg38, (b) we did not find any fault in the LongTarget program during our use, nor received any one from users, (c) predicting lncRNAs’ DBSs using the LongTarget program is highly time-consuming.

(2) Second, to assess the influence of newer Ensembl releases, we compared the promoters annotated in release 79 and in release 115. We found that the vast majority (87.3%) of promoters newly annotated in release 115 belong to non-coding genes. Thus, using release 115 may predict more DBSs in non-coding genes, but downstream analyses based on protein-coding genes would be essentially the same (meaning that all figures and tables would be the same).

(3) Third, a key element of this study is GTEx data analysis, and these data were also published years ago.

(4) Finally, some lncRNA genes have new gene symbols in new Ensembl releases. To allow researchers to use our data conveniently, we have added a new column titled "Gene symbol (Ensembl release115)" to Supplementary Tables 2A and 2B.

Summary:

Major changes based on Reviewer’s comments:

(1) The following revisions are made to address the comment on “the 0.034 threshold”: (a) Section 2.3, section 2.4, Supplementary Note 3, and related contents in Discussion and Methods are revised, (b) new Figure 2, Supplementary Figure 15, new Supplementary Table 5,6,7, (c) Table 2 and Supplementary Table 8 are revised.

(2) To address the comment on “new TFBS section”, section 2.6 and section 4.13 are revised.

(3) To address the comment on “97.81% and 2.2% of DBSs”, section 2.3 is revised.

(4) The following revisions are made to address the comment on “release 79”: (a) the old Supplementary Table 2, 3 are merged to Supplementary Table 2AB, and the new column "Gene symbol (Ensembl release115)" is added to Supplementary Table 2AB, (b) accordingly, Supplementary Table 4,5 are renamed to Supplementary Table 3,4.

Additional revisions:

(1) Section 2.5 “Young weak DBSs may have greatly promoted recent human evolution” is moved into Supplementary Note 3 (which now has the subtitle “Target genes with specific DBS features are enriched in specific functions”), because this section is short and lacking sufficient cross-validation.

(2) Considerable minor revisions of sentences have been made.

(3) Since there are many supplementary figures, the main text now cites only Supplementary Notes, as the reader can easily access supplementary figures in Supplementary Notes.